# Extending the Design Space of Graph Neural Networks by Rethinking Folklore Weisfeiler-Lehman

**Jiarui Feng**[1]  **Lecheng Kong**[1]  **Hao Liu**[1]  **Dacheng Tao**[2]  **Fuhai Li**[1]
**Muhan Zhang**[3*]  **Yixin Chen**[1]
{feng.jiarui, jerry.kong, liuhao, fuhai.li, ychen25}@wustl.edu,
dacheng.tao@gmail.com, muhan@pku.edu.cn
[1]Washington University in St. Louis  [2]JD Explore Academy  [3]Peking University

## Abstract

Message passing neural networks (MPNNs) have emerged as the most popular framework of graph neural networks (GNNs) in recent years. However, their expressive power is limited by the 1-dimensional Weisfeiler-Lehman (1-WL) test. Some works are inspired by $k$-WL/FWL (Folklore WL) and design the corresponding neural versions. Despite the high expressive power, there are serious limitations in this line of research. In particular, (1) $k$-WL/FWL requires at least $O(n^k)$ space complexity, which is impractical for large graphs even when $k = 3$; (2) The design space of $k$-WL/FWL is rigid, with the only adjustable hyper-parameter being $k$. To tackle the first limitation, we propose an extension, $(k, t)$-FWL. We theoretically prove that even if we fix the space complexity to $O(n^k)$ (for any $k \geq 2$) in $(k, t)$-FWL, we can construct an expressiveness hierarchy up to solving the graph isomorphism problem. To tackle the second problem, we propose $k$-FWL+, which considers any equivariant set as neighbors instead of all nodes, thereby greatly expanding the design space of $k$-FWL. Combining these two modifications results in a flexible and powerful framework $(k, t)$-FWL+. We demonstrate $(k, t)$-FWL+ can implement most existing models with matching expressiveness. We then introduce an instance of $(k, t)$-FWL+ called Neighborhood²-FWL (N²-FWL), which is practically and theoretically sound. We prove that N²-FWL is no less powerful than 3-WL, and can encode many substructures while only requiring $O(n^2)$ space. Finally, we design its neural version named **N²-GNN** and evaluate its performance on various tasks. N²-GNN achieves record-breaking results on ZINC-Subset (**0.059**) outperforming previous SOTA results by 10.6%. Moreover, N²-GNN achieves new SOTA results on the BREC dataset (**71.8%**) among all existing high-expressive GNN methods.

## 1   Introduction

In recent years, graph neural networks (GNNs) have become one of the most popular and powerful methods for graph representation learning, following a message passing framework [1–4]. However, the expressive power of message passing GNNs is bounded by the one-dimensional Weisfeiler-Lehman (1-WL) test [5, 6]. As a result, numerous efforts have been made to design GNNs with higher expressive power. We provide a more detailed discussion in Section 5.

Several works have drawn inspiration from the $k$-dimensional Weisfeiler-Lehman ($k$-WL) or Folklore Weisfeiler-Lehman ($k$-FWL) test [7] and developed corresponding neural versions [6, 8–10]. However, $k$-WL/FWL has two inherent limitations. First, while the expressive power increases

---

*Corresponding author

with higher values of $k$, the space and time complexity also grows exponentially, requiring $O(n^k)$ space complexity and $O(n^{k+1})$ time complexity, which makes it impractical even when $k = 3$. Thus, the question arises: **Can we retain high expressiveness without exploding both time and space complexities?** Second, the design space of WL-based algorithms is rigid, with the only adjustable hyper-parameter being $k$. However, there is a significant gap in expressive power even between consecutive values of $k$, making it hard to fine-tune the tradeoffs. Moreover, increasing the expressive power does not necessarily translate into better real-world performance, as it may lead to overfitting [8, 11]. Although some works try to tackle this problem [8, 10], there is still limited understanding of **how to expand the design space of the original $k$-FWL to a broader space** that enables us to identify the most appropriate instance to match the complexity of real-world tasks.

To tackle the first limitation, we notice that $k$-FWL and $k$-WL have the same space complexity but $k$-FWL can achieve the same expressive power as $(k+1)$-WL. We found the key component that allows $k$-FWL to have stronger power is the tuple aggregation style. Enlightened by this observation, we propose $(k, t)$-FWL, which extends the tuple aggregation style in $k$-FWL. Specifically, in the original $k$-FWL, a neighbor of a $k$-tuple is defined by iteratively replacing its $i$-th element with a node $u$, and $u$ traverses all nodes in the graph. In $(k, t)$-FWL, we extend a single node $u$ to a $t$-tuple of nodes and carefully design a replacement scheme to insert the $t$-tuple into a $k$-tuple to define its neighbor. We demonstrate that even with a fixed space complexity of $O(n^k)$ (for any $k \geq 2$), $(k, t)$-FWL can construct an expressive hierarchy capable of solving the graph isomorphism problem. To deal with the second limitation, we revisit the definition of neighborhood in $k$-FWL. Inspired by previous works [8, 12] which consider only local neighbors (i.e., $u$ must be connected to the $k$-tuple) instead of global neighbors in $k$-WL/FWL, we find that the neighborhood (i.e., which $u$ are used to construct a $k$-tuple's neighbors) can actually be extended to any equivariant set related to the $k$-tuple, resulting in $k$-FWL+. Combining the two modifications leads to a novel and powerful FWL-based algorithm $(k, t)$-FWL+. $(k, t)$-FWL+ is highly flexible and can be used to design different versions to fit the complexity of real-world tasks. Based on the proposed $(k, t)$-FWL+ framework, we implement many different instances that are closely related to existing powerful GNN/WL models, further demonstrating the flexibility of $(k, t)$-FWL+.

Finally, we propose an instance of $(2, 2)$-FWL+ named **Neighborhood$^2$-FWL** that is both theoretically expressive and practically powerful. It considers the local neighbors of both two nodes in a 2-tuple. Despite having a space complexity of $O(n^2)$, which is lower than 3-WL's $O(n^3)$ space complexity, this instance can still partially outperform 3-WL and is able to count many substructures. We implement a neural version named **Neighborhood$^2$-GNN (N$^2$-GNN)** and evaluate its performance on various synthetic and real-world datasets. Our results demonstrate that N$^2$-GNN outperforms existing SOTA methods across most tasks. Particularly, it achieves **0.059** in ZINC-Subset, surpassing existing state-of-the-art models by significant margins. Meanwhile, it achieves **71.8%** on the BREC dataset, the new SOTA among all existing high-expressive GNN methods.

## 2 Preliminaries

**Notations.** Let $\{\cdot\}$ denote a set, $\{\!\{\cdot\}\!\}$ denote a multiset (a set that allows repetition), and $(\cdot)$ denote a tuple. As usual, let $[n] = \{1, 2, \ldots, n\}$. Let $G = (V(G), E(G), l_G)$ be an undirected, colored graph, where $V(G) = [n]$ is the node set with $n$ nodes, $E(G) \subseteq V(G) \times V(G)$ is the edge set, and $l_G: V(G) \to C$ is the graph coloring function with $C = \{c_1, \ldots, c_d\}$ denote a set of $d$ distinct colors. Let $\mathcal{N}_k(v)$ denote a set of nodes within $k$ hops of node $v$ and $Q_k(v)$ denote the $k$-th hop neighbors of node $v$ and we have $\mathcal{N}_k(v) = \bigcup_{i=0}^{k} Q_i(v)$. Let $\text{SPD}(u, v)$ denote the shortest path distance between $u$ and $v$. We use $x_v \in \mathbb{R}^{d_n}$ to denote attributes of node $v \in V(G)$ and $e_{uv} \in \mathbb{R}^{d_e}$ to denote attributes of edge $(u, v) \in E(G)$. They are usually the one-hot encoding of the node and edge color respectively. We say that two graphs $G$ and $H$ are *isomorphic* (denoted as $G \simeq H$) if there exists a bijection $\varphi: V(G) \to V(H)$ such that $\forall u, v \in V(G), (u, v) \in E(G) \Leftrightarrow (\varphi(u), \varphi(v)) \in E(H)$ and $\forall v \in V(G), l_G(v) = l_H(\varphi(v))$. Denote $V(G)^k$ the set of $k$-tuples of vertices and $\mathbf{v} = (v_1, \ldots, v_k) \in V(G)^k$ a $k$-tuple of vertices. Let $S_n$ denote the permutation group of $[n]$ and $g \in S_n : [n] \to [n]$ be a particular permutation. When a permutation $g \in S_n$ operates on any target $X$, we denote it by $g \cdot X$. Particularly, a permutation operating on an edge set $E(G)$ is $g \cdot E(G) = \{(g(u), g(v))|(u, v) \in E(G)\}$. A permutation operating on a $k$-tuple $\mathbf{v}$ is $g \cdot \mathbf{v} = (g(v_1), \ldots, g(v_k))$. A permutation operating on a graph is $g \cdot G = (g \cdot V(G), g \cdot E(G), g \cdot l_G)$.

$k$**-dimensional Weisfeiler-Lehman test.** The $k$-dimensional Weisfeiler-Lehman ($k$-WL) test is a family of algorithms used to test graph isomorphism. There are two variants of the $k$-WL test: $k$-WL and $k$-FWL (Folklore WL). We first describe the procedure of 1-WL, which is also called the color refinement algorithm [7]. Let $\mathcal{C}^0_{1wl}(v) = l_G(v)$ be the initial color of node $v \in V(G)$. At the $l$-th iteration, 1-WL updates the color of each node using the following equation:

$$\mathcal{C}^l_{1wl}(v) = \text{HASH}\left(\mathcal{C}^{l-1}_{1wl}(v), \{\!\{\mathcal{C}^{l-1}_{1wl}(u)|u \in Q_1(v)\}\!\}\right). \tag{1}$$

After the algorithm converges, a color histogram is constructed using the colors assigned to all nodes. If the color histogram is different for two graphs, then the two graphs are non-isomorphic. However, if the color histogram is the same for two graphs, they can still be non-isomorphic.

The $k$-WL and $k$-FWL, for $k \geq 2$, are generalizations of the 1-WL, which do not color individual nodes but node tuples $\mathbf{v} \in V(G)^k$. Let $\mathbf{v}_{w/j}$ be a $k$-tuple obtained by replacing the $j$-th element of $\mathbf{v}$ with $w$. That is $\mathbf{v}_{w/j} = (v_1, \ldots, v_{j-1}, w, v_{j+1}, \ldots, v_k)$. The main difference between $k$-WL and $k$-FWL lies in their aggregation way. For $k$-WL, the set of $j$-th neighbors of tuple $\mathbf{v}$ is denoted as $Q_j(\mathbf{v}) = \{\mathbf{v}_{w/j}|w \in V(G)\}$, $j \in [k]$. Instead, the $w$-neighbor of tuple $\mathbf{v}$ for $k$-FWL is denoted as $Q^F_w(\mathbf{v}) = (\mathbf{v}_{w/1}, \mathbf{v}_{w/2}, \ldots, \mathbf{v}_{w/k})$. Let $\mathcal{C}^0_{kwl}(\mathbf{v}) = \mathcal{C}^0_{kfwl}(\mathbf{v})$ be the initial color for $k$-WL and $k$-FWL, respectively. They are usually the isomorphism types of tuple $\mathbf{v}$. At the $l$-th iteration, $k$-WL and $k$-FWL update the color of each tuple according to the following equations:

$$\textbf{WL:} \quad \mathcal{C}^l_{kwl}(\mathbf{v}) = \text{HASH}\left(\mathcal{C}^{l-1}_{kwl}(\mathbf{v}), \left(\{\!\{\mathcal{C}^{l-1}_{kwl}(\mathbf{u})|\mathbf{u} \in Q_j(\mathbf{v})\}\!\} \mid j \in [k]\right)\right), \tag{2}$$

$$\textbf{FWL:} \quad \mathcal{C}^l_{kfwl}(\mathbf{v}) = \text{HASH}\left(\mathcal{C}^{l-1}_{kfwl}(\mathbf{v}), \{\!\{\left(\mathcal{C}^{l-1}_{kfwl}(\mathbf{u})|\mathbf{u} \in Q^F_w(\mathbf{v})\right)|w \in V(G)\}\!\}\right). \tag{3}$$

The procedure described above is repeated until convergence, resulting in a color histogram of the graph that can be compared with the histograms of other graphs. Prior research has shown that $(k{+}1)$-WL and $(k{+}1)$-FWL are strictly more powerful than $k$-WL and $k$-FWL, respectively, except in the case where 1-WL is equivalent to 2-WL. Additionally, it has been shown that $k$-FWL is equivalent to $(k{+}1)$-WL [13, 14, 6]. We leave the additional discussion on $k$-WL/FWL in Appendix A.

**Message passing neural networks.** Message Passing Neural Networks (MPNNs) are a type of GNNs that update node representations by iteratively aggregating information from their direct neighbors. Let $h^l_v$ be the output of MPNNs of node $v \in V(G)$ after $l$ layers and we let $h^0_v = x_v$. At $l$-th layer, the representation is updated by:

$$h^l_v = \mathbf{U}^l\left(h^{l-1}_v, \{\!\{m^l_{vu}|u \in Q_1(v)\}\!\}\right), \quad m^l_{vu} = \mathbf{M}^l\left(h^{l-1}_v, h^{l-1}_u, e_{vu}\right), \tag{4}$$

where $\mathbf{U}^l$ and $\mathbf{M}^l$ are learnable update and message functions, usually parameterized by multi-layer perceptrons (MLPs). After $L$ layers, MPNNs output the final representation $h^L_v$ for all nodes $v \in V(G)$. The graph-level representation is obtained by:

$$h_G = \mathbf{R}(\{\!\{h^L_v|v \in V(G)\}\!\}), \tag{5}$$

where $\mathbf{R}$ is a readout function. The expressive power of MPNNs is at most as powerful as 1-WL [5, 6].

## 3 Rethinking and extending the Folklore Weisfeiler-Lehman test

### 3.1 Rethinking and extending the aggregation style in $k$-FWL

Increasing $k$ can increase the expressive power of $k$-FWL for distinguishing graph structures. However, increasing $k$ brings significant memory costs, as $k$-FWL requires $O(n^k)$ memory. It becomes impractical even when $k = 3$ [9]. Therefore, it is natural to ask:

*Can we achieve higher expressive power without exploding the memory cost?*

To achieve this goal, we first notice that $k$-FWL has a higher expressive power than $k$-WL but only requires the same $O(n^k)$ memory cost. To understand why, let's use 2-WL and 2-FWL as examples and rewrite Equation (2) and Equation (3):

**2-WL:** $\mathcal{C}^l_{2wl}(v_1, v_2) = \text{HASH}\left(\mathcal{C}^{l-1}_{2wl}(v_1, v_2), \{\!\{\mathcal{C}^{l-1}_{2wl}(v_1, w)|w \in V(G)\}\!\}, \{\!\{\mathcal{C}^{l-1}_{2wl}(w, v_2)|w \in V(G)\}\!\}\right),$

**2-FWL:** $\mathcal{C}^l_{2fwl}(v_1, v_2) = \text{HASH}\left(\mathcal{C}^{l-1}_{2fwl}(v_1, v_2), \{\!\{\left(\mathcal{C}^{l-1}_{2fwl}(v_1, w), \mathcal{C}^{l-1}_{2fwl}(w, v_2)\right)|w \in V(G)\}\!\}\right),$

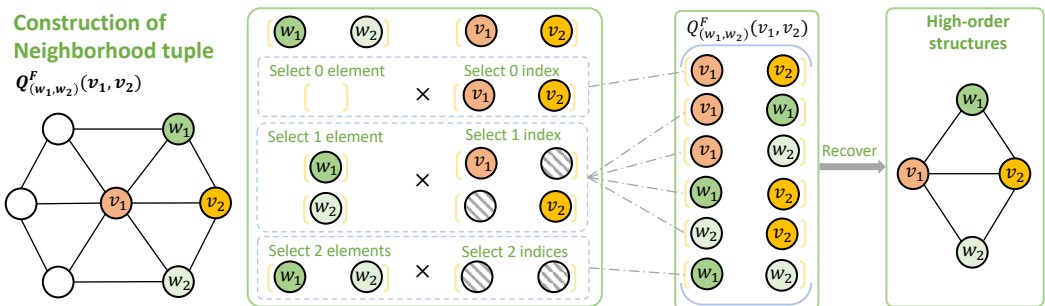

Figure 1: Illustration of the construction of neighborhood tuple $Q^F_{(w_1,w_2)}(v_1, v_2)$ in $(2,2)$-FWL. We sequentially select $0, 1, 2$ elements from $(w_1, w_2)$ to replace $0, 1, 2$ elements in $(v_1, v_2)$, resulting in three sub-tuple of length $1, 4, 1$, respectively. The final neighborhood tuple is the concatenation of three sub-tuples. We can easily recover high-order graph structures from the constructed neighborhood tuple with the isomorphism type of 2-tuples.

where $\mathcal{C}^l_{2wl}(v_1, v_2)$ and $\mathcal{C}^l_{2fwl}(v_1, v_2)$ are the color of tuple $(v_1, v_2)$ at iteration $l$ for 2-WL and 2-FWL, respectively. We can see the key difference between 2-WL and 2-FWL is that: in 2-FWL, a tuple of color $(\mathcal{C}^{l-1}_{2fwl}(v_1, w), \mathcal{C}^{l-1}_{2fwl}(w, v_2))$ is aggregated. While in 2-WL, the colors of nodes are considered in separate multisets. To understand why the first aggregation is more powerful, we can further rewrite the update equation of 2-FWL as follows:

$$\mathcal{C}^l_{2fwl}(v_1, v_2) = \text{HASH}\left(\{\!\!\{\left(\mathcal{C}^{l-1}_{2fwl}(v_1, v_2), \mathcal{C}^{l-1}_{2fwl}(v_1, w), \mathcal{C}^{l-1}_{2fwl}(w, v_2)\right) | w \in V(G)\}\!\!\}\right).$$

It is easy to verify that the above equation is equivalent to the original one. This means that 2-FWL updates the color of tuple $(v_1, v_2)$ by aggregating different tuples of $((v_1, v_2), (v_1, w), (w, v_2))$, which can be viewed as aggregating information of 3-tuples $(v_1, v_2, w)$. For example, $\text{HASH}(\mathcal{C}^0_{2fwl}(v_1, v_2), \mathcal{C}^0_{2fwl}(v_1, w), \mathcal{C}^0_{2fwl}(w, v_2))$ can fully recover the isomorphism type of tuple $(v_1, v_2, w)$ used in 3-WL. The above observations can be easily extended to $k$-FWL. The key insight here is that **tuple-style aggregation is the key to lifting the expressive power of $k$-FWL.** Since aggregating $k$-tuple can boost the expressive power of $k$-FWL to be equivalent to $(k+1)$-WL without the increase of space complexity, we may wonder, can we further extend the size of the tuple?

We first introduce the neighborhood tuple to extend the original $w$-neighbor used in $k$-FWL. Let $Q^F_\mathbf{w}(\mathbf{v})$ denote a neighborhood tuple related to $k$-tuple $\mathbf{v}$ and $t$-tuple $\mathbf{w}$. The neighborhood tuple contains all possible results such that we sequentially select $m \in [0, 1, \ldots, min(k,t)]$ elements from $\mathbf{w}$ to replace $m$ elements in $\mathbf{v}$. For each $m$, we first select all possible combinations of $m$-tuples (no repeated elements) from tuple $\mathbf{w}$ with a pre-defined order to form a new tuple, denoted as $P_m(\mathbf{w})$. At the same time, we select all possible combinations of $m$ indices from $\mathbf{i} = (1, 2, \ldots, k)$ with a pre-defined order to form another new tuple $P_m(\mathbf{i})$. Finally, we iterate all $(\mathbf{w}', \mathbf{i}') \in P_m(\mathbf{w}) \times P_m(\mathbf{i})$ to get a $|P_m(\mathbf{w})||P_m(\mathbf{i})|$-sub-tuple, where for each $(\mathbf{w}', \mathbf{i}')$ we replace the elements with indices $\mathbf{i}'$ in the original $\mathbf{v}$ with $\mathbf{w}'$. The final neighborhood tuple is the concatenation of all sub-tuples. Note that the pre-defined orders can be flexible as long as they are consistent for any $k$ and $t$.

Here is an example of a possible construction of a neighborhood tuple when $k = t = 2$. Let $Q^F_{(w_1,w_2)}(v_1, v_2)$ denote the neighborhood tuple. In this case, we need to enumerate $m$ in $[0, 1, 2]$. First, when $m = 0$, we select 0 elements from $(w_1, w_2)$ to replace $(v_1, v_2)$, resulting in a sub-tuple with only one element $((v_1, v_2))$. Next, when $m = 1$. we sequentially select $w_1$ and $w_2$, resulting in $P_1(\mathbf{w}) = (w_1, w_2)$. Similarly, we can get $P_1(\mathbf{i}) = (2, 1)$. Thus, the final sub-tuple is $((v_1, w_1), (v_1, w_2), (w_1, v_2), (w_2, v_2))$. The case of $m = 2$ is simple and we have the result sub-tuple be $((w_1, w_2))$. By concatenating all three sub-tuples, we have $Q^F_{(w_1,w_2)}(v_1, v_2) = ((v_1, v_2), (v_1, w_1), (v_1, w_2), (w_1, v_2), (w_2, v_2), (w_1, w_2))$. The construction procedure is shown in Figure 1. By default, we adopt the above order in $Q^F_\mathbf{w}(\mathbf{v})$ in the rest of the paper if $k = t = 2$. We leave a more formal definition of the neighborhood tuple $Q^F_\mathbf{w}(\mathbf{v})$ in Appendix A.

With the neighborhood tuple, we are ready to introduce $(k,t)$**-FWL**, which extends $k$-FWL by extending the tuple size in the update function. Let $\mathcal{C}^l_{ktfwl}(\mathbf{v})$ be the color of tuple $\mathbf{v}$ at iteration $l$ for $(k,t)$-FWL, the update function of $(k,t)$-FWL is:

$$(k,t)\text{-}\mathbf{FWL}: \quad \mathcal{C}^l_{ktfwl}(\mathbf{v}) = \text{HASH}(\mathcal{C}^{l-1}_{ktfwl}(\mathbf{v}), \{\!\!\{\left(\mathcal{C}^{l-1}_{ktfwl}(\mathbf{u})|\mathbf{u} \in Q^F_\mathbf{w}(\mathbf{v})\right) | \mathbf{w} \in V^t(G)\}\!\!\}_t), \quad (6)$$

where $\{\!\!\{\cdot\}\!\!\}_t$ is hierarchical multiset over $t$-tuples. In a hierarchical multiset $\{\!\!\{\mathbf{v}|\mathbf{v} \in V^t(G)\}\!\!\}_t$, elements are grouped hierarchically according to the node order of the tuple. For example, to construct $\{\!\!\{(v_1, v_2, v_3)|(v_1, v_2, v_3) \in V^3(G)\}\!\!\}_3$ from $\{\!\!\{(v_1, v_2, v_3)|(v_1, v_2, v_3) \in V^3(G)\}\!\!\}$, we first group together all elements with the same $v_2$ and $v_3$. That is, $\forall v_2, v_3 \in V(G)$, we denote $t(v_2, v_3) = \{\!\!\{(v_1, v_2, v_3)|v_1 \in V(G)\}\!\!\}$ as the grouped result. Next, use the similar procedure, we have $\forall v_3 \in V(G)$, $t(v_3) = \{\!\!\{t(v_2, v_3)|v_2 \in V(G)\}\!\!\}$. Finally, we group all possible $v_3 \in V(G)$ to get $\{\!\!\{(v_1, v_2, v_3)|(v_1, v_2, v_3) \in V^3(G)\}\!\!\}_3 = \{\!\!\{t(v_3)|v_3 \in V(G)\}\!\!\}$.

It is easy to see $(k, 1)$-FWL is equivalent to $k$-FWL under our definition of $Q_{\mathbf{w}}^F(\mathbf{v})$. Further, as we only need to maintain representations of all $k$-tuples. $(k, t)$-FWL has a fixed space complexity of $O(n^k)$. Here we show the expressive power of $(k, t)$-FWL.

**Proposition 3.1.** *For $k \geq 2$ and $t \geq 1$, if $t \geq n - k$, $(k, t)$-FWL can solve the graph isomorphism problems with the size of the graph less than or equal to $n$.*

**Theorem 3.2.** *For $k \geq 2$, $t \geq 1$, $(k, t)$-FWL is at most as powerful as $(k + t)$-WL. In particular, $(k, 1)$-FWL is as powerful as $(k + 1)$-WL.*

**Proposition 3.3.** *For $k \geq 2$, $t \geq 1$, $(k, t + 1)$-FWL is strictly more powerful than $(k, t)$-FWL; $(k + 1, t)$-FWL is strictly more powerful than $(k, t)$-FWL.*

We leave all formal proofs and complexity analysis in Appendix B. Briefly speaking, even if we fix the size of $k$, $(k, t)$-FWL can still construct an expressive hierarchy by varying $t$. Further, if $t$ is large enough, $(k, t)$-FWL can actually enumerate all possible combinations of tuples with the size of the graph, and thus equivalent to the relational pooling on graph [15]. It is worth noting that the size of $Q_{\mathbf{w}}^F(\mathbf{v})$ will grow exponentially with an increase in the size of $t$. However, the key contribution of $(k, t)$-FWL is that even when $k = 2$, $(k, t)$-FWL can still construct an expressive hierarchy for solving graph isomorphism problems. Therefore, high-order embedding may not be necessary for building high-expressivity WL algorithms. Note that our $(k, t)$-FWL is also different from subgraph GNNs such as $k, l$-WL [16] and $l$-OSAN [17], where $l$-tuples are labeled independently to enable learning $k$-tuple representations in all $l$-tuples' subgraphs, resulting in $O(n^{k+l})$ space complexity.

## 3.2 Rethinking and extending the aggregation scope of $k$-FWL

Another problem of $k$-FWL is its limited design space, as the only adjustable hyperparameter is $k$. It is well known that there is a huge gap in expressive power even if we increase from $k$ to $k + 1$. For example, 1-WL cannot count any cycle even with a length of 3, but 2-FWL can already count up to 7-cycle [18–20]. Moreover, increasing the expressive power does not always bring better performance when designing the corresponding neural version as it quickly leads to overfitting [6, 9, 11]. Therefore, we ask another question:

> *Can we extend the $k$-FWL to a more flexible and fine-grained design space?*

To address this issue, we identify that the inflexibility of $k$-FWL's design space arises from the definition of the neighbor used in the aggregation step. Unlike 1-WL, $k$-FWL lacks the concept of local neighbors and instead requires the aggregation of all $|V(G)|$ global neighbors to update the color of each tuple $\mathbf{v}$. Recently, some works have extended $k$-WL by incorporating local information [8, 12]. Inspired by previous works, we find that the definition of neighbor can actually be much more flexible than just considering local neighbors or global neighbors. Specifically, for each $k$-tuple $\mathbf{v}$ in graph $G$, we define equivariant set $ES(\mathbf{v})$ to be the neighbors set of tuple $\mathbf{v}$ and propose $k$-**FWL+**.

**Definition 3.4.** *An **equivariant set** $ES(\mathbf{v})$ is a set of nodes related to $\mathbf{v}$ and equivariant given the permutation $g \in S_n$. That is, $\forall w \in ES(\mathbf{v})$ in graph $G$ implies $g(w) \in ES(g \cdot \mathbf{v})$ in graph $g \cdot G$.*

Some nature equivariant sets $ES(v)$ including $V(G)$, $\mathcal{N}_k(v)$, and $Q_k(v)$, etc. Let $\mathcal{C}_{kfwl+}^l(\mathbf{v})$ be the color of tuple $\mathbf{v}$ at iteration $l$ for $k$-FWL+, we have:

$$k\text{-}\textbf{FWL+:} \quad \mathcal{C}_{kfwl+}^l(\mathbf{v}) = \text{HASH}(\mathcal{C}_{kfwl+}^{l-1}(\mathbf{v}), \{\!\!\{\left(\mathcal{C}_{kfwl+}^{l-1}(\mathbf{u})|\mathbf{u} \in Q_w^F(\mathbf{v})\right)|w \in ES(\mathbf{v})\}\!\!\}). \quad (7)$$

The key of $k$-FWL+ is that the equivariant set $ES(\mathbf{v})$ can be any set of nodes as long as it is equivariant to any permutation $g \in S_n$. For example, if $ES(\mathbf{v}) = V(G)$, the $k$-FWL+ will reduce to the original $k$-FWL. Instead, if $ES(\mathbf{v}) = \bigcup_{i=0}^k \mathcal{N}_1(v_i)$, it becomes the localized $k$-FWL [12]. We can also design other more innovative equivariant sets $ES(\mathbf{v})$. For example, denote $\mathcal{SP}(\mathbf{v})$ to be a set that contains all nodes in the shortest paths between any $v_i, v_j \in \mathbf{v}$. It is still a valid equivariant set of tuple $\mathbf{v}$. We can see that the design space of $k$-FWL+ is much broader than $k$-FWL.

### 3.3 $(k,t)$**-FWL+: A flexible and powerful extension of** $k$**-FWL**

In this section, we propose $(k,t)$**-FWL+**, which combines both $(k,t)$-FWL in Equation (6) and $k$-FWL+ in Equation (7). Let $\mathcal{C}^l_{ktf+}(\mathbf{v})$ be the color of tuple $\mathbf{v}$ at iteration $l$ for $(k,t)$-FWL+, the color update equation of $(k,t)$-FWL+ is:

$$(k,t)\text{-}\textbf{FWL+:} \quad \mathcal{C}^l_{ktf+}(\mathbf{v}) = \text{HASH}(\mathcal{C}^{l-1}_{ktf+}(\mathbf{v}), \{\!\{ \Big( \mathcal{C}^{l-1}_{ktf+}(\mathbf{u})|\mathbf{u} \in Q^F_{\mathbf{w}}(\mathbf{v}) \Big) \,|\mathbf{w} \in ES^t(\mathbf{v}) \}\!\}_t), \quad (8)$$

where $ES^t(\mathbf{v}) = ES_1(\mathbf{v}) \times ES_2(\mathbf{v}) \times \cdots \times ES_t(\mathbf{v})$ and $ES_i(\mathbf{v})$ can be any $ES(\mathbf{v})$. $(k,t)$**-FWL+** integrates advantages from both $(k,t)$-FWL and $k$-FWL+. First, by extending $V^t(G)$ to $ES^t(\mathbf{v})$, we can hugely reduce the time complexity of $(k,t)$-FWL as $ES^t(\mathbf{v})$ usually contains much less elements than $V^t(G)$. Second, by extending the tuple size in aggregation, we only require a small $k$ (even with $k = 2$) to achieve high expressive power instead of being bounded by $k$-FWL. Finally, the most important advantage is that the design space of $(k,t)$-FWL+ is much more flexible than $k$-FWL and we can design the most suitable instance for different tasks. However, as the expressive power of $(k,t)$-FWL+ highly relies on the choice of $ES^t(\mathbf{v})$, it is hard to analyze the expressive power of $(k,t)$-FWL+ in a systematic way. Instead, in the next section, we provide some practical implementations of $(k,t)$-FWL+ that can cover many existing powerful models.

### 3.4 Gallery of $(k,t)$**-FWL+ instances**

In this section, we provide some practical designs of $ES^t(\mathbf{v})$ that are closely correlated to many existing powerful GNN/WL models. Note that any model directly follows the original $k$-FWL like PPGN [9] can obviously be implemented by $(k,t)$-FWL+ as $(k,t)$-FWL+ also includes the original $k$-FWL. We specify that $k$-tuple $\mathbf{v} = (v_1, v_2, \ldots, v_k)$ to avoid any confusion.

**Proposition 3.5.** *Let $t = 1$, $k = 2$, and $ES(\mathbf{v}) = \mathcal{N}_1(v_1) \cup \mathcal{N}_1(v_2)$, the corresponding $(k,t)$-FWL+ instance is equivalent to SLFWL(2) [12] and strictly more powerful than any existing node-based subgraph GNNs.*

**Proposition 3.6.** *Let $t = 1$, $k = k$, and $ES(\mathbf{v}) = \bigcup_{i=1}^k Q_1(v_i)$, the corresponding $(k,t)$-FWL+ instance is more powerful than $\delta$-$k$-LWL [8].*

**Proposition 3.7.** *Let $t = 2$, $k = 2$, and $ES^2(\mathbf{v}) = (Q_1(v_1) \cap Q_1(v_2)) \times (Q_1(v_1) \cap Q_1(v_2))$, the corresponding $(k,t)$-FWL+ instance is more powerful than GraphSNN [21].*

**Proposition 3.8.** *Let $t = 2$, $k = 2$, and $ES^2(\mathbf{v}) = Q_1(v_2) \times Q_1(v_1)$, the corresponding $(k,t)$-FWL+ instance is more powerful than edge-based subgraph GNNs like $I^2$-GNN [19].*

**Proposition 3.9.** *Let $t = 1$, $k = 2$, and $ES(\mathbf{v}) = Q_{SPD(v_1,v_2)}(v_1) \cap Q_1(v_2)$, the corresponding $(k,t)$-FWL+ instance is at most as powerful as KP-GNN [22] with the peripheral subgraph encoder as powerful as 1-WL.*

**Proposition 3.10.** *Let $t = 2$, $k = 2$, and $ES^2(\mathbf{v}) = Q_1(v_2) \times \mathcal{SP}(v_1, v_2)$, the corresponding $(k,t)$-FWL+ is more powerful than GDGNN [23].*

We leave all formal proofs and discussion on complexity in Appendix C. Significantly, Proposition 3.8 indicates that $(k,t)$-FWL+ can be more powerful than edge-based subgraph GNNs but only require $O(n^2)$ space, strictly lower than $O(nm)$ space used in edge-based subgraph GNNs, where $m$ is the number of edges in the graph. However, most existing works employ MPNNs as their basic encoder, which differs from the tuple aggregation in $(k,t)$-FWL+. Thus it is still non-trivial to find an instance that can exactly fit many existing works. Nevertheless, given the strict hierarchy between node-based subgraph GNNs and SLFWL(2) [12], we conjecture there also exists a strict hierarchy between the $(k,t)$-FWL+ instances and corresponding existing works based on MPNNs and leave the detailed proof in our future works.

## 4 Neighborhood²-GNN: a practical and powerful $(2,2)$-FWL+ implementation

Due to the extremely flexible and broad design space of $(k,t)$-FWL+, it is hard to evaluate all possible instances. Therefore, we focus on a particular instance that is both theoretically and practically sound. All detailed proof and complexity analysis related to this section can be found in Appendix D.

## 4.1 Neighborhood$^2$-FWL

In this section, we propose a practical and powerful implementation of $(2,2)$-FWL+ named Neighborhood$^2$-FWL (N$^2$-FWL). Let $\mathcal{C}^l_{n^2fwl}(\mathbf{v})$ denote the color of tuple $\mathbf{v}$ at iteration $l$ for N$^2$-FWL, the color is updated by:

$$\textbf{N}^2\textbf{-FWL:} \quad \mathcal{C}^l_{n^2fwl}(\mathbf{v}) = \text{HASH}(\mathcal{C}^{l-1}_{n^2fwl}(\mathbf{v}), \{\!\{ \left( \mathcal{C}^{l-1}_{n^2fwl}(\mathbf{u}) | \mathbf{u} \in Q^F_{\mathbf{w}}(\mathbf{v}) \right) | \mathbf{w} \in N^2(\mathbf{v}) \}\!\}_2), \quad (9)$$

where $N^2(\mathbf{v}) = (\mathcal{N}_1(v_2) \times \mathcal{N}_1(v_1)) \cap (\mathcal{N}_h(v_1) \cap \mathcal{N}_h(v_2))^2$ and $h$ is the number of hop. Briefly speaking, N$^2$-FWL considers neighbors of both node $v_1$ and $v_2$ within the overlapping $h$-hop subgraph between $v_1$ and $v_2$. In the following, we show the expressive power and counting power of N$^2$-FWL.

**Theorem 4.1.** *Given $h$ is large enough, N$^2$-FWL is more powerful than SLFWL(2) [12] and edge-based subgraph GNNs.*

**Corollary 4.2.** *N$^2$-FWL is strictly more powerful than all existing node-based subgraph GNNs and no less powerful than 3-WL.*

**Theorem 4.3.** *N$^2$-FWL can count up to (1) 6-cycles; (2) all connected graphlets with size 4; (3) 4-paths at node level.*

## 4.2 Neighborhood$^2$-GNN

In this section, we provide a neural version of N$^2$-FWL called Neighborhood$^2$-GNN (N$^2$-GNN). Let $h^l_{\mathbf{v}}$ be the output of N$^2$-GNN of tuple $\mathbf{v}$ at layer $l$ and $h^0_{\mathbf{v}}$ encode the isomorphism type of tuple $\mathbf{v}$. At the $l$-th layer, the representation of each tuple is updated by:

$$h^l_{\mathbf{v}} = \mathbf{U}^l(h^{l-1}_{\mathbf{v}}, \{\!\{ m^l_{\mathbf{vw}} | \mathbf{w} \in N^2(\mathbf{v}) \}\!\}_2), \quad m^l_{\mathbf{vw}} = \mathbf{M}^l((h^{l-1}_{\mathbf{u}} | \mathbf{u} \in Q^F_{\mathbf{w}}(\mathbf{v}))). \quad (10)$$

After $L$ layers of message passing, the final graph representation is obtained by:

$$h_G = \mathbf{R}(\{\!\{ h^L_{\mathbf{v}} | \mathbf{v} \in V^2(G) \}\!\}_2). \quad (11)$$

We leave the detailed implementation in Appendix D. Finally, we show the expressive power of N$^2$-GNN as follows:

**Proposition 4.4.** *If $\mathbf{U}^l$, $\mathbf{M}^l$, and $\mathbf{R}$ are injective functions, there exist a N$^2$-GNN instance that can be as powerful as N$^2$-FWL.*

# 5 Related works

After the expressive power of MPNNs has been demonstrated to be limited by the 1-WL test [5, 6], significant research efforts have focused on breaking the limitations of MPNNs.

**Subgraph-based methods**. Subgraph-based methods aim to address the limitation of MPNNs by leveraging the fact that although some graphs may be difficult to distinguish, they contain distinct substructures that can be easily differentiated by MPNNs. To this end, various subgraph selection policies and aggregation strategies have been proposed to enhance the expressive power of MPNNs. These include K-ego subgraphs of nodes [24–28], distance labeling [29], graph elements deletion [30, 31], edge-based subgraph selection [19, 32, 33], etc. ESAN [34] summarizes different subgraph selection policies and proposes a general framework for subgraph-based GNNs. $l$-OSAN [17] further extends the subgraph-based model to $l$-tuples and $k, l$-WL [16] considers running $k$-WL on $l$-tuples subgraphs. KP-GNN [22] takes a slightly different view by considering the peripheral subgraph at each hop of nodes. Recently, Frasca et al. [27] proved that all node-based subgraph GNNs are bounded by 3-WL test. Zhang et al. [12] further show that they are even strictly less powerful than localized 2-FWL, highlighting the inherent limitations of neighbor aggregation-based methods compared to tuple-based aggregation.

**WL-based methods**. WL-based methods try to simulate $k$-WL/FWL test. In particular, Morris et al. [6] proposed $k$-GNNs to approximate $k$-WL test. To reduce the complexity of $k$-GNNs, Morris et al. [8] considered only local neighbors. Similarly, PPGN [9] adopted $k$-FWL instead of $k$-WL and proposed a tensor-based model with 3-WL power but requiring only $O(n^2)$ space. To further reduce complexity, KC-SetGNN [10] leveraged node sets rather than tuples and considered only

connected node sets. However, these approaches can still be impractical despite careful design. Our work hugely extends the design space of FWL-based methods and demonstrates that the resulting model can be expressive even with a lower space complexity cost.

**Permutation equivariant and invariant networks**. This line of research aims to leverage the permutation equivariant and invariant properties of graphs and devise corresponding architectures. For example, Maron et al. [35] proposed $k$-IGNs, which employ permutational equivariant linear layers with point-wise activation functions to the adjacency matrix. $k$-IGNs have been proven to have the same expressive power as $k$-WL [9, 36, 37]. Another approach utilizes the Reynold operator to consider all possible permutations on either global [15] or local [20] levels. However, they consider all possible permutations in order to achieve universality, which may lead to overfitting in real-world tasks and are more valuable from a theoretical perspective. In contrast, our framework is highly flexible and can be tailored to fit the complexity of real-world tasks.

**Feature-based methods**. Feature-based methods aim to incorporate graph-related features that cannot be computed by MPNNs. For instance, some methods add random features to nodes to break the symmetry [38, 39]. GSN [40] introduces substructure counting features into MPNNs. GD-WL [41] applies general distance features to enhance the expressiveness of MPNNs and shows it can solve the graph biconnectivity problem. GDGNN [23] integrates the geodesic information between node pairs and achieves a great balance between performance and efficiency. ESC-GNN [42] employs subgraph features to simulate subgraph GNNs without running them explicitly. Puny et al. [43] computes graph polynomial features and injects them into PPGN to achieve greater expressive power than 3-WL within $O(n^2)$ space. Although feature-based methods are highly efficient, they often suffer from overfitting in real-world tasks and produce suboptimal results.

# 6 Experiments

In this section, we conduct various experiments on synthetic and real-world tasks to verify the effectiveness of $N^2$-GNN. The details of all experiments can be found in Appendix E. Additional experimental results and ablation studies are included in Appendix F. The source code is provided in https://github.com/JiaruiFeng/N2GNN.

## 6.1 Expressive power

Table 1: Expressive power verification (Accuracy).

| Datasets | EXP | CSL | SR25 |
|---|---|---|---|
| $N^2$-GNN | 100 | 100 | 100 |

**Datasets**. To verify the expressive power of $N^2$-GNN, we select four synthetic datasets, all containing graphs that cannot be distinguished by 1-WL/MPNNs. The datasets we selected are (1) EXP [38], which consists of 600 pairs of non-isomorphic graphs that 1-WL fails to distinguish; (2) CSL [15], which contains 150 4-regular graphs divided into 10 isomorphism classes; and (3) SR25 [44], which contains 15 non-isomorphic strongly regular graphs, each has 25 nodes. Even 3-WL is unable to distinguish between these graphs. (4) BREC [45], which contains 400 non-isomorphic graph pairs that range from 1-WL-indistinguishable to 4-WL-indistinguishable.

**Models**. For EXP, CSL, and SR25, we only report results for $N^2$-GNN. For BREC, we compare $N^2$-GNN with $I^2$-GNN [19] as it is the current SOTA on BREC among all GNN methods.

**Results**. We report results in Table 1 and Table 2. For EXP and CSL, we report the average accuracy for 10-time cross-validation; For SR25, we report single-time accuracy. We can see $N^2$-GNN achieves perfect results on all three datasets, empirically verifying its expressive power. Notably, $N^2$-GNN is able to distinguish strongly regular graphs. This empirically verified Corollary 4.2. Moreover, $N^2$-GNN achieves the best result on the BREC dataset among all GNNs methods, surpassing the previous SOTA, $I^2$-GNN (See the complete comparison and further discussion in Appendix F). Despite this improved performance, $N^2$-GNN only requires at most $O(n^2)$ space complexity, which is less than 3-WL. This further demonstrates the superiority of $N^2$-GNN.

Table 2: Pair distinguishing accuracies on BREC

| Model | Basic Graphs (60) | | Regular Graphs (140) | | Extension Graphs (100) | | CFI Graphs (100) | | Total (400) | |
|---|---|---|---|---|---|---|---|---|---|---|
| | Number | Accuracy | Number | Accuracy | Number | Accuracy | Number | Accuracy | Number | Accuracy |
| $I^2$-GNN | 60 | 100% | 100 | 71.4% | 100 | 100% | 21 | 21% | 281 | 70.2% |
| $N^2$-GNN | 60 | 100% | 100 | 71.4% | 100 | 100% | 27 | 27% | **287** | **71.8%** |

Table 3: Evaluation on Counting Substructures (norm MAE), cells with MAE less than 0.01 are colored.

| Target | ID-GNN [28] | NGNN [24] | GIN-AK+ [25] | PPGN [9] | $I^2$-GNN [19] | $N^2$-GNN |
|---|---|---|---|---|---|---|
| Tailed Triangle | 0.1053 | 0.1044 | 0.0043 | 0.0026 | 0.0011 | 0.0025 |
| Chordal Cycle | 0.0454 | 0.0392 | 0.0112 | 0.0015 | 0.0010 | 0.0019 |
| 4-Clique | 0.0026 | 0.0045 | 0.0049 | 0.1646 | 0.0003 | 0.0005 |
| 4-Path | 0.0273 | 0.0244 | 0.0075 | 0.0041 | 0.0041 | 0.0042 |
| Tri.-Rec. | 0.0628 | 0.0729 | 0.1311 | 0.0144 | 0.0013 | 0.0055 |
| 3-Cycles | 0.0006 | 0.0003 | 0.0004 | 0.0003 | 0.0003 | 0.0002 |
| 4-Cycles | 0.0022 | 0.0013 | 0.0041 | 0.0009 | 0.0016 | 0.0024 |
| 5-Cycles | 0.0490 | 0.0402 | 0.0133 | 0.0036 | 0.0028 | 0.0039 |
| 6-Cycles | 0.0495 | 0.0439 | 0.0238 | 0.0071 | 0.0082 | 0.0075 |

## 6.2 Substructure counting

**Datasets**. To verify the substructure counting power of $N^2$-GNN, we select the synthetic dataset from [25, 19]. The dataset consists of 5000 randomly generated graphs from different distributions, which are split into training, validation, and test sets with a ratio of 0.3/0.2/0.5. The task is to perform node-level counting regression. Specifically, we choose tailed-triangle, chordal cycles, 4-cliques, 4-paths, triangle-rectangle, 3-cycles, 4-cycles, 5-cycles, and 6-cycles as target substructures.

**Models**. We compare $N^2$-GNN with the following baselines: Identity-aware GNN (ID-GNN) [28], Nested GNN (NGNN) [24], GNN-AK+ [25], PPGN [9], $I^2$-GNN [19]. Results for all baselines are reported from [19].

**Results**. We report the results for counting substructures in Table 3. All results are the average normalized test MAE of three runs with different random seeds. We colored all results that are less than 0.01, which is an indication of successful counting, the same as in [19]. We can see that $N^2$-GNN achieves MAE less than 0.01 among all different substructures/cycles, which empirically verified Theorem 4.3. Specifically, $N^2$-GNN achieves the best result on 4-path counting and comparable performance to $I^2$-GNN on most substructures. Moreover, $N^2$-GNN can count 4-clique extremely well, which is theoretically and empirically infeasible for 3-WL [42] and PPGN [9].

Table 4: MAE results on QM9 (smaller the better).

| Target | DTNN [46] | MPNN [46] | PPGN [9] | NGNN [24] | KP-GIN′ [22] | $I^2$-GNN [19] | $N^2$-GNN |
|---|---|---|---|---|---|---|---|
| $\mu$ | 0.244 | 0.358 | **0.231** | 0.433 | 0.358 | 0.428 | 0.333 |
| $\alpha$ | 0.95 | 0.89 | 0.382 | 0.265 | 0.233 | 0.230 | **0.193** |
| $\varepsilon_{\text{HOMO}}$ | 0.00388 | 0.00541 | 0.00276 | 0.00279 | 0.00240 | 0.00261 | **0.00217** |
| $\varepsilon_{\text{LUMO}}$ | 0.00512 | 0.00623 | 0.00287 | 0.00276 | 0.00236 | 0.00267 | **0.00210** |
| $\Delta\varepsilon$ | 0.0112 | 0.0066 | 0.00406 | 0.00390 | 0.00333 | 0.00380 | **0.00304** |
| $\langle R^2 \rangle$ | 17.0 | 28.5 | 16.7 | 20.1 | 16.51 | 18.64 | **14.47** |
| ZPVE | 0.00172 | 0.00216 | 0.00064 | 0.00015 | 0.00017 | 0.00014 | **0.00013** |
| $U_0$ | 2.43 | 2.05 | 0.234 | 0.205 | 0.0682 | 0.211 | **0.0247** |
| $U$ | 2.43 | 2.00 | 0.234 | 0.200 | 0.0696 | 0.206 | **0.0315** |
| $H$ | 2.43 | 2.02 | 0.229 | 0.249 | 0.0641 | 0.269 | **0.0182** |
| $G$ | 2.43 | 2.02 | 0.238 | 0.253 | 0.0484 | 0.261 | **0.0178** |
| $C_v$ | 0.27 | 0.42 | 0.184 | 0.0811 | 0.0869 | **0.0730** | 0.0760 |

## 6.3 Molecular properties prediction

**Datasets**. To evaluate the performance of $N^2$-GNN on real-world tasks, we select two popular molecular graphs datasets: QM9 [46, 50] and ZINC [51]. QM9 dataset contains over 130K molecules with 12 different molecular properties as the target regression task. The dataset is split into training, validation, and test sets with a ratio of 0.8/0.1/0.1. The ZINC dataset has two variants: ZINC-subset (12k graphs) and ZINC-full (250k graphs), and the task is graph regression. The training, validation, and test splits for the ZINC datasets are provided.

**Models**. For QM9, we report baseline results of DTNN and MPNN from [46]. We further adopt PPGN [9], NGNNs [24], KP-GIN′ [22], $I^2$-GNN [19]. We report results of PPGN, NGNN, and KP-GIN′ from [22] and results of $I^2$-GNN from [19]. For ZINC, the baseline including CIN [47], $\delta$-2-GNN [8], KC-SetGNN [10], PPGN [9], Graphormer-GD [41], GPS [48], Specformer [49], NGNN [24], GNN-AK-ctx [25], ESAN [34], SUN [27], KP-GIN′ [22], $I^2$-GNN [19], SSWL+ [12]. We report results of all baselines from [12, 22, 43, 10, 48, 49, 19].

Table 5: MAE results on ZINC (smaller the better).

| Model | # Param | ZINC-Subset Test MAE | ZINC-Full Test MAE |
|---|---|---|---|
| CIN [47] | ~100k | $0.079 \pm 0.006$ | $\mathbf{0.022 \pm 0.002}$ |
| $\delta$-2-GNN [8] | - | - | $0.042 \pm 0.003$ |
| KC-SetGNN [10] | - | $0.075 \pm 0.003$ | - |
| PPGN [9] | - | $0.079 \pm 0.005$ | $0.022 \pm 0.003$ |
| GD-WL [41] | 503k | $0.081 \pm 0.009$ | $0.025 \pm 0.004$ |
| GPS [48] | 424k | $0.070 \pm 0.004$ | - |
| Specformer [49] | ~500k | $0.066 \pm 0.003$ | - |
| NGNN [24] | ~500k | $0.111 \pm 0.003$ | $0.029 \pm 0.001$ |
| GNN-AK [25] | ~500k | $0.093 \pm 0.002$ | - |
| ESAN [34] | 446k | $0.097 \pm 0.006$ | $0.025 \pm 0.003$ |
| SUN [27] | 526k | $0.083 \pm 0.003$ | $0.024 \pm 0.003$ |
| KP-GIN$'$ [22] | 489k | $0.093 \pm 0.007$ | - |
| I$^2$-GNN [19] | - | $0.083 \pm 0.001$ | $0.023 \pm 0.001$ |
| SSWL+ [12] | 387k | $0.070 \pm 0.005$ | $\mathbf{0.022 \pm 0.002}$ |
| N$^2$-GNN | 316k/414k | $\mathbf{0.059 \pm 0.002}$ | $\mathbf{0.022 \pm 0.002}$ |

**Results**. For QM9, we report the single-time final test MAE in Table 4. We can see that N$^2$-GNN outperforms all other methods in almost all targets with a large margin. Particularly, we achieve 63.8%, 54.7%, 71.6%, and 63.2% improvement comparing to the second-best results on target $U_0$, $U$, $H$, and $G$ respectively. For ZINC-Subset and ZINC-Full, we report the average test MAE and standard deviation of 10 runs with different random seeds in Table 5. We can see N$^2$-GNN still surpasses all previous methods with only the smallest model parameter size among all baselines. It achieves 10.6% improvement over the second-best methods on ZINC-Subset and the same performance over the previous SOTA on ZINC-full. All these results demonstrate the superiority of N$^2$-GNN on real-world tasks.

## 7 Limitations, future works, and conclusions

**Limitations:** For $(k, t)$-FWL, although we can control the space complexity by fixing the $k$, the time complexity of $(k, t)$-FWL will grow exponentially with the increase of $t$ (as discussed in Appendix B). Therefore, the resulting model can still be impractical when $t$ is large. We believe this result is evidence of the "no free lunch" in developing a more expressive GNN model. For $(k, t)$-FWL+, although we demonstrate its capability by implementing many instances corresponding to different existing GNN models, it is still unclear what is the whole space of $(k, t)$-FWL+, especially the space of $ES(\mathbf{v})$. Moreover, the quantitative expressiveness analysis of $(k, t)$-FWL+ is still unexplored. Finally, for N$^2$-GNN, the practical complexity can still be unbearable, especially for dense graphs if the aggregation is implemented in a parallel way, as we need to aggregate many more neighbors for each tuple than normal MPNN. It also introduces the optimization issue on N$^2$-GNN, which makes it hard to achieve its theoretical expressiveness. In Appendix B, C, and D, we provide more detailed discussion on the limitation of $(k, t)$-FWL, $(k, t)$-FWL+, and N$^2$-GNN, respectively.

**Future works**: There are several directions that are worth further exploration. First, can we characterize the whole space of $ES(\mathbf{v})$ and thus can theoretically and quantitatively analyze the expressive power of $(k, t)$-FWL+. Further, how to design advanced model architecture to achieve the theoretical power of $(k, t)$-FWL+. Finally, can we provide a systematic implementation code package and practical usage of different downstream tasks such that researchers can easily develop an instance of $(k, t)$-FWL+ that can best fit their downstream tasks? We leave these to our future work.

**Conclusions:** In this work, we propose $(k, t)$-FWL+, a flexible and powerful extension of $k$-FWL. First, $(k, t)$-FWL+ expands the tuple aggregation style in $k$-FWL. We theoretically prove that given any fixed pace complexity of $O(n^k)$, $(k, t)$-FWL+ can still construct an expressiveness hierarchy up to solving the graph isomorphism problem. Second, $(k, t)$-FWL+ extends the global neighborhood definition in $k$-FWL to any equivariant set, which enables a finer-grained design space. We show that the $(k, t)$-FWL+ framework can implement many existing powerful GNN models with matching expressiveness. We further implement an instance named N$^2$-GNN which is both practically powerful and theoretically expressive. Theoretically, it partially outperforms 3-WL but only requires $O(n^2)$ space. Empirically, it achieves new SOTA on BREC, ZINC-Subset, and ZINC-Full datasets. We envision $(k, t)$-FWL+ can serve as a promising general framework for designing highly effective GNNs tailored to real-world problems.

## 8 Acknowledgments

Jiarui Feng, Lecheng Kong, Hao Liu, and Yixin Chen are supported by NSF grant CBE-2225809. Muhan Zhang is partially supported by the National Natural Science Foundation of China (62276003) and Alibaba Innovative Research Program.

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

# Appendix

## Table of Contents

# A Additional preliminaries

In this section, we provide additional preliminaries.

## A.1 Additional definitions

**Definition A.1.** When we say we **select a $t$-tuple $\mathbf{w}' = (w'_1, w'_2, \cdots, w'_t)$** from a $p$-tuple $\mathbf{v} = (v_1, v_2, \cdots, v_p)$ $(p \geq t)$, it means that we select $t$ unique indices $(i_1, i_2, \cdots, i_t)$ from $[p]$ such that $i_1 < i_2 < \cdots < i_t$. Next, to construct $\mathbf{w}'$, we let $w'_1 = v_{i_1}, \ldots, w'_t = v_{i_t}$. When we say we **select all possible $t$-tuple $\mathbf{w}'$** from a $p$-tuple $\mathbf{v}$, we enumerate all possible $t$ indices and construct $t$-tuples correspondingly. There are $\begin{pmatrix} p \\ t \end{pmatrix}$ different selection results and we denote $\tilde{P}_t(\mathbf{v})$ as the set of all possible selections. Finally, when we say **select all possible $t$-tuple $\mathbf{w}'$ from a $p$-tuple $\mathbf{v}$ with order**, we further assume that the selection process is taken with a predefined order such that the sequence of selected $t$ indices will be the same if we repeat the selection. For example, we can always assume we start from indices $(1, 2, \ldots, t)$. Next, fix other indices and enumerate all possible $i_t$ in ascending order. Then, increase $i_{t-1}$ by 1 and repeat the procedure. If we select all possible 2-tuple from $(v_1, v_2, v_3)$, the result is $((v_1, v_2), (v_1, v_3), (v_2, v_3))$. We denote $P_t(\mathbf{v})$ as the tuple of the selection result.

**Definition A.2.** **Neighborhood tuple** $Q^F_\mathbf{w}(\mathbf{v})$ is a generalization of neighbor set $Q^F_w(\mathbf{v})$ in $k$-FWL. Briefly speaking, the neighborhood tuple enumerates all possible combinations of elements in $\mathbf{v}$ and $\mathbf{w}$ follows a pre-defined order. Suppose the length of the tuple is $k$ and $t$ for $\mathbf{v}$ and $\mathbf{w}$, respectively. To construct the neighborhood tuple $Q^F_\mathbf{w}(\mathbf{v})$, we enumerate $m$ from 0 to $min(k, t)$. For each $m$, we construct a sub-tuple, and the final neighborhood tuple $Q^F_\mathbf{w}(\mathbf{v})$ is the concatenation of all sub-tuples. Given a particular $m$, the sub-tuple is constructed by selecting all possible $m$-tuple $\mathbf{w}' = (w'_1, \cdots, w'_m)$ from $t$-tuple $\mathbf{w}$ with order to construct a tuple $P_m(\mathbf{w})$. At the same time, we select all possible $m$ indices $\mathbf{i}' = (i'_1, \ldots, i'_m)$ from $\mathbf{i} = (1, 2, \ldots, k)$ with order to construct another tuple $P_m(\mathbf{i})$. Finally, for each $(\mathbf{w}', \mathbf{i}') \in P_m(\mathbf{w}) \times P_m(\mathbf{i})$, we add a tuple to the resulted sub-tuple by replacing $v_{i'_o}$ with $w'_o$ for $o = 1, 2, \ldots, m$. The neighborhood tuple is constructed by concatenating all sub-tuples. The length of the neighborhood tuple is $\sum_{m=0}^{min(k,t)} \begin{pmatrix} k \\ m \end{pmatrix} \times \begin{pmatrix} t \\ m \end{pmatrix}$.

**Definition A.3.** Let $F_1$ and $F_2$ be two color refinement algorithms, and denote $\mathcal{C}_i(G), i \in \{1, 2\}$ as the color histogram computed by $F_i$ for graph $G$ after convergence. We say:

- $F_1$ is **more powerful** than $F_2$, denoted as $F_2 \preceq F_1$, if for any pair of graph $G$ and $H$, $\mathcal{C}_1(G) = \mathcal{C}_1(H)$ also implies $\mathcal{C}_2(G) = \mathcal{C}_2(H)$.

- $F_1$ is **as powerful as** $F_2$, denoted as $F_1 \simeq F_2$, if we have $F_1 \preceq F_2$ and $F_2 \preceq F_1$.

- $F_1$ is **strictly more powerful** than $F_2$, denoted as $F_2 \prec F_1$, if we have $F_2 \preceq F_1$ and there exist graphs $G$ and $H$ such that $\mathcal{C}_1(G) \neq \mathcal{C}_1(H)$ and $\mathcal{C}_2(G) = \mathcal{C}_2(H)$.

- $F_1$ and $F_2$ are **incomparable**, denoted as $F_1 \nsim F_2$, if neither $F_1 \preceq F_2$ nor $F_2 \preceq F_1$ holds.

*Remark* A.4. To prove that $F_1$ is more powerful than $F_2$, one way is to show that the color of node/tuple cannot be further refined by $F_2$ if the color output from $F_1$ is already stable (convergence). This was also indicated in Remark B.2 in [12]. Note that the initial color of $F_1$ should be at least as powerful as the initial color of $F_2$. Namely, if we use the initial color to compute the color histogram, we have $F_2 \preceq F_1$.

## A.2 More on Weisfeiler-Lehman test and Folklore Weisfeiler-Lehman test

First, we restate the color update equation of both $k$-WL and $k$-FWL.

$$\textbf{WL:} \quad \mathcal{C}^l_{kwl}(\mathbf{v}) = \text{HASH}\left(\mathcal{C}^{l-1}_{kwl}(\mathbf{v}), \left(\{\!\!\{\mathcal{C}^{l-1}_{kwl}(\mathbf{u}) | \mathbf{u} \in Q_j(\mathbf{v})\}\!\!\} | j \in [k]\right)\right),$$

$$\textbf{FWL:} \quad \mathcal{C}^l_{kfwl}(\mathbf{v}) = \text{HASH}\left(\mathcal{C}^{l-1}_{kfwl}(\mathbf{v}), \{\!\!\{\left(\mathcal{C}^{l-1}_{kfwl}(\mathbf{u}) | \mathbf{u} \in Q^F_w(\mathbf{v})\right) | w \in V(G)\}\!\!\}\right).$$

The first step of both $k$-WL and $k$-FWL is to encoder the isomorphic type of tuple $\mathbf{v} = (v_1, v_2, ..., v_k)$ as initial color such that for any two tuples $\mathbf{u}, \mathbf{v} \in v^k(G)$, $\mathcal{C}^0_{kfwl}(\mathbf{v}) = \mathcal{C}^0_{kfwl}(\mathbf{u})$ and $\mathcal{C}^0_{kwl}(\mathbf{v}) =$

$\mathcal{C}^0_{kwl}(\mathbf{u})$ if and only if the ordered subgraphs induced by two tuples have the same isomorphism type. Namely, $\forall i,j \in [k]$ (1) $l_G(v_i) = l_G(u_i)$ and (2) $(v_i,v_j) \in E(G) \iff (u_i,u_j) \in E(G)$. Another thing is that the HSAH here is an injective function. As long as the input is different, the output of HASH is different. This will serve as a basic concept in the later proof. The algorithm converge means that $\forall \mathbf{v} \in V^k(G)$, we have $\mathcal{C}^{l+1}_{kwl}(\mathbf{v}) = \mathcal{C}^l_{kwl}(\mathbf{v})$ or $\mathcal{C}^{l+1}_{kfwl}(\mathbf{v}) = \mathcal{C}^l_{kfwl}(\mathbf{v})$. Denote the convergent iteration as $\infty$. After the algorithm convergence, the color histogram can be computed by another injective function:

$$\mathcal{C}_{kfwl}(G) = \text{HASH}(\{\!\{\mathcal{C}^\infty_{kfwl}(\mathbf{v})|\mathbf{v} \in V^k(G)\}\!\}). \tag{12}$$

Here we use $k$-FWL as an example, but the procedure for other color refinement algorithms is the same.

## B   Additional discussion on $(k,t)$-FWL

### B.1   Detailed proofs for $(k,t)$-FWL

In this section, we provide all detailed proofs related to $(k,t)$-FWL. Let $\mathbf{p} = (\mathbf{v},\mathbf{w})$ be a new tuple that is the concatenation of a $k$-tuple $\mathbf{v}$ and a $t$-tuple $\mathbf{w}$. Let $\mathcal{C}^0_{(k+t)wl}(\mathbf{p})$ be the isomorphic type of tuple $\mathbf{p}$. We start to prove the expressive power of $(k,t)$-FWL with the first lemma:

**Lemma B.1.** *Suppose $k \geq 2$ and $t \geq 1$. Given two graphs $G$, $H$, two tuples $\mathbf{v}_1 \in V^k(G), \mathbf{v}_2 \in V^k(H)$, and another two tuples $\mathbf{w}_1 \in V^t(G), \mathbf{w}_2 \in V^t(H)$. Let $\mathbf{p}_1 = (\mathbf{v}_1,\mathbf{w}_1)$, $\mathbf{p}_2 = (\mathbf{v}_2,\mathbf{w}_2)$. We have $\mathcal{C}^0_{(k+t)wl}(\mathbf{p}_1) = \mathcal{C}^0_{(k+t)wl}(\mathbf{p}_2) \iff \left(\mathcal{C}^0_{ktfwl}(\mathbf{u}_1)|\mathbf{u}_1 \in Q^F_{\mathbf{w}_1}(\mathbf{v}_1)\right) = \left(\mathcal{C}^0_{ktfwl}(\mathbf{u}_2)|\mathbf{u}_2 \in Q^F_{\mathbf{w}_2}(\mathbf{v}_2)\right)$.*

*Proof.* First, we prove the forward direction. That is, if $\mathcal{C}^0_{(k+t)wl}(\mathbf{p}_1) = \mathcal{C}^0_{(k+t)wl}(\mathbf{p}_2)$, we have $\left(\mathcal{C}^0_{ktfwl}(\mathbf{u}_1)|\mathbf{u}_1 \in Q^F_{\mathbf{w}_1}(\mathbf{v}_1)\right) = \left(\mathcal{C}^0_{ktfwl}(\mathbf{u}_2)|\mathbf{u}_2 \in Q^F_{\mathbf{w}_2}(\mathbf{v}_2)\right)$. Recall the definition of the initial color, if $\mathcal{C}^0_{(k+t)wl}(\mathbf{p}_1) = \mathcal{C}^0_{(k+t)wl}(\mathbf{p}_2)$, we must have the ordered subgraph induced by $\mathbf{p}_1$ is isomorphic to the ordered subgraph induced by $\mathbf{p}_2$. Namely, $\forall i,j \in [k+t]$, $(p_{1i},p_{1j}) \in E(G) \iff (p_{2i},p_{2j}) \in E(H)$ and $l_G(p_{1i}) = l_G(p_{2i})$. Next, given Definition A.2, we know that for any neighborhood tuple, there exists a single indices mapping function $O$ that given $\mathbf{p} = (\mathbf{v},\mathbf{w})$, for any $\mathbf{u} \in Q^F_{\mathbf{w}}(\mathbf{v})$, we have $O(\mathbf{u},Q^F_{\mathbf{w}}(\mathbf{v})) = (o_1,\ldots,o_k)$ such that $(u_1,\ldots,u_k) = (p_{o_1},\ldots,p_{o_k})$. Further, the construction of the neighborhood tuple follows the same ordering rule no matter the input. This means that, for both $\mathbf{u}_1 \in Q^F_{\mathbf{w}_1}(\mathbf{v}_1)$ and $\mathbf{u}_2 \in Q^F_{\mathbf{w}_2}(\mathbf{v}_2)$, as long as $\mathbf{u}_1$ and $\mathbf{u}_2$ are in the same position of neighborhood tuple $Q^F_{\mathbf{w}_1}(\mathbf{v}_1)$ and $Q^F_{\mathbf{w}_2}(\mathbf{v}_2)$, we must have $O(\mathbf{u}_1,Q^F_{\mathbf{w}_1}(\mathbf{v}_1)) = O(\mathbf{u}_2,Q^F_{\mathbf{w}_2}(\mathbf{v}_2)) = (o_1,\ldots,o_k)$. Finally, as for any $o_i$ and $o_j$, we have $(p_{1o_i},p_{1o_j}) \in E(G) \iff (p_{2o_i},p_{2o_j}) \in E(H)$, $l_G(p_{1o_i}) = l_G(p_{2o_i})$, and $l_G(p_{1o_j}) = l_G(p_{2o_j})$, we conclude that $\mathcal{C}^0_{ktfwl}(\mathbf{u}_1) = \mathcal{C}^0_{ktfwl}(\mathbf{u}_2)$. As it is the same for any $\mathbf{u}_1 \in Q^F_{\mathbf{w}_1}(\mathbf{v}_1)$ and $\mathbf{u}_2 \in Q^F_{\mathbf{w}_2}(\mathbf{v}_2)$, it must holds that $\left(\mathcal{C}^0_{ktfwl}(\mathbf{u}_1)|\mathbf{u}_1 \in Q^F_{\mathbf{w}_1}(\mathbf{v}_1)\right) = \left(\mathcal{C}^0_{ktfwl}(\mathbf{u}_2)|\mathbf{u}_2 \in Q^F_{\mathbf{w}_2}(\mathbf{v}_2)\right)$. The proof of the backward direction is similar to the forward direction. As for any $i,j \in [k+t]$, we can find $p_{1i},p_{1j}$ exist in some $\mathbf{u}_1 \in Q^F_{\mathbf{w}_1}(\mathbf{v}_1)$ and $p_{2i},p_{2j}$ in the corresponding $\mathbf{u}_2 \in Q^F_{\mathbf{w}_2}(\mathbf{v}_2)$. Since we have $\mathcal{C}^0_{ktfwl}(\mathbf{u}_1) = \mathcal{C}^0_{ktfwl}(\mathbf{u}_2)$, we can easily conclude that $(p_{1i},p_{1j}) \in E(G) \iff (p_{2i},p_{2j}) \in E(G)$, $l_G(p_{1i}) = l_G(p_{2i})$, and $l_G(p_{1j}) = l_G(p_{2j})$. This concludes the proof. $\square$

Lemma B.1 indicates that $(k,t)$-FWL can recover the isomorphism type of high-order tuple, which is impossible in the original $k$-FWL. Therefore, as long as $t$ is large enough, we can recover the isomorphism type of the whole graph. Further, Lemma B.1 also implies that we can resort to high-order tuples to prove the expressive power of $(k,t)$-FWL. Define the $H_t(\mathbf{v}) = \{(\mathbf{v},\mathbf{w})|\mathbf{w} \in V^t(G)\}$, we slightly rewrite Equation (6):

$$\mathcal{C}^l_{ktfwl}(\mathbf{p}) = \text{HASH}\left(\left(\mathcal{C}^{l-1}_{ktfwl}(\mathbf{u})|\mathbf{u} \in Q^F_{\mathbf{w}}(\mathbf{v})\right)\right), \tag{13}$$

$$\mathcal{C}^l_{ktfwl}(\mathbf{v}) = \text{HASH}\left(\mathcal{C}^{l-1}_{ktfwl}(\mathbf{v}), \{\!\{\mathcal{C}^l_{ktfwl}(\mathbf{p})|\mathbf{p} \in H_t(\mathbf{v})\}\!\}_t\right). \tag{14}$$

It is easy to verify the rewritten will not affect the expressive power of $(k,t)$-FWL. Now, we are ready to prove the Proposition 3.1 in the main paper. We restate it here:

**Proposition B.2.** *For $k \geq 2$ and $t \geq 1$, if $t \geq n - k$, $(k,t)$-FWL can solve the graph isomorphism problems with the size of the graph less than or equal to $n$.*

*Proof.* Here we show that with only 1 iteration, $(k,t)$-FWL can solve the graph isomorphism problem with the size of graph $n \leq t + k$. Given Lemma B.1, for any $\mathbf{v} \in V^k(G)$, we have $\mathcal{C}^1_{ktfwl}(\mathbf{p}_1) = \mathcal{C}^1_{ktfwl}(\mathbf{p}_2) \iff \mathcal{C}^0_{(k+t)wl}(\mathbf{p}_1) = \mathcal{C}^0_{(k+t)wl}(\mathbf{p}_2)$ for any $\mathbf{p}_1, \mathbf{p}_2 \in H_t(\mathbf{v})$. Therefore $\{\!\{\mathcal{C}^1_{ktfwl}(\mathbf{p}) | \mathbf{p} \in H_t(\mathbf{v})\}\!\}$ is equivalent to $\{\!\{\mathcal{C}^0_{(k+t)wl}(\mathbf{p}) | \mathbf{p} \in H_t(\mathbf{v})\}\!\}$, which is essentially the multiset of isomorphism type of $\mathbf{p} \in H_t(\mathbf{v})$. According to Equation (14), $\mathcal{C}^1_{ktfwl}(\mathbf{v})$ can injectively encode this information. Next, the color histogram of a graph is computed by $\mathcal{C}_{ktfwl}(G) = \text{HASH}(\{\!\{\mathcal{C}^1_{ktfwl}(\mathbf{v}) | \mathbf{v} \in V^k(G)\}\!\})$. It is easy to verify that $\mathcal{C}_{ktfwl}(G)$ can encode the isomorphism type of all $(k+t)$-tuple $\mathbf{p} \in V^{k+t}(G)$. Then, if the graph has size $n \leq k + t$, $\mathcal{C}_{ktfwl}(G)$ can give different color histograms for any non-isomorphic graph pairs. This concludes the proof. $\qquad\square$

Before we prove the relationship between $(k,t)$-FWL and $(k+t)$-WL, we first show some properties of $(k+t)$-WL. Let $\mathcal{C}^\infty_{(k+t)wl}(\mathbf{p})$ be the stable color of tuple $\mathbf{p} \in V^{k+t}(G)$ output from $(k+t)$-WL. We first show that:

**Lemma B.3.** *Suppose $k \geq 2$, $t \geq 1$. Given a graph $G$, for any $\mathbf{v}_1, \mathbf{v}_2 \in V^k(G)$, $\mathbf{w}_1, \mathbf{w}_2 \in V^t(G)$, $\mathbf{p}_1 = (\mathbf{v}_1, \mathbf{w}_1), \mathbf{p}_2 = (\mathbf{v}_2, \mathbf{w}_2)$, $\mathcal{C}^\infty_{(k+t)wl}(\mathbf{p}_1) = \mathcal{C}^\infty_{(k+t)wl}(\mathbf{p}_2)$ implies that $\{\!\{\mathcal{C}^\infty_{(k+t)wl}(\mathbf{q}_1) | \mathbf{q}_1 \in H_t(\mathbf{v}_1)\}\!\}_t = \{\!\{\mathcal{C}^\infty_{(k+t)wl}(\mathbf{q}_2) | \mathbf{q}_2 \in H_t(\mathbf{v}_2)\}\!\}_t$.*

*Proof.* Let $\mathbf{p}_1 = (p_{11}, \ldots, p_{1(k+t)}), \mathbf{p}_2 = (p_{21}, \ldots, p_{2(k+t)})$. As we have $\mathcal{C}^\infty_{(k+t)wl}(\mathbf{p}_1) = \mathcal{C}^\infty_{(k+t)wl}(\mathbf{p}_2)$, given it is the stable color, we must have $\{\!\{\mathcal{C}^\infty_{(k+t)wl}((p_{11}, \ldots, p_{1k}, w, p_{1(k+2)}, \ldots, p_{1(k+t)})) | w \in V(G)\}\!\} = \{\!\{\mathcal{C}^\infty_{(k+t)wl}((p_{21}, \ldots, p_{2k}, w, p_{2(k+2)}, \ldots, p_{2(k+t)})) | w \in V(G)\}\!\}$. Otherwise, by doing one more iteration, we will have $\mathcal{C}^{\infty+1}_{(k+t)wl}(\mathbf{p}_1) \neq \mathcal{C}^{\infty+1}_{(k+t)wl}(\mathbf{p}_2)$, which is a contradiction. Further, we can do another unrolling to have $\{\!\{\{\!\{\mathcal{C}^\infty_{(k+t)wl}((p_{11}, \ldots, p_{1k}, w_1, w_2, \ldots, p_{1(k+t)})) | w_2 \in V(G)\}\!\} | w_1 \in V(G)\}\!\} = \{\!\{\{\!\{\mathcal{C}^\infty_{(k+t)wl}((p_{21}, \ldots, p_{2k}, w_1, w_2, \ldots, p_{2(k+t)})) | w_2 \in V(G)\}\!\} | w_1 \in V(G)\}\!\}$. This implies that $\{\!\{\mathcal{C}^\infty_{(k+t)wl}((p_{11}, \ldots, p_{1k}, w_1, w_2, \ldots, p_{1(k+t)})) | (w_1, w_2) \in V^2(G)\}\!\}_2 = \{\!\{\mathcal{C}^\infty_{(k+t)wl}((p_{21}, \ldots, p_{2k}, w_1, w_2, \ldots, p_{2(k+t)})) | (w_1, w_2) \in V^2(G)\}\!\}_2$ Therefore, by iteratively doing the unrolling, we must have $\{\!\{\mathcal{C}^\infty_{(k+t)wl}((p_{11}, \ldots, p_{1k}, w_1, \ldots, w_t)) | (w_1, \ldots, w_t) \in V^t(G)\}\!\}_t = \{\!\{\mathcal{C}^\infty_{(k+t)wl}((p_{21}, \ldots, p_{2k}, w_1, \ldots, w_t)) | (w_1, \ldots, w_t) \in V^t(G)\}\!\}_t$, which is exactly $\{\!\{\mathcal{C}^\infty_{(k+t)wl}(\mathbf{q}_1) | \mathbf{q}_1 \in H_t(\mathbf{v_1})\}\!\}_t = \{\!\{\mathcal{C}^\infty_{(k+t)wl}(\mathbf{q}_2) | \mathbf{q}_2 \in H_t(\mathbf{v}_2)\}\!\}_t$. This concludes the proof. $\qquad\square$

Lemma B.3 indicates that if the $(k+t)$-WL converges, the hierarchical set will not increase the expressive power of $(k+t)$-WL. Based on this observation, we can slightly rewrite the equation for computing the graph color histogram of $(k+t)$-WL:

$$\mathcal{C}^\infty_{(k+t)wl}(\mathbf{v}) = \text{HASH}(\{\!\{\mathcal{C}^\infty_{(k+t)wl}(\mathbf{p}) | \mathbf{p} \in H_t(\mathbf{v})\}\!\}_t), \tag{15}$$

$$\mathcal{C}^\infty_{(k+t)wl}(G) = \text{HASH}(\{\!\{\mathcal{C}^\infty_{(k+t)wl}(\mathbf{v}) | \mathbf{v} \in V^k(G)\}\!\}), \tag{16}$$

where $\mathcal{C}^\infty_{(k+t)wl}(\mathbf{v})$ is the stable color of $k$-tuple $v$. Here we change the computation of graph color histogram by first hierarchically pooling all tuples in $H_t(\mathbf{v})$ to get a color for $\mathbf{v}$ and then pooling all tuples $\mathbf{v} \in V^k(G)$. It is straightforward that the rewritten will not change the expressive power of $(k+t)$-WL given Lemma B.3. Equation (15) and Equation (16) will be used in the latter proofs. Not surprisingly, we have similar result for $(k,t)$-FWL:

**Lemma B.4.** *Suppose $k \geq 2$, $t \geq 1$. Given a graph $G$, for any $\mathbf{v}_1, \mathbf{v}_2 \in V^k(G)$, $\mathbf{w}_1, \mathbf{w}_2 \in V^t(G)$, $\mathbf{p}_1 = (\mathbf{v}_1, \mathbf{w}_1), \mathbf{p}_2 = (\mathbf{v}_2, \mathbf{w}_2)$, $\mathcal{C}^\infty_{ktfwl}(\mathbf{p}_1) = \mathcal{C}^\infty_{ktfwl}(\mathbf{p}_2)$ implies that $\{\!\{\mathcal{C}^\infty_{ktfwl}(\mathbf{q}_1) | \mathbf{q}_1 \in H_t(\mathbf{v}_1)\}\!\}_t = \{\!\{\mathcal{C}^\infty_{ktfwl}(\mathbf{q}_2) | \mathbf{q}_2 \in H_t(\mathbf{v}_2)\}\!\}_t$.*

*Proof.* Given the definition of the neighborhood tuple, we must have $\mathbf{v}_1 \in Q^F_{\mathbf{w}_1}(\mathbf{v}_1)$ and $\mathbf{v}_2 \in Q^F_{\mathbf{w}_1}(\mathbf{v}_2)$. This means we also have $\mathcal{C}^\infty_{ktfwl}(\mathbf{v}_1) = \mathcal{C}^\infty_{ktfwl}(\mathbf{v}_2)$. Otherwise, by doing one more iteration of $(k,t)$-FWL, we will get different colors for $\mathbf{p}_1$ and $\mathbf{p}_2$, which is a contradiction. $\mathcal{C}^\infty_{ktfwl}(\mathbf{v}_1) = \mathcal{C}^\infty_{ktfwl}(\mathbf{v}_2)$ directly implies that $\{\!\!\{\mathcal{C}^\infty_{ktfwl}(\mathbf{q}_1)|\mathbf{q}_1 \in H_t(\mathbf{v}_1)\}\!\!\}_t = \{\!\!\{\mathcal{C}^\infty_{ktfwl}(\mathbf{q}_2)|\mathbf{q}_2 \in H_t(\mathbf{v}_2)\}\!\!\}_t$ given Equation (14) and thus conclude the proof. $\qquad\square$

Next, we show another property of $(k+t)$-WL. Denote $I_{k+t}$ as a permutation group operates on the indices of a $k+t$-tuple. For $\sigma \in I_{k+t}$, we denote it operates on a $k+t$-tuple $\mathbf{p} = (p_1, \ldots, p_{k+t})$ as $\sigma \cdot \mathbf{p} = (p_{\sigma(1)}, \ldots, p_{\sigma(k+t)})$. For example, for a $\sigma \in I_3$ with $\sigma(1) = 2, \sigma(2) = 1, \sigma(3) = 3$, we have $\sigma \cdot (p_1, p_2, p_3) = (p_2, p_1, p_3)$.

**Lemma B.5.** *Given two graphs $G$ and $H$, for any $\mathbf{p}_1 = (p_{11}, \ldots, p_{1(k+t)}) \in V^{k+t}(G)$, $\mathbf{p}_2 = (p_{21}, \ldots, p_{2(k+t)}) \in V^{k+t}(H)$ and a permutation $\sigma \in I_{k+t}$ operates on the indices of tuples, $\mathcal{C}^l_{(k+t)wl}(\mathbf{p}_1) = \mathcal{C}^l_{(k+t)wl}(\mathbf{p}_2)$ implies that $\mathcal{C}^l_{(k+t)wl}(\sigma \cdot \mathbf{p}_1) = \mathcal{C}^l_{(k+t)wl}(\sigma \cdot \mathbf{p}_2)$.*

*Proof.* We prove it by induction on $l$. At iteration 0, $\mathcal{C}^0_{(k+t)wl}(\mathbf{p}_1) = \mathcal{C}^0_{(k+t)wl}(\mathbf{p}_2)$ means the ordered subgraph induced by $\mathbf{p}_1$ is isomorphic to the ordered subgraph induced by $\mathbf{p}_2$. As the permutation is operating on the indices, it will not change the nodes in the tuple but only change the order of the tuple. As we use the same permutation $\sigma$ for both $\mathbf{p}_1$ and $\mathbf{p}_2$, it is easy to verify $\mathcal{C}^0_{(k+t)wl}(\sigma \cdot \mathbf{p}_1) = \mathcal{C}^0_{(k+t)wl}(\sigma \cdot \mathbf{p}_2)$. Suppose it follows for $1, \ldots, l-1$. At iteration $l$, if $\mathcal{C}^l_{(k+t)wl}(\mathbf{p}_1) = \mathcal{C}^l_{(k+t)wl}(\mathbf{p}_2)$, we have:

$$\{\!\!\{\mathcal{C}^{l-1}_{(k+t)wl}(\mathbf{p}_{1,w/j})|w \in V(G)\}\!\!\} = \{\!\!\{\mathcal{C}^{l-1}_{(k+t)wl}(\mathbf{p}_{2,w/j})|w \in V(H)\}\!\!\}, \quad \forall j \in [k+t].$$

Then, since the statement follows at iteration $l-1$, we must have

$$\{\!\!\{\mathcal{C}^{l-1}_{(k+t)wl}((\sigma \cdot \mathbf{p}_1)_{w/j})|w \in V(G)\}\!\!\} = \{\!\!\{\mathcal{C}^{l-1}_{(k+t)wl}((\sigma \cdot \mathbf{p}_2)_{w/j})|w \in V(H)\}\!\!\}, \quad \forall j \in [k+t].$$

The above equation directly implies that $\mathcal{C}^l_{(k+t)wl}(\sigma \cdot \mathbf{p}_1) = \mathcal{C}^l_{(k+t)wl}(\sigma \cdot \mathbf{p}_2)$. This concludes the proof. $\qquad\square$

With all conclusions at hand, we can start to prove the main proposition in this section.

**Proposition B.6.** *Suppose $k \geq 2$, $t \geq 1$. Given any two graphs $G$ and $H$. If $\mathcal{C}^\infty_{(k+t)wl}(G) = \mathcal{C}^\infty_{(k+t)wl}(H)$, then for any $\mathbf{v}_1 \in V^k(G)$, $\mathbf{v}_2 \in V^k(H)$, $\mathbf{w}_1 \in V^t(G)$, $\mathbf{w}_2 \in V^t(H)$, $\mathbf{p}_1 = (\mathbf{v}_1, \mathbf{w}_1)$, $\mathbf{p}_2 = (\mathbf{v}_2, \mathbf{w}_2)$ with $\mathcal{C}^\infty_{(k+t)wl}(\mathbf{p}_1) = \mathcal{C}^\infty_{(k+t)wl}(\mathbf{p}_2)$, we also have $\mathcal{C}^\infty_{ktfwl}(\mathbf{p}_1) = \mathcal{C}^\infty_{ktfwl}(\mathbf{p}_2)$.*

*Proof.* As indicated in Remark A.4, this can be proved by doing one iteration of $(k,t)$-FWL using the stable color computed by the $k+t$-WL as the initial color. If the color is not refined anymore, we can conclude it. However, since there is no stable color of low dimensional tuples in original $(k+t)$-WL, we resort to Equation (15) for help:

$$\mathcal{C}^\infty_{(k+t)wl}(\mathbf{v}_i) = \mathrm{HASH}(\{\!\!\{\mathcal{C}^\infty_{(k+t)wl}(\mathbf{q})|\mathbf{q} \in H_t(\mathbf{v}_i)\}\!\!\}_t), \quad i = 1, 2. \tag{17}$$

Using Lemma B.3, we must have:

$$\mathcal{C}^\infty_{(k+t)wl}(\mathbf{v}_1) = \mathcal{C}^\infty_{(k+t)wl}(\mathbf{v}_2). \tag{18}$$

Further, given the definition of the neighborhood tuple, any $\mathbf{u}_1 \in Q^F_{\mathbf{w}_1}(\mathbf{v}_1)$ is a sub-tuple of $\mathbf{p}_1$, which means that we can find a permutation $\sigma \in I_{[k+t]}$, such that the first $k$ position of $\sigma \cdot \mathbf{p}_1$ is exactly $\mathbf{u}_1$. As the order of the neighborhood tuple is predefined and the same for both $Q^F_{\mathbf{w}_1}(\mathbf{v}_1)$ and $Q^F_{\mathbf{w}_2}(\mathbf{v}_2)$, the first $k$ position of $\sigma \cdot \mathbf{p}_2$ will be $\mathbf{u}_2$, as long as $\mathbf{u}_1$ and $\mathbf{u}_2$ is at the same position in the neighborhood tuple. Therefore, for any $\mathbf{u}_1 \in Q^F_{\mathbf{w}_1}(\mathbf{v}_1)$ and $\mathbf{u}_2 \in Q^F_{\mathbf{w}_2}(\mathbf{v}_2)$ at the same position $o$ in the neighborhood tuple, we can find a permutation $\sigma_o \in I_{[k+t]}$ to achieve it. Then, given Lemma B.5, we have $\mathcal{C}^\infty_{(k+t)wl}(\sigma_o \cdot \mathbf{p}_1) = \mathcal{C}^\infty_{(k+t)wl}(\sigma_o \cdot \mathbf{p}_2)$, which indicates $\mathcal{C}^\infty_{(k+t)wl}(\mathbf{u}_1) = \mathcal{C}^\infty_{(k+t)wl}(\mathbf{u}_2)$ by using Lemma B.3. This means:

$$\left(\mathcal{C}^\infty_{(k+t)wl}(\mathbf{u}_1)|\mathbf{u}_1 \in Q^F_{\mathbf{w}_1}(\mathbf{v}_1)\right) = \left(\mathcal{C}^\infty_{(k+t)wl}(\mathbf{u}_2)|\mathbf{u}_2 \in Q^F_{\mathbf{w}_2}(\mathbf{v}_2)\right). \tag{19}$$

Now, suppose we use the stable color generated from $(k+t)$-wl as the initial color to perform one more iteration of $(k,t)$-FWL. First, Equation (19) indicates that we have $\mathcal{C}_{ktfwl}^{\infty+1}(\mathbf{p}_1) = \mathcal{C}_{ktfwl}^{\infty+1}(\mathbf{p}_2)$ by doing one iteration of Equation (13). Next, Equation (18) further indicates we have $\mathcal{C}_{ktfwl}^{\infty}(\mathbf{v}_1) = \mathcal{C}_{ktfwl}^{\infty}(\mathbf{v}_2)$ by doing one iteration of Equation (14). This means the color of $\mathbf{p}_1$ and $\mathbf{p}_2$ cannot be further refined by $(k,t)$-FWL. This concludes the proof. □

Now, we can prove Theorem 3.2 in the main paper. We restate it here:

**Theorem B.7.** *For $k \geq 2$, $t \geq 1$, $(k,t)$-FWL is at most as powerful as $(k+t)$-WL. In particular, $(k,1)$-FWL is as powerful as $(k+1)$-WL.*

*Proof.* Based on Proposition B.6, given two graphs $G$, $H$ with $\mathcal{C}_{(k+t)wl}^{\infty}(G) = \mathcal{C}_{(k+t)wl}^{\infty}(H)$. For any $\mathbf{v}_1 \in V^k(G)$, $\mathbf{v}_2 \in V^k(H)$, $\mathbf{w}_1 \in V^t(G)$, $\mathbf{w}_2 \in V^t(H)$, $\mathbf{p}_1 = (\mathbf{v}_1, \mathbf{w}_1)$, $\mathbf{p}_2 = (\mathbf{v}_2, \mathbf{w}_2)$, we have $\mathcal{C}_{ktfwl}^{\infty}(\mathbf{p}_1) = \mathcal{C}_{ktfwl}^{\infty}(\mathbf{p}_2)$ if $\mathcal{C}_{(k+t)wl}^{\infty}(\mathbf{p}_1) = \mathcal{C}_{(k+t)wl}^{\infty}(\mathbf{p}_2)$. This further indicate that $\mathcal{C}_{ktfwl}^{\infty}(\mathbf{v}_1) = \mathcal{C}_{ktfwl}^{\infty}(\mathbf{v}_2)$ if $\mathcal{C}_{(k+t)wl}^{\infty}(\mathbf{v}_1) = \mathcal{C}_{(k+t)wl}^{\infty}(\mathbf{v}_2)$. As we have $\mathcal{C}_{(k+t)wl}^{\infty}(G) = \mathcal{C}_{(k+t)wl}^{\infty}(H)$, for any $\mathbf{v}_1 \in V^k(G)$, we can find a corresponding $\mathbf{v}_2 \in V^k(H)$ such that $\mathcal{C}_{(k+t)wl}^{\infty}(\mathbf{v}_1) = \mathcal{C}_{(k+t)wl}^{\infty}(\mathbf{v}_2)$. This means $\mathcal{C}_{ktfwl}^{\infty}(G) = \text{HASH}(\{\!\{\mathcal{C}_{ktfwl}^{\infty}(\mathbf{v}_1) | \mathbf{v}_1 \in V^k(G)\}\!\}) = \text{HASH}(\{\!\{\mathcal{C}_{ktfwl}^{\infty}(\mathbf{v}_2) | \mathbf{v}_2 \in V^k(H)\}\!\}) = \mathcal{C}_{ktfwl}^{\infty}(H)$. Based on Definition A.3, we complete the proof of the first part. As $(k,1)$-FWL is exactly $k$-FWL, we have $(k,1)$-FWL is as powerful as $(k+1)$-WL. □

Finally, we prove the strict hierarchy of $(k,t)$-FWL. First, we provide the following lemma.

**Lemma B.8.** *For $k \geq 2$, $t \geq 1$, $(k,t)$-FWL $\preceq (k,t+1)$-FWL; $(k,t)$-FWL $\preceq (k+1,t)$-FWL.*

*Proof.* First, we prove $(k,t)$-FWL $\preceq (k,t+1)$-FWL. Actually, we can prove a strong result: given two graphs $G$ and $H$, $\mathbf{v}_1 \in V^k(G)$ and $\mathbf{v}_2 \in V^k(H)$, $\mathcal{C}_{k(t+1)fwl}^{l}(\mathbf{v}_1) = \mathcal{C}_{k(t+1)fwl}^{l}(\mathbf{v}_2)$ also implies $\mathcal{C}_{ktfwl}^{l}(\mathbf{v}_1) = \mathcal{C}_{ktfwl}^{l}(\mathbf{v}_2)$ for any $l \geq 0$. We prove it by induction. For base case, it is easy to verify that if $\mathcal{C}_{k(t+1)fwl}^{0}(\mathbf{v}_1) = \mathcal{C}_{k(t+1)fwl}^{0}(\mathbf{v}_2)$, we have $\mathcal{C}_{ktfwl}^{0}(\mathbf{v}_1) = \mathcal{C}_{ktfwl}^{0}(\mathbf{v}_2)$ as both $(k,t)$-FWL and $(k,t+1)$-FWL use the isomorphism type of $k$-tuples. Suppose it follows for $l = 1, \ldots, l-1$. At iteration $l$, if $\mathcal{C}_{k(t+1)fwl}^{l}(\mathbf{v}_1) = \mathcal{C}_{k(t+1)fwl}^{l}(\mathbf{v}_2)$, according to Equation (14), we have:

$$\mathcal{C}_{k(t+1)fwl}^{l-1}(\mathbf{v}_1) = \mathcal{C}_{k(t+1)fwl}^{l-1}(\mathbf{v}_2), \tag{20}$$

$$\{\!\{\mathcal{C}_{k(t+1)fwl}^{l}(\mathbf{p}_1) | \mathbf{p}_1 \in H_{t+1}(\mathbf{v}_1)\}\!\}_{t+1} = \{\!\{\mathcal{C}_{k(t+1)fwl}^{l}(\mathbf{p}_2) | \mathbf{p}_2 \in H_{t+1}(\mathbf{v}_2)\}\!\}_{t+1}. \tag{21}$$

This means, for any $\mathbf{p}_1 = (\mathbf{v}_1, \mathbf{w}_1) \in H_{t+1}(\mathbf{v}_1)$, we can find a $\mathbf{p}_2 = (\mathbf{v}_2, \mathbf{w}_2) \in H_{t+1}(\mathbf{v}_2)$ such that $\mathcal{C}_{k(t+1)fwl}^{l}(\mathbf{p}_1) = \mathcal{C}_{k(t+1)fwl}^{l}(\mathbf{p}_2)$. According to Equation (13), we further obtain:

$$\left(\mathcal{C}_{k(t+1)fwl}^{l-1}(\mathbf{u}_1) | \mathbf{u}_1 \in Q_{\mathbf{w}_1}^{F}(\mathbf{v}_1)\right) = \left(\mathcal{C}_{k(t+1)fwl}^{l-1}(\mathbf{u}_2) | \mathbf{u}_2 \in Q_{\mathbf{w}_2}^{F}(\mathbf{v}_2)\right). \tag{22}$$

Therefore, we have $\mathcal{C}_{k(t+1)fwl}^{l-1}(\mathbf{u}_1) = \mathcal{C}_{k(t+1)fwl}^{l-1}(\mathbf{u}_2)$ for $\mathbf{u}_1$ and $\mathbf{u}_2$ at the same position. As the statement follows for $l = l-1$, we have $\mathcal{C}_{ktfwl}^{l-1}(\mathbf{u}_1) = \mathcal{C}_{ktfwl}^{l-1}(\mathbf{u}_2)$ for $\mathbf{u}_1$ and $\mathbf{u}_2$ at the same position. Now, suppose we select a $t$-tuple $\mathbf{w}_1'$ from $\mathbf{w}_1$ and let the selection indices be $(i_1, i_2, ..., i_t)$. That is, $\mathbf{w}_1' = (\mathbf{w}_{1i_1}, \ldots, \mathbf{w}_{1i_t})$. Based on the definition A.2, for any $\mathbf{u}_1' \in Q_{\mathbf{w}_1'}^{F}(\mathbf{v}_1)$, it must exists in $Q_{\mathbf{w}_1}^{F}(\mathbf{v}_1)$. Further, let $\mathbf{w}_2' = (\mathbf{w}_{2i_1}, \ldots, \mathbf{w}_{2i_t})$, for any $\mathbf{u}_1' \in Q_{\mathbf{w}_1'}^{F}(\mathbf{v}_1)$, we can find a corresponding $\mathbf{u}_2' \in Q_{\mathbf{w}_2'}^{F}(\mathbf{v}_2)$ at the same position and they must be the same position in $Q_{\mathbf{w}_1}^{F}(\mathbf{v}_1)$ and $Q_{\mathbf{w}_2}^{F}(\mathbf{v}_2)$, respectively. This means, let $\mathbf{p}_1' = (\mathbf{v}_1, \mathbf{w}_1')$ and $\mathbf{p}_2' = (\mathbf{v}_2, \mathbf{w}_2')$, we have $\mathcal{C}_{ktfwl}^{l}(\mathbf{p}_1') = \mathcal{C}_{ktfwl}^{l}(\mathbf{p}_2')$. Let selection indices be $(1, 2, \ldots, t)$, which is the first $t$ elements in the tuple, for any $\mathbf{p}_1' = (\mathbf{v}_1, \mathbf{w}_1') \in H_t(v_1)$, we can find at least one $\mathbf{p}_1 = (\mathbf{v}_1, \mathbf{w}_1) \in H_{t+1}(\mathbf{v}_1)$ that the first $t$ elements of $\mathbf{w}_1$ is $\mathbf{w}_1'$. For the corresponding $\mathbf{p}_2 = (\mathbf{v}_2, \mathbf{w}_2) \in H_{t+1}(\mathbf{v}_2)$ with $\mathcal{C}_{k(t+1)fwl}^{l}(\mathbf{p}_1) = \mathcal{C}_{k(t+1)fwl}^{l}(\mathbf{p}_2)$, we have $\mathcal{C}_{ktfwl}^{l}(\mathbf{p}_1') = \mathcal{C}_{ktfwl}^{l}(\mathbf{p}_2')$, where $\mathbf{p}_2' = (\mathbf{v}_2, \mathbf{w}_2')$ and $\mathbf{w}_2'$ is the first $t$ elements of $\mathbf{w}_2$. Based on Equation (21), we have:

$$\{\!\{\mathcal{C}_{ktfwl}^{l}(\mathbf{p}_1') | \mathbf{p}_1' \in H_t(\mathbf{v}_1)\}\!\}_t = \{\!\{\mathcal{C}_{ktfwl}^{l}(\mathbf{p}_2') | \mathbf{p}_2' \in H_t(\mathbf{v}_2)\}\!\}_t. \tag{23}$$

Finally, given Equation (20), we also have $\mathcal{C}_{ktfwl}^{l-1}(\mathbf{v}_1) = \mathcal{C}_{ktfwl}^{l-1}(\mathbf{v}_2)$, which implies $\mathcal{C}_{ktfwl}^{l}(\mathbf{v}_1) = \mathcal{C}_{ktfwl}^{l}(\mathbf{v}_2)$. This concludes the proof.

For the second part, the procedure is similar, as for any $\mathbf{v} \in V^k(G)$, we can find at least one $\mathbf{v}' \in V^{k+1}(G)$ that contains all nodes and persevere the order of $\mathbf{v}$. Further, given a $\mathbf{w} \in V^t(G)$, for any $\mathbf{u} \in Q_{\mathbf{w}}^F(\mathbf{v})$, we can find at least one $\mathbf{u}' \in Q_{\mathbf{w}}^F(\mathbf{v}')$ that contains all nodes and persevere the order of $\mathbf{u}$. Then, an induction on $l$ can be performed and we omit the details.

$\square$

Next, we show a useful property of $(k, t)$-FWL.

**Lemma B.9.** *For $k \geq 2$, $t \geq 1$, given a graph $G$, for any $\mathbf{v}_1, \mathbf{v}_2 \in V^k(G)$, $\mathcal{C}_{ktfwl}^{\infty}(\mathbf{v}_1) = \mathcal{C}_{ktfwl}^{\infty}(\mathbf{v}_2)$ only if $\forall i, j \in [k]$, $SPD(v_{1i}, v_{1j}) = SPD(v_{2i}, v_{2j})$.*

*Proof.* Based on Lemma B.8, we only need to show it is true for $(k, 1)$-FWL, which is the original $k$-FWL. At the iteration 0, $\mathcal{C}_{k1fwl}^0(\mathbf{v}_1) = \mathcal{C}_{k1fwl}^0(\mathbf{v}_2)$ if and only if the induced subgraph of $\mathbf{v}_1$ and $\mathbf{v}_2$ are isomorphic. Therefore, $\forall i, j \in [k]$, if $SPD(v_{1i}, v_{1j}) = 0$ or $SPD(v_{1i}, v_{1j}) = 1$, we must have $SPD(v_{2i}, v_{2j}) = 0$ or $SPD(v_{2i}, v_{2j}) = 1$, respectively. After the first iteration, $\forall i, j \in [k]$, if there exist at least one $w_1 \in V(G)$ such that $(v_{1i}, w_1), (w_1, v_{1j}) \in E(G)$ but no such $w_2 \in V(G)$ exist with $(v_{2i}, w_2), (w_2, v_{2j}) \in E(G)$, $(k, 1)$-FWL will output different color for $\mathbf{v}_1$ and $\mathbf{v}_2$, given the color update equation of $(k, 1)$-FWL. This means $\mathcal{C}_{k1fwl}^1(\mathbf{v}_1) = \mathcal{C}_{k1fwl}^1(\mathbf{v}_2)$ implies that for $\forall i, j \in [k]$, if $SPD(v_{1i}, v_{1j}) = 2$, we must have $SPD(v_{2i}, v_{2j}) = 2$. Similarly, after the second iteration, $(k, 1)$-FWL will output different color for $\mathbf{v}_1$ and $\mathbf{v}_2$ if there exist at least one $w_1 \in V(G)$ with $SPD(v_{1i}, w_1) = 2$ and $SPD(w_1, v_{1j}) = 1$ but no such $w_2 \in V(G)$ exists for $\mathbf{v}_2$. Therefore, we can conclude that $l$ iteration of $(k, 1)$-FWL can generate different color for $\mathbf{v}_1$ and $\mathbf{v}_2$ if there exist some $i, j \in [k]$ with $SPD(v_{1i}, v_{1j}) = d$ but $SPD(v_{2i}, v_{2j}) \neq d$ ($d \leq l + 1$). Therefore, by performing enough iterations, $(k, 1)$-FWL can compute the shortest path distance between any pair of nodes in a $k$-tuple and it is easy to verify $\mathcal{C}_{ktfwl}^{\infty}(\mathbf{v}_1) = \mathcal{C}_{ktfwl}^{\infty}(\mathbf{v}_2)$ only if $\forall i, j \in [k]$, $SPD(v_{1i}, v_{1j}) = SPD(v_{2i}, v_{2j})$. This concludes the proof. $\square$

Next, we introduce Cai-Furer-Immermman (CFI) graphs [13]. CFI graphs are a family of graphs that $k + 1$-WL can differentiate while $k$-WL cannot. Here we first state the construction of CFI graphs [8].
**Construction.** Let $K_{k+1}$ denote a complete graph on $k + 1$ nodes (without self-loop). Nodes in $K_{k+1}$ are labeled from 0 to $k$. Let $E(v)$ denote the set of edges incident to $v$ in $K_{k+1}$. We have $|E(v)| = k$ for all $v \in V(K_{k+1})$. Then, we define the graph $G_k$ as follows:

1. For the node set $V(G_k)$, we add
    (a) $(v, S)$ for each $v$ in $V(K_{k+1})$ and for each *even* subset $S \subseteq E(v)$;
    (b) two nodes $e^1$ and $e^0$ for each edge $e \in E(K_{k+1})$.
2. For the edge set $E(G_k)$, we add
    (a) an edge $(e^0, e^1)$ for each $e \in E(K_{k+1})$;
    (b) an edge between $(v, S)$ and $e^1$ if $v \in e$ and $e \in S$;
    (c) an edge between $(v, S)$ and $e^0$ if $v \in e$ and $e \notin S$.

Further, we construct a companion graph $H_k$ in a similar way to $G_k$, except that: in Step 1(a), for the vertex $0 \in V(K_{k+1})$, we choose all *odd* subsets of $E(0)$. It is easy to verify that both $G_k$ and $H_k$ have $(k + 1) \cdot 2^{k-1} + \binom{k}{2} \cdot 2$ nodes.

We say a set $S$ of nodes forms a *distance-two-clique* if the distance between any two vertices in $S$ is exactly two.

**Lemma B.10.** *The following holds for the graphs $G_k$ and $H_k$ defined above.*

- *There exists a distance-two-clique of size $(k + 1)$ in $G_k$.*

- *There does not exist a distance-two-clique of size $(k + 1)$ inside $H_k$.*

*Proof.* Please see the Lemma 8 in [8] for detailed proof. □

Next, we show that $(k, t)$-FWL cannot distinguish $G_{k+t}$ and $H_{k+t}$ but $(k+1, t)$-FWL and $(k, t+1)$-FWL can.

**Lemma B.11.** *For $k \geq 2$, $t \geq 1$, $(k, t)$-FWL cannot distinguish $G_{k+t}$ and $H_{k+t}$.*

*Proof.* Given Theorem 3.2, $(k, t)$-FWL is at most as powerful as $(k+t)$-WL. As we know $(k+t)$-WL cannot distinguish $G_{k+t}$ and $H_{k+t}$, we conclude that $(k, t)$-FWL cannot distinguish between them neither. □

**Lemma B.12.** *For $k \geq 2$, $t \geq 1$, $(k+1, t)$-FWL and $(k, t+1)$-FWL can distinguish $G_{k+t}$ and $H_{k+t}$.*

*Proof.* The proof idea is to show both $(k+1, t)$-FWL and $(k, t+1)$-FWL are powerful enough to detect distance-two-cliques of size $(k+1)$, which is sufficient to show that $(k+1, t)$-FWL and $(k, t+1)$-FWL can distinguish between $G_{k+t}$ and $H_{k+t}$. Given Lemma B.9, we know that both $(k+1, t)$-FWL and $(k, t+1)$-FWL can compute the distance between any two nodes in a $k+1$-tuple and $k$-tuple, respectively. We first discuss $(k+1, t)$-FWL. In $G_k$, there exist a $k+t+1$-tuple $\mathbf{p} = (\mathbf{v}, \mathbf{w})$ with $\mathbf{v} \in V^{k+1}(G_k)$ and $\mathbf{w} \in V^t(G_k)$ that exactly form a distance-two-clique. Based on Equation 13, there exists one $\mathbf{p} \in V^{k+t+1}(G_k)$ such that $\mathcal{C}^l_{(k+1)tfwl}(\mathbf{p})$ encodes the distance between any pair of nodes in $\mathbf{p}$ is 2. However, there is no such $\mathbf{p}$ in $H_k$. This is sufficient to show that $(k+1, t)$-FWL can distinguish $G_k$ and $H_k$. The proof for $(k, t+1)$-FWL is similar, as $(k, t+1)$-FWL can also encode pairwise distance information for any $k+t+1$-tuple, thus detecting distance-two-clique with the size of $k+t+1$. This concludes the proof. □

Now, we are ready to prove Proposition 3.3 in the main paper. We restate it here:

**Proposition B.13.** *For $k \geq 2$, $t \geq 1$, $(k, t+1)$-FWL is strictly more powerful than $(k, t)$-FWL; $(k+1, t)$-FWL is strictly more powerful than $(k, t)$-FWL.*

*Proof.* It is clear the proposition holds given Lemma B.8, Lemma B.12, and Definition A.3. □

## B.2 Complexity analysis

For $(k, t)$-FWL, the space complexity is $O(n^k)$ as we only need to store representations of all $k$-tuples. For time complexity, if $k$ and $t$ are relatively small, to update the color of each $k$-tuple, $(k, t)$-FWL needs to aggregate all $V^t(G)$ possible neighborhood tuples, resulting in $O(n^{k+t})$ time complexity. If both $k$ and $t$ are large enough (related to $n$), the time complexity further increases to $O(n^{k+t} \cdot m \cdot (\frac{q!}{m!})^2)$, where $m = min(k, t)$ and $q = max(k, t)$.

## B.3 Limitations

As discussed in the above section, although $(k, t)$-FWL can have a fixed space complexity of $O(n^k)$, the time complexity will grow exponentially if we increase the $t$. Further, the increase of the $t$ will also make the length of the neighborhood tuple increase. This makes the practical usage of $(k, t)$-FWL limited (This will be hugely alleviated by $(k, t)$-FWL+). This is evidence of the "no free lunch" issue in developing an expressive GNN model. However, we do believe there are many repeat computations in the $(k, t)$-FWL that can be safely removed without the hurt of the expressive power. We leave it to our future work.

## C Additional discussion on $(k, t)$-FWL+

### C.1 Detailed proofs for $(k, t)$-FWL+

In this section, we provide all detailed proofs for $(k, t)$-FWL+. Specifically, we will use $(k, t)$-FWL+ to implement many existing powerful GNN/WL models with the best matching expressive power. To simplify notations and make it more clear, let $\hat{\mathcal{C}}$ denote the color output by $(k, t)$-FWL+, $\tilde{\mathcal{C}}$ denote the

color output by any existing model we want to compare. Further, denote $\hat{\mathcal{C}}^l(v_1, v_2)$ as the color of tuple $(v_1, v_2)$ at iteration $l$ if $k = 2$.

**SLFWL(2)** [12]: We prove Proposition 3.5 in the main paper. We restate it here:

**Proposition C.1.** *Let $t = 1$, $k = 2$, and $ES(\mathbf{v}) = \mathcal{N}_1(v_1) \cup \mathcal{N}_1(v_2)$, the corresponding $(k, t)$-FWL+ instance is equivalent to SLFWL(2) [12] and strictly more powerful than any existing node-based subgraph GNNs.*

*Proof.* Given $t = 1$, $k = 2$, and $ES(\mathbf{v}) = \mathcal{N}_1(v_1) \cup \mathcal{N}_1(v_2)$, the color update equation of $(k, t)$-FWL+ can be written as:

$$\hat{\mathcal{C}}^l(v_1, v_2) = \text{HASH}(\hat{\mathcal{C}}^{l-1}(v_1, v_2), \{\!\!\{ \left( \hat{\mathcal{C}}^{l-1}(v_1, w), \hat{\mathcal{C}}^{l-1}(w, v_2) \right) | w \in \mathcal{N}_1(v_1) \cup \mathcal{N}_1(v_2)) \}\!\!\}). \quad (24)$$

We can see Equation (24) exactly matches SLFWL(2) stated in [12] and trivially this instance is as powerful as SLFWL(2). Further, given Theorem 7.1 in [12], we conclude that this instance is strictly more powerful than all existing node-based subgraph GNNs. $\qquad \square$

**$\delta$-k-LWL**: The aggregation scheme of $\delta$-k-LWL can be written as:

$$\tilde{\mathcal{C}}^l(\mathbf{v}) = \text{HASH}(\tilde{\mathcal{C}}^{l-1}(\mathbf{v}), (\{\!\!\{ \mathcal{C}^{l-1}(\mathbf{u}) | \mathbf{u} \in \tilde{Q}_j(\mathbf{v}) \}\!\!\} | j \in [k])), \quad (25)$$

where $Q_j(\mathbf{v}) = \{\mathbf{v}_{w/j} | w \in Q_1(v_j)\}$. Next, we prove Proposition 3.6 in the main paper. We restate it here:

**Proposition C.2.** *Let $t = 1$, $k = k$, and $ES(\mathbf{v}) = \bigcup_{i=1}^k Q_1(v_i)$, the corresponding $(k, t)$-FWL+ instance is more powerful than $\delta$-k-LWL [8].*

*Proof.* Given $t = 1$, $k = k$, and $ES(\mathbf{v}) = \bigcup_{i=1}^k Q_1(v_i)$, the color update equation of $(k, t)$-FWL+ can be written as:

$$\hat{\mathcal{C}}^l(\mathbf{v}) = \text{HASH}(\hat{\mathcal{C}}^{l-1}(\mathbf{v}), \{\!\!\{ (\hat{\mathcal{C}}^{l-1}(\mathbf{u}) | \mathbf{u} \in Q_w^F(\mathbf{v})) | w \in \bigcup_{i=1}^k Q_1(v_i) \}\!\!\}). \quad (26)$$

Given two $k$-tuple $\mathbf{v}_1, \mathbf{v}_2 \in V^k(G)$, it is easy to see that for any node $w \in V(G)$, $\hat{\mathcal{C}}^0(\mathbf{v}_{1,w/j})$ will be different to $\hat{\mathcal{C}}^0(\mathbf{v}_{2,w/j})$ as long as there exist any $v_{1i}$ and $v_{2i}$ such that $v_{1i}$ is a neighbor of node $w$ but $v_{2i}$ is not for any $i \in [k], i \neq j$. This means that for any $\mathbf{u} \in Q_w^F(\mathbf{v})$, $\hat{\mathcal{C}}^0(\mathbf{u})$ can injectively encode whether $w$ is a neighbor to all other nodes in tuples. Further, as $\bigcup_{i=1}^k Q_1(v_i)$ includes neighbors of all nodes in tuple $\mathbf{v}$, we can injectively encode $(\{\!\!\{ \hat{\mathcal{C}}^l(\mathbf{v}_{w/j}) | w \in Q_1(v_j) \}\!\!\} | j \in [k])$ for any $l$ by $\{\!\!\{ (\hat{\mathcal{C}}^{l-1}(\mathbf{u}) | \mathbf{u} \in Q_w^F(\mathbf{v})) | w \in \bigcup_{i=1}^k Q_1(v_i) \}\!\!\}$. Now, given any two graphs $G$, $H$ with $\hat{\mathcal{C}}^\infty(G) = \hat{\mathcal{C}}^\infty(H)$. For any $\mathbf{v}_1 \in V^k(G)$ and $\mathbf{v}_2 \in V^k(H)$ with $\hat{\mathcal{C}}^\infty(\mathbf{v}_1) = \hat{\mathcal{C}}^\infty(\mathbf{v}_2)$, we must have $\{\!\!\{ \mathcal{C}^\infty(\mathbf{v}_{1,w/j}) | w \in Q_1(v_{1j}) \}\!\!\} = \{\!\!\{ \mathcal{C}^\infty(\mathbf{v}_{2,w/j}) | w \in Q_1(v_{2j}) \}\!\!\}$ for any $j \in [k]$ given the above statement. Then, it is easy to verify that we cannot further refine the color of tuple $\mathbf{v}_1$ and $\mathbf{v}_2$ if we use the stable color of $(k, t)$-FWL+ as initial color to perform another iteration of $\delta$-k-LWL, so as the final graph color histogram. Therefore, we conclude that $(k, t)$-FWL+ is more powerful than $\delta$-k-LWL. $\qquad \square$

**GraphSNN** [21]: The GNN aggregation scheme of GraphSNN can be written as:

$$\tilde{\mathcal{C}}^l(v_1, v_1) = \text{HASH}(\tilde{\mathcal{C}}^{l-1}(v_1, v_1), \{\!\!\{ \left( \tilde{\mathcal{C}}^{l-1}(v_1, w), \tilde{\mathcal{C}}^{l-1}(w, w) \right) | w \in Q_1(v_1) \}\!\!\}, \quad (27)$$

$$\tilde{\mathcal{C}}^l(v_1, v_2) = \text{HASH}(\frac{|E_{v_1 v_2}||V_{v_1 v_2}|^\lambda}{|V_{v_1 v_2}||V_{v_1 v_2} - 1|}), \quad (28)$$

where $|E_{v_1 v_2}|$, $|V_{v_1 v_2}|$ are the number of edges and nodes in the overlapping 1-hop subgraph between node $v_1$ and $v_2$, $\lambda$ is a constant. Next, we prove Proposition 3.7 in the main paper. We restate it here:

**Proposition C.3.** *Let $t = 2$, $k = 2$, and $ES^2(\mathbf{v}) = (Q_1(v_1) \cap Q_1(v_2)) \times (Q_1(v_1) \cap Q_1(v_2))$, the corresponding $(k, t)$-FWL+ instance is more powerful than GraphSNN [21]..*

*Proof.* Here we prove a slightly stronger one: even without a hierarchical multiset but only a plain multiset, the resulted $(k, t)$-FWL+ is already powerful enough to match GraphSNN. Given $t = 2$, $k = 2$, and $ES^2(\mathbf{v}) = (Q_1(v_1) \cap Q_1(v_2)) \times (Q_1(v_1) \cap Q_1(v_2))$, the color update equation of $(k, t)$-FWL+ can be written as:

$$\hat{\mathcal{C}}^l(v_1, v_2) = \text{HASH}(\hat{\mathcal{C}}^{l-1}(v_1, v_2), \{\!\!\{ \left( \hat{\mathcal{C}}^{l-1}(u_1, u_2) | (u_1, u_2) \in Q^F_{(w_1, w_2)}(v_1, v_2) \right) \tag{29}$$
$$| (w_1, w_2) \in (Q_1(v_1) \cap Q_1(v_2)) \times (Q_1(v_1) \cap Q_1(v_2)) \}\!\!\}).$$

First, we show that $\hat{\mathcal{C}}^1(v_1, v_2)$ is already powerful to recover all the information in Equation (28). For any $(w_1, w_2) \in (Q_1(v_1) \cap Q_1(v_2)) \times (Q_1(v_1) \cap Q_1(v_2))$, $w_1$ and $w_2$ must all be common neighbors of node $v_1$ and $v_2$. Therefore $(Q_1(v_1) \cap Q_1(v_2)) \times (Q_1(v_1) \cap Q_1(v_2))$ will contain all node pairs in the 1-hop overlapping subgraph of $(v_1, v_2)$. Thus we have the size of $(Q_1(v_1) \cap Q_1(v_2)) \times (Q_1(v_1) \cap Q_1(v_2))$ equal to $|V_{v_1 v_2}|^2$ and this information will be definitely encoded in $\hat{\mathcal{C}}^1(v_1, v_2)$ by aggregating all possible tuple $\left( \hat{\mathcal{C}}^0(u_1, u_2) | (u_1, u_2) \in Q^F_{(w_1, w_2)}(v_1, v_2) \right)$. Further, we have $(w_1, w_2) \in Q^F_{(w_1, w_2)}(v_1, v_2)$, which means $\hat{\mathcal{C}}^1(v_1, v_2)$ will also encode $\{\!\!\{ \hat{\mathcal{C}}^0(w_1, w_2) | (w_1, w_2) \in (Q_1(v_1) \cap Q_1(v_2)) \times (Q_1(v_1) \cap Q_1(v_2)) \}\!\!\}$. As $\hat{\mathcal{C}}^0(w_1, w_2)$ encode whether there is an edge between $w_1$ and $w_2$, it is easy to see $\{\!\!\{ \hat{\mathcal{C}}^0(w_1, w_2) | (w_1, w_2) \in (Q_1(v_1) \cap Q_1(v_2)) \times (Q_1(v_1) \cap Q_1(v_2)) \}\!\!\}$ can encode $|E_{v_1 v_2}|$. Thus we can conclude that $\hat{\mathcal{C}}^1(v_1, v_2)$ can encode Equation (28). Namely, for any two tuples $(v_{11}, v_{12}), (v_{21}, v_{22}) \in V^2(G)$, we have: $\hat{\mathcal{C}}^l(v_{11}, v_{12}) = \hat{\mathcal{C}}^l(v_{21}, v_{22}) \Rightarrow \tilde{\mathcal{C}}^l(v_{11}, v_{12}) = \tilde{\mathcal{C}}^l(v_{21}, v_{22})$, for any $l \geq 1$.

Second, for any tuple $(v_1, v_1) \in V^2(G)$, the neighbor in $(k, t)$-FWL+ is $Q^2_1(v_1)$. For any $(w_1, w_2) \in Q^2_1(v_1)$, it is easy to see that $\hat{\mathcal{C}}^0(w_1, w_2)$ will be different between tuples with $w_1 \neq w_2$ and $w_1 = w_2$. This implies that $(k, t)$-FWL+ can injectively encode $\{\!\!\{ \hat{\mathcal{C}}^l(w, w) | w \in Q_1(v_1) \}\!\!\}$ for any $l$ as $(w_1, w_2) \in Q^F_{(w_1, w_2)}(v_1, v_1)$. Namely, for any two nodes $v_1, v_2 \in V(G)$, we have: $\hat{\mathcal{C}}^l(v_1, v_1) = \hat{\mathcal{C}}^l(v_2, v_2) \Rightarrow \{\!\!\{ \hat{\mathcal{C}}^{l-1}(w_1, w_1) | w_1 \in Q_1(v_1) \}\!\!\} = \{\!\!\{ \hat{\mathcal{C}}^{l-1}(w_2, w_2) | w_2 \in Q_1(v_2) \}\!\!\}$.

Third, since $(v_1, w_1) \in Q^F_{(w_1, w_2)}(v_1, v_1)$, it is easy to verify that $(k, t)$-FWL+ can injectively encode $\{\!\!\{ \hat{\mathcal{C}}^l(v_1, w_1) | w_1 \in Q_1(v_1) \}\!\!\}$ for any $l$. Namely, for any two nodes $v_1, v_2 \in V(G)$, we have: $\hat{\mathcal{C}}^l(v_1, v_1) = \hat{\mathcal{C}}^l(v_2, v_2) \Rightarrow \{\!\!\{ \hat{\mathcal{C}}^{l-1}(v_1, w_1) | w_1 \in Q_1(v_1) \}\!\!\} = \{\!\!\{ \hat{\mathcal{C}}^{l-1}(v_2, w_2) | w_2 \in Q_1(v_2) \}\!\!\}$.

Now, given any two graphs $G$, $H$ with $\hat{\mathcal{C}}^\infty(G) = \hat{\mathcal{C}}^\infty(H)$. For any $(v_{11}, v_{12}) \in V^2(G)$ and $(v_{21}, v_{22}) \in V^2(H)$ with $\hat{\mathcal{C}}^\infty(v_{11}, v_{12}) = \hat{\mathcal{C}}^\infty(v_{21}, v_{22})$, if $v_{11} \neq v_{12}$, we must have $v_{21} \neq v_{22}$ and we have $\tilde{\mathcal{C}}^\infty(v_{11}, v_{12}) = \tilde{\mathcal{C}}^\infty(v_{21}, v_{22})$ if $\hat{\mathcal{C}}^\infty(v_{11}, v_{12}) = \hat{\mathcal{C}}^\infty(v_{21}, v_{22})$ given the first statement. If $v_{11} = v_{12}$ and $v_{21} = v_{22}$, we further denote it by $(v_{11}, v_{11})$ and $(v_{22}, v_{22})$. We have $\{\!\!\{ \hat{\mathcal{C}}^\infty(w_{11}, w_{11}) | w_{11} \in Q_1(v_{11}) \}\!\!\} = \{\!\!\{ \hat{\mathcal{C}}^\infty(w_{21}, w_{21}) | w_{21} \in Q_1(v_{21}) \}\!\!\}$ given the second statement and $\{\!\!\{ \hat{\mathcal{C}}^\infty(v_{11}, w_{11}) | w_{11} \in Q_1(v_{11}) \}\!\!\} = \{\!\!\{ \hat{\mathcal{C}}^\infty(v_{21}, w_{21}) | w_{21} \in Q_1(v_{21}) \}\!\!\}$ given the third statemtn. Now, given the stable color output by $(k, t)$-FWL+, we can verify that performing another iteration using Equation (27) and Equation (28) will not further refine the color of $(v_{11}, v_{11})$ and $(v_{22}, v_{22})$, resulting in the same final graph color histogram. This concludes the proof. $\qquad \square$

**Edge-based subgraph GNNs** The GNN aggregation scheme of edge-based subgraph GNNs like $I^2$-GNN [19] can be written as:

$$\tilde{\mathcal{C}}^l(v_1, v_2, w_2) = \text{HASH}(\tilde{\mathcal{C}}^{l-1}(v_1, v_2, w_2), \{\!\!\{ \tilde{\mathcal{C}}^{l-1}(v_1, w_1, w_2) | w_1 \in Q_1(v_2) \}\!\!\}), \quad \forall w_2 \in Q_1(v_1), \tag{30}$$

$$\tilde{\mathcal{C}}^\infty(G) = \text{HASH}(\{\!\!\{ \{\!\!\{ \{\!\!\{ \tilde{\mathcal{C}}^\infty(v_1, v_2, w_2) | v_2 \in V(G) \}\!\!\} | w_2 \in Q_1(v_1) \}\!\!\} | v_1 \in V(G) \}\!\!\}), \tag{31}$$

where we assume that the initial color of $\tilde{\mathcal{C}}^0(v_1, v_2, w_2)$ is $l_G(v_2)$ if $v_2 \neq [v_1, w_2]$ or $\text{HASH}(l_G(v_2), 1)$ if $v_2 = v_1$ or $\text{HASH}(l_G(v_2), 2)$ if $v_2 = w_2$. Equation (30) can be described as in the subgraph rooted at edge $(v_1, w_2)$, we do the 1-WL color update on all node $v_2$. Meanwhile, the initial color is the node marking of $v_1$ and $w_2$, which is the most powerful subgraph selection policy [19]. First, we show some useful lemma for edge-based subgraph GNNs.

**Lemma C.4.** *Given two graphs $G$, $H$, for any two tuples $(v_{11}, v_{12}) \in V^2(G)$, $(v_{21}, v_{22}) \in V^2(H)$, $w_{12} \in Q_1(v_{11})$, and $w_{22} \in Q_1(v_{21})$, we have $\tilde{\mathcal{C}}^\infty(v_{11}, v_{11}, w_{12}) = \tilde{\mathcal{C}}^\infty(v_{21}, v_{21}, w_{22})$ and $\tilde{\mathcal{C}}^\infty(v_{11}, w_{21}, w_{21}) = \tilde{\mathcal{C}}^\infty(v_{21}, w_{22}, w_{22})$ if $\tilde{\mathcal{C}}^\infty(v_{11}, v_{12}, w_{12}) = \tilde{\mathcal{C}}^\infty(v_{21}, v_{22}, w_{22})$.*

*Proof.* We prove it by contradiction. First, if $v_{12} = v_{11}$ or $v_{12} = w_{12}$ we must have $v_{22} = v_{21}$ or $v_{22} = w_{22}$, respectively, as the initial color of $(v_1, v_1, w_2)$ and $(v_1, w_2, w_2)$ are different from other nodes in the subgraph. Further it is easy to show that $\tilde{\mathcal{C}}^\infty(v_{11}, v_{11}, w_{12}) = \tilde{\mathcal{C}}^\infty(v_{21}, v_{21}, w_{22}) \iff \tilde{\mathcal{C}}^\infty(v_{11}, w_{12}, w_{12}) = \tilde{\mathcal{C}}^\infty(v_{21}, w_{22}, w_{22})$ as they are uniquely labeled. Second, if $v_{12} \neq v_{11}, w_{12}$ and $v_{22} \neq v_{21}, w_{22}$, we can leverage the fact that the information of tuple $(v_1, v_1, w_2)$ and $(v_1, w_2, w_2)$ can be passed to other nodes injectively through message passing. Suppose we have $\tilde{\mathcal{C}}^\infty(v_{11}, v_{12}, w_{12}) = \tilde{\mathcal{C}}^\infty(v_{21}, v_{22}, w_{22})$, if $\tilde{\mathcal{C}}^\infty(v_{11}, v_{11}, w_{12}) \neq \tilde{\mathcal{C}}^\infty(v_{21}, v_{21}, w_{22})$, by doing another $m$ iterations ($m$ is large enough), this discrepancy must be reflected by $\tilde{\mathcal{C}}^\infty(v_{11}, v_{12}, w_{12})$ and $\tilde{\mathcal{C}}^\infty(v_{21}, v_{22}, w_{22})$, respectively. Therefore, we must have $\tilde{\mathcal{C}}^{\infty+m}(v_{11}, v_{12}, w_{12}) \neq \tilde{\mathcal{C}}^{\infty+m}(v_{21}, v_{22}, w_{22})$, which is a contradiction to the stable color. The case of $(v_1, w_2, w_2)$ is similar to the first one. This concludes the proof. $\square$

**Lemma C.5.** *Given two graphs $G$, $H$, for any two tuples $(v_{11}, v_{12}) \in V^2(G)$, $(v_{21}, v_{22}) \in V^2(H)$, $w_{12} \in Q_1(v_{11})$, and $w_{22} \in Q_1(v_{21})$, we have $\{\!\{\tilde{\mathcal{C}}^\infty(v_{11}, w_{11}, w_{12})|w_{11} \in V(G))\}\!\} = \{\!\{\tilde{\mathcal{C}}^\infty(v_{21}, w_{21}, w_{22})|w_{21} \in V(H)\}\!\}$ if $\tilde{\mathcal{C}}^\infty(v_{11}, v_{11}, w_{12}) = \tilde{\mathcal{C}}^\infty(v_{21}, v_{21}, w_{22})$ and $\tilde{\mathcal{C}}^\infty(v_{11}, w_{12}, w_{12}) = \tilde{\mathcal{C}}^\infty(v_{21}, w_{22}, w_{22})$.*

*Proof.* This lemma is an extension of Lemma B.6 in [12]. First, iven Lemma B.6 in [12], it is easy to conclude that we have $\{\!\{\tilde{\mathcal{C}}^\infty(v_{11}, w_{11}, w_{12})|w_{11} \in Q_{k_1}(v_{11}) \cap Q_{k_2}(w_{12})\}\!\} = \{\!\{\tilde{\mathcal{C}}^\infty(v_{21}, w_{21}, w_{22})|w_{21} \in Q_{k_1}(v_{21}) \cap Q_{k_2}(w_{22})\}\!\}$ for any $k_1, k_2 \in \mathbb{N}$. This directly implies $\{\!\{\tilde{\mathcal{C}}^\infty(v_{11}, w_{11}, w_{12})|w_{11} \in V(G))\}\!\} = \{\!\{\tilde{\mathcal{C}}^\infty(v_{21}, w_{21}, w_{22})|w_{21} \in V(H)\}\!\}$. This concludes the proof. $\square$

**Lemma C.6.** *Given two graphs $G$, $H$, for any two tuples $(v_{11}, v_{12}) \in V^2(G)$, $(v_{21}, v_{22}) \in V^2(H)$, $w_{12} \in Q_1(v_{11})$, and $w_{22} \in Q_1(v_{21})$, we have $\{\!\{\tilde{\mathcal{C}}^\infty(v_{11}, w_{11}, w_{12})|w_{11} \in V(G))\}\!\} = \{\!\{\tilde{\mathcal{C}}^\infty(v_{21}, w_{21}, w_{22})|w_{21} \in V(H)\}\!\}$ if $\tilde{\mathcal{C}}^\infty(v_{11}, v_{12}, w_{12}) = \tilde{\mathcal{C}}^\infty(v_{21}, v_{22}, w_{22})$.*

*Proof.* Given Lemma C.4, if $\tilde{\mathcal{C}}^\infty(v_{11}, v_{12}, w_{12}) = \tilde{\mathcal{C}}^\infty(v_{21}, v_{22}, w_{22})$, we have $\tilde{\mathcal{C}}^\infty(v_{11}, v_{11}, w_{12}) = \tilde{\mathcal{C}}^\infty(v_{21}, v_{21}, w_{22})$ and $\tilde{\mathcal{C}}^\infty(v_{11}, w_{21}, w_{21}) = \tilde{\mathcal{C}}^\infty(v_{21}, w_{22}, w_{22})$. Next, given Lemma C.5, we have $\{\!\{\tilde{\mathcal{C}}^\infty(v_{11}, w_{11}, w_{12})|w_{11} \in V(G))\}\!\} = \{\!\{\tilde{\mathcal{C}}^\infty(v_{21}, w_{21}, w_{22})|w_{21} \in V(H)\}\!\}$. This concludes the proof. $\square$

**Lemma C.7.** *For any two graphs $G$, $H$ and two nodes $v_{11} \in V(G)$ and $v_{21} \in V(H)$, we have $\{\!\{\{\!\{\tilde{\mathcal{C}}^\infty(v_{11}, w_{11}, w_{12})|w_{11} \in V(G)\}\!\}|w_{12} \in Q_1(v_{11})\}\!\} = \{\!\{\{\!\{\tilde{\mathcal{C}}^\infty(v_{21}, w_{21}, w_{22})|w_{21} \in V(H)\}\!\}|w_{22} \in Q_1(v_{21})\}\!\}$ if $\{\!\{\{\!\{\tilde{\mathcal{C}}^\infty(v_{11}, w_{11}, w_{12})|w_{12} \in Q_1(v_{11})\}\!\}|w_{11} \in V(G)\}\!\} = \{\!\{\{\!\{\tilde{\mathcal{C}}^\infty(v_{21}, w_{21}, w_{22})|w_{22} \in Q_1(v_{21})\}\!\}|w_{21} \in V(H)\}\!\}$.*

*Proof.* If we have $\{\!\{\{\!\{\tilde{\mathcal{C}}^\infty(v_{11}, w_{11}, w_{12})|w_{12} \in Q_1(v_{11})\}\!\}|w_{11} \in V(G)\}\!\} = \{\!\{\{\!\{\tilde{\mathcal{C}}^\infty(v_{21}, w_{21}, w_{22})|w_{22} \in Q_1(v_{21})\}\!\}|w_{21} \in V(H)\}\!\}$, for any $w_{11} \in V(G)$, we can find a $w_{21} \in V(H)$ such that $\{\!\{\tilde{\mathcal{C}}^\infty(v_{11}, w_{11}, w_{12})|w_{12} \in Q_1(v_{11})\}\!\} = \{\!\{\tilde{\mathcal{C}}^\infty(v_{21}, w_{21}, w_{22})|w_{22} \in Q_1(v_{21})\}\!\}$. Next, as the color is stable, based on Lemma C.6, we can conclude that $\{\!\{\{\!\{\tilde{\mathcal{C}}^\infty(v_{11}, w_{11}, w_{12})|w_{11} \in V(G)\}\!\}|w_{12} \in Q_1(v_{11})\}\!\} = \{\!\{\{\!\{\tilde{\mathcal{C}}^\infty(v_{21}, w_{21}, w_{22})|w_{21} \in V(H)\}\!\}|w_{22} \in Q_1(v_{21})\}\!\}$. $\square$

Now, given $t = 2$, $k = 2$, and $ES^2(\mathbf{v}) = Q_1(v_2) \times Q_1(v_1)$, the corresponding $(k, t)$-FWL+ instance can be written as:

$$\hat{\mathcal{C}}^l(v_1, v_2) = \text{HASH}\Big(\hat{\mathcal{C}}^{l-1}(v_1, v_2), \{\!\{\{\!\{\Big(\hat{\mathcal{C}}^{l-1}(u_1, u_2)|(u_1, u_2) \in Q^F_{(w_1, w_2)}(v_1, v_2)\Big)$$
$$|w_1 \in Q_1(v_2)\}\!\}w_2 \in Q_1(v_1)\}\!\}\Big). \tag{32}$$

However, edge-based subgraph GNNs use 3-tuple representations instead of 2-tuples in the $(k, t)$-FWL+, which makes the comparison not trivial. Therefore, we slightly rewrite the above equation:

$$\hat{\mathcal{C}}^l(v_1, v_2, w_1, w_2) = \text{HASH}(\hat{\mathcal{C}}^{l-1}(v_1, v_2, w_1, w_2), \left(\hat{\mathcal{C}}^{l-1}(u_1, u_2)|(u_1, u_2) \in Q^F_{(w_1, w_2)}(v_1, v_2)\right)), \tag{33}$$

$$\hat{\mathcal{C}}^l(v_1, v_2, w_2) = \text{HASH}(\hat{\mathcal{C}}^{l-1}(v_1, v_2, w_2), \{\!\!\{\hat{\mathcal{C}}^l(v_1, v_2, w_1, w_2)|w_1 \in Q_1(v_2)\}\!\!\}), \tag{34}$$

$$\hat{\mathcal{C}}^l(v_1, v_2) = \text{HASH}(\hat{\mathcal{C}}^{l-1}(v_1, v_2), \{\!\!\{\hat{\mathcal{C}}^l(v_1, v_2, w_2)|w_2 \in Q_1(v_1)\}\!\!\}), \tag{35}$$

where we let $\hat{\mathcal{C}}^0(v_1, v_2, w_2)$ and $\hat{\mathcal{C}}^0(v_1, v_2, w_1, w_2)$ be the same for any $(v_1, v_2) \in V^2(G)$, $w_1 \in Q_1(v_2)$, and $w_2 \in Q_1(v_1)$. It is easy to verify that the above equations the equivalent to the original one. Now, we prove a strong lemma:

**Lemma C.8.** *For any two graphs G, H, given two tuples $(v_{11}, v_{12}) \in V^2(G)$ and $(v_{21}, v_{22}) \in V^2(H)$, for any $(w_{11}, w_{12}) \in Q_1(v_{12}) \times Q_1(v_{11})$ and $(w_{21}, w_{22}) \in Q_1(v_{22}) \times Q_1(v_{21})$, we have $\tilde{\mathcal{C}}^{l-1}(v_{11}, w_{11}, w_{12}) = \tilde{\mathcal{C}}^{l-1}(v_{21}, w_{21}, w_{22})$ if $\hat{\mathcal{C}}^l(v_{11}, v_{12}, w_{11}, w_{12}) = \hat{\mathcal{C}}^l(v_{21}, v_{22}, w_{21}, w_{22})$ for any $l \geq 1$.*

*Proof.* We prove it by induction on $l$. The base case is easy to verify given Lemma B.1 and the definition of initial color in edge-based subgraph GNNs.

Now suppose it is true for $l = 2, \dots, l - 1$. At iteration $l$, if $\hat{\mathcal{C}}^l(v_{11}, v_{12}, w_{11}, w_{12}) = \hat{\mathcal{C}}^l(v_{21}, v_{22}, w_{21}, w_{22})$, given Equation 33, we must have:

$$\hat{\mathcal{C}}^{l-1}(v_{11}, v_{12}, w_{11}, w_{12}) = \hat{\mathcal{C}}^{l-1}(v_{21}, v_{22}, w_{21}, w_{22}), \tag{36}$$

$$\hat{\mathcal{C}}^{l-1}(u_{11}, u_{12}) = \hat{\mathcal{C}}^{l-1}(u_{21}, u_{22}), \tag{37}$$

for any $(u_{11}, u_{12}) \in Q^F_{(w_{11}, w_{12})}(v_{11}, v_{12})$ and $(u_{21}, u_{22}) \in Q^F_{(w_{21}, w_{22})}(v_{21}, v_{22})$ at the same position. Particularly, we have $(v_{11}, w_{11}) \in Q^F_{(w_{11}, w_{12})}(v_{11}, v_{12})$ and $(v_{21}, w_{21}) \in Q^F_{(w_{21}, w_{22})}(v_{21}, v_{22})$ and they are in the same position. Given Equation (35), we have:

$$\{\!\!\{\hat{\mathcal{C}}^{l-1}(v_{11}, w_{11}, y_{12})|y_{12} \in Q_1(v_{11})\}\!\!\} = \{\!\!\{\hat{\mathcal{C}}^{l-1}(v_{21}, w_{21}, y_{22})|y_{22} \in Q_1(v_{21})\}\!\!\}. \tag{38}$$

Equation (38) further indicates that for any $y_{12} \in Q_1(v_{11})$, we can find a $y_{22} \in Q_1(v_{21})$ such that

$$\hat{\mathcal{C}}^{l-1}(v_{11}, w_{11}, y_{12}) = \hat{\mathcal{C}}^{l-1}(v_{21}, w_{21}, y_{22}). \tag{39}$$

Next, given Equation (34), we can conclude that:

$$\{\!\!\{\hat{\mathcal{C}}^{l-1}(v_{11}, w_{11}, x_{11}, y_{12})|x_{11} \in Q_1(w_{11})\}\!\!\} = \{\!\!\{\hat{\mathcal{C}}^{l-1}(v_{21}, w_{21}, x_{21}, y_{12})|x_{21} \in Q_1(w_{21})\}\!\!\}. \tag{40}$$

Since the statement follows at iteration $l - 1$, we have $\{\!\!\{\tilde{\mathcal{C}}^{l-2}(v_{11}, x_{11}, y_{12})|x_{11} \in Q_1(w_{11})\}\!\!\} = \{\!\!\{\tilde{\mathcal{C}}^{l-2}(v_{21}, x_{21}, y_{12})|x_{21} \in Q_1(w_{21})\}\!\!\}$. We also have $\tilde{\mathcal{C}}^{l-2}(v_{11}, w_{11}, y_{12}) = \tilde{\mathcal{C}}^{l-2}(v_{21}, w_{21}, y_{22})$ given Equation (39). This means that we must have $\tilde{\mathcal{C}}^{l-1}(v_{11}, w_{11}, y_{12}) = \tilde{\mathcal{C}}^{l-1}(v_{21}, w_{21}, y_{22})$ based on Equation (30). Finally, as $w_{12}$ is one of $y_{12}$ and $w_{22}$ is one of $y_{22}$ and it is one such pair that holds the statement at iteration $l - 1$. It must be the case that they are also the pair at iteration $l$. This means we have $\tilde{\mathcal{C}}^{l-1}(v_{11}, w_{11}, w_{12}) = \tilde{\mathcal{C}}^{l-1}(v_{21}, w_{21}, w_{22})$, which concludes the proof. $\square$

Now we prove Proposition 3.8 in the main paper. We restate it here:

**Proposition C.9.** *Let $t = 2$, $k = 2$, and $ES^2(\mathbf{v}) = Q_1(v_2) \times Q_1(v_1)$, the corresponding $(k, t)$-FWL+ instance is more powerful than edge-based subgraph GNNs like $I^2$-GNN [19].*

*Proof.* Note that the initial color of edge-based subgraph GNNs is weaker than $(k, t)$-FWL+ given Lemma C.8, but it is still sufficient to show that the stable color from $(k, t)$-FWL+ cannot be further refined by edge-based subgraph GNNs. For any two graphs $G$ and $H$ with $\hat{\mathcal{C}}^\infty(G) = \hat{\mathcal{C}}^\infty(H)$, it is easy to see that for any $(v_{11}, v_{12}) \in V^2(G)$ and $w_{12} \in Q_1(v_{11})$, we can find a $(v_{21}, v_{22}) \in V^2(H)$ and $w_{22} \in Q_1(v_{21})$ with $\hat{\mathcal{C}}^\infty(v_{11}, v_{12}, w_{12}) = \hat{\mathcal{C}}^\infty(v_{21}, v_{22}, w_{22})$. Then, given Equation (34), we have $\{\!\!\{\hat{\mathcal{C}}^\infty(v_{11}, v_{12}, w_{11}, w_{12})|w_{11} \in Q_1(v_{12})\}\!\!\} = \{\!\!\{\hat{\mathcal{C}}^\infty(v_{21}, v_{22}, w_{21}, w_{22})|w_{21} \in Q_1(v_{22})\}\!\!\}$. Further, given Lemma C.8, we have $\{\!\!\{\tilde{\mathcal{C}}^\infty(v_{11}, w_{11}, w_{12})|w_{11} \in Q_1(v_{12})\}\!\!\} = \{\!\!\{\tilde{\mathcal{C}}^\infty(v_{21}, w_{21}, w_{22})|w_{21} \in$

$Q_1(v_{22})\}\!\}$ Now, if we use the stable color from $(k,t)$-FWL+ as the initial color to do one more iteration of edge-based subgraph GNNs, it is easy to see that the color of $(v_{11}, w_{11}, w_{12})$ and $(v_{21}, w_{21}, w_{22})$ cannot get further refined and we have $\tilde{\mathcal{C}}^\infty(v_{11}, v_{12}, w_{12}) = \tilde{\mathcal{C}}^\infty(v_{21}, v_{22}, w_{22})$. Note that the graph color histogram of edge-based subgraph GNNs is slightly different than $(k,t)$-FWL+ as in edge-based subgraph GNNs, the order of multiset is $v_2 \to w_2 \to v_1$. In $(k,t)$-FWL+, the order of multiset is $w_2 \to v_2 \to v_1$. However, based on Lemma C.7, the first order is less powerful than the second order. Therefore we can conclude that the final graph color histogram of edge-based subgraph GNNs will also be the same. This concludes the proof. $\qquad\square$

**KP-GNN** [22]: The GNN aggregation scheme of KP-GNN with the shortest path distance kernel and peripheral subgraph encoder as 1-WL can be written as:

$$\tilde{\mathcal{C}}^l(v_1, v_1) = \text{HASH}(\tilde{\mathcal{C}}^{l-1}(v_1, v_1), \{\!\{(\tilde{\mathcal{C}}^{l-1}(w, w), \tilde{\mathcal{C}}^{l-1}(v_1, w)) | w \in V(G)\}\!\}), \qquad (41)$$

$$\tilde{\mathcal{C}}^l(v_1, v_2) = \text{HASH}(\tilde{\mathcal{C}}^{l-1}(v_1, v_2), \{\!\{\tilde{\mathcal{C}}^{l-1}(v_1, w) | w \in Q_{\text{SPD}(v_1, v_2)}(v_1) \cap Q_1(v_2)\}\!\}), \qquad (42)$$

where the initial color of $\tilde{\mathcal{C}}^0(v_1, v_2)$ is the isomorphism type of $(v_1, v_2)$ and $\text{SPD}(v_1, v_2)$. Before we prove Proposition 3.9, we first show that a slightly stronger version of $(k,t)$-FWL+ is still bounded by KP-GNN.

**Lemma C.10.** *Let $t = 1$, $k = 2$, and $ES^t(\mathbf{v}) = Q_{\text{SPD}(v_1, v_2)}(v_1) \cap Q_1(v_2)$. Further let the initial color $\hat{\mathcal{C}}^0(v_1, v_2)$ encode $\text{SPD}(v_1, v_2)$, the corresponding $(k,t)$-FWL+ instance is at most as powerful as KP-GNN [22] with the peripheral subgraph encoder as powerful as 1-WL.*

*Proof.* Given $t = 1$, $k = 2$, and $ES^t(\mathbf{v}) = Q_{\text{SPD}(v_1, v_2)}(v_1) \cap Q_1(v_2)$, the color update equation of $(k,t)$-FWL+ can be written as:

$$\hat{\mathcal{C}}^l(v_1, v_2) = \text{HASH}(\hat{\mathcal{C}}^{l-1}(v_1, v_2), \{\!\{\hat{\mathcal{C}}^{l-1}(v_1, w) | w \in Q_{\text{SPD}(v_1, v_2)}(v_1) \cap Q_1(v_2)\}\!\}). \qquad (43)$$

We can see Equation (43) exactly match Equation (42). However, the update of $\hat{\mathcal{C}}^l(v_1, v_1)$ is not equal to Equation (41) as the neighbor of $(v_1, v_1)$ in $(k,t)$-FWL+ only contain itself. Therefore it is easy to see that for any $l$ given two nodes $v_1, v_2 \in V(G)$, must have $\tilde{\mathcal{C}}^l(v_1, v_1) = \tilde{\mathcal{C}}^l(v_2, v_2)$ if $\hat{\mathcal{C}}^l(v_1, v_1) = \hat{\mathcal{C}}^l(v_2, v_2)$. Given two graphs $G$ and $H$ such that $\tilde{\mathcal{C}}^\infty(G) = \tilde{\mathcal{C}}^\infty(H)$, for any $(v_{11}, v_{12}) \in V^2(G)$ and $(v_{21}, v_{22}) \in V^2(H)$ with $\tilde{\mathcal{C}}^\infty(v_{11}, v_{12}) = \tilde{\mathcal{C}}^\infty(v_{21}, v_{22})$, we must have $\{\!\{\tilde{\mathcal{C}}^\infty(v_{11}, w) | w \in Q_{\text{SPD}(v_{11}, v_{12})}(v_{11}) \cap Q_1(v_{12})\}\!\} = \{\!\{\tilde{\mathcal{C}}^\infty(v_{21}, w) | w \in Q_{\text{SPD}(v_{21}, v_{22})}(v_{21}) \cap Q_1(v_{22})\}\!\}$. Thus, it is easy to verify that if we use the stable color output by KP-GNN as initial color to perform 1 iteration of $(k,t)$-FWL+, we have $\hat{\mathcal{C}}^\infty(v_{11}, v_{12}) = \hat{\mathcal{C}}^\infty(v_{21}, v_{22})$. Further, given Equation (41), for any $(v_{11}, v_{11}) \in V^2(G)$, we can find a $(v_{22}, v_{22}) \in V^2(H)$ such that $\tilde{\mathcal{C}}^\infty(v_{11}, v_{11}) = \tilde{\mathcal{C}}^\infty(v_{22}, v_{22})$, which means $\{\!\{\tilde{\mathcal{C}}^\infty(v_{11}, w_{12}) | w_{12} \in V(G)\}\!\} = \{\!\{\tilde{\mathcal{C}}^\infty(v_{22}, w_{22}), | w_{22} \in V(H)\}\!\}$, and thus $\{\!\{\tilde{\mathcal{C}}^\infty(w_{11}, w_{12}) | (w_{11}, w_{12}) \in V^2(G)\}\!\} = \{\!\{\tilde{\mathcal{C}}^\infty(w_{21}, w_{22}) | (w_{21}, w_{22}) \in V^2(H)\}\!\}$. This indicates that even if $(k,t)$-FWL+ use the shortest path distance as the initial color, we still have $\{\!\{\hat{\mathcal{C}}^\infty(w_{11}, w_{12}) | (w_{11}, w_{12}) \in V^2(G)\}\!\} = \{\!\{\hat{\mathcal{C}}^\infty(w_{21}, w_{22}) | (w_{21}, w_{22}) \in V^2(H)\}\!\}$ and thus the final graph color histogram cannot be refined. This concludes the proof. $\qquad\square$

Now, it is easy to prove Proposition 3.9 in the main paper. We restate it here:

**Proposition C.11.** *Let $t = 1$, $k = 2$, and $ES(\mathbf{v}) = Q_{\text{SPD}(v_1, v_2)}(v_1) \cap Q_1(v_2)$, the corresponding $(k,t)$-FWL+ instance is at most as powerful as KP-GNN [22] with the peripheral subgraph encoder as powerful as 1-WL.*

*Proof.* It is easy to see that the corresponding $(k,t)$-FWL+ is weaker than Equation (43) with the initial color as the shortest path distance, we omit the detailed proof. Then, given Lemma C.10, we can directly conclude the proof. $\qquad\square$

We conjecture that if we further equip $(k,t)$-FWL+ with Equation (41), $(k,t)$-FWL+ can be as powerful as KP-GNN. We also conjecture that if we let $t = p$ and $k = 2$, $ES(\mathbf{v}) = (Q_{\text{SPD}}(v_1, v_2)(v_1) \cap Q_1(v_2))^p$ with initial color as the shortest path distance, the resulting $(k,t)$-FWL+ has a close relationship with KP-GNN with the peripheral subgraph encoder as powerful as LFWL$(p)$ [12] and leave the detailed proof in the future.

**GDGNN** [23]. Before we give the formal update equation of GDGNN, we first formally define $\mathcal{SP}(v_1, v_2)$ as a set of nodes that are in the shortest distance path between node $v_1$ and $v_2$. Namely, $\mathcal{SP}(v_1, v_2) = \{w | \text{SPD}(v_1, w) + \text{SPD}(w, v_2) = \text{SPD}(v_1, v_2), \forall w \in V(G)\}$ if $v_1 \neq v_2$ and $\mathcal{SP}(v_1, v_1) = \{v_1\}$. Then, the color update equation of GDGNN can be written as:

$$\tilde{\mathcal{C}}^l(v_1, v_2) = \text{HASH}(\tilde{\mathcal{C}}^{l-1}(v_1, v_2), \{\!\{\tilde{\mathcal{C}}^{l-1}(v_1, w) | w \in Q_1(v_2)\}\!\}, \{\!\{\tilde{\mathcal{C}}^{l-1}(v_1, w) | w \in \mathcal{SP}(v_1, v_2)\}\!\}). \tag{44}$$

Note that Equation (44) is slightly different from the original version of GDGNN [23] stated in their paper. In the original paper, the author use MPNNs in $O(n)$ space. Meanwhile, they only consider the direct neighbor of node $v_1$ and $v_2$ in $\mathcal{SP}(v_1, v_2)$, and they only add it in the last layer. However, it is easy to verify that Equation (44) is at least as powerful as the original version and we show that the $(k, t)$-FWL+ can be more powerful than this version. Now we prove Proposition 3.10. We restate it here:

**Proposition C.12.** *Let* $t = 2$, $k = 2$, *and* $ES^2(\mathbf{v}) = Q_1(v_2) \times \mathcal{SP}(v_1, v_2)$, *the corresponding* $(k, t)$-*FWL+ is more powerful than GDGNN [23].*

*Proof.* Given $t = 2$, $k = 2$, and $ES^2(\mathbf{v}) = Q_1(v_2) \times \mathcal{SP}(v_1, v_2)$, the color update equation of $(k, t)$-FWL+ can be written as:

$$\hat{\mathcal{C}}^l(v_1, v_2) = \text{HASH}(\hat{\mathcal{C}}^{l-1}(v_1, v_2), \{\!\{\{\!\{(\hat{\mathcal{C}}^{l-1}(u_1, u_2) | (u_1, u_2) \in Q^F_{(w_1, w_2)}(v_1, v_2))}$$
$$| w_1 \in Q_1(v_2)\}\!\} | w_2 \in \mathcal{SP}(v_1, v_2)\}\!\}). \tag{45}$$

First, as $(v_1, w_1) \in Q^F_{(w_1, w_2)}(v_1, v_2)$, we can directly conclude that $(k, t)$-FWL+ can injectively encode $\{\!\{\hat{\mathcal{C}}^l(v_1, w_1) | w_1 \in Q_1(v_2)\}\!\}$ for any $l$.

Second, we also have $(v_1, w_2) \in Q^F_{(w_1, w_2)}(v_1, v_2)$. Further, as in the inner multiset, each element has the same $w_2$. This means that the result of the inner multiset will not affect the information in the outer multiset and it is easy to verify $(k, t)$-FWL+ can injectively encode $\{\!\{\hat{\mathcal{C}}^l(v_1, w_2) | w_2 \in \mathcal{SP}(v_1, v_2)\}\!\}$.

Now, given any two graphs $G$, $H$ with $\hat{\mathcal{C}}^\infty(G) = \hat{\mathcal{C}}^\infty(H)$. For any $(v_{11}, v_{12}) \in V^2(G)$ and $(v_{21}, v_{22}) \in V^2(H)$ with $\hat{\mathcal{C}}^\infty(v_{11}, v_{12}) = \hat{\mathcal{C}}^\infty(v_{21}, v_{22})$ we must have $\{\!\{\hat{\mathcal{C}}^\infty(v_{11}, w_{11}) | w_{11} \in Q_1(v_{12})\}\!\} = \{\!\{\hat{\mathcal{C}}^\infty(v_{21}, w_{21}) | w_{21} \in Q_1(v_{22})\}\!\}$ and $\{\!\{\hat{\mathcal{C}}^\infty(v_{11}, v_{12}) | w_{12} \in \mathcal{SP}(v_{11}, v_{12})\}\!\} = \{\!\{\hat{\mathcal{C}}^\infty(v_{21}, w_{22}) | w_{22} \in \mathcal{SP}(v_{21}, v_{22})\}\!\}$ based on two statements. Then, it is easy to verify that if we use the stable color from $(k, t)$-FWL+ as the initial color to do one more iteration of GDGNN, the color of $(v_{11}, v_{12})$ and $(v_{21}, v_{22})$ cannot be further refined so as the final graph color histogram. This concludes the proof.

$\square$

## C.2 Discussion on the complexity

The space complexity of $(k, t)$-FWL+ is still $O(n^k)$. However, the time complexity of $(k, t)$-FWL+ is dependent on the choice of $ES(\mathbf{v})$ and it is hard to give a formal analysis.

## C.3 Limitations

In the paper, we implement several different instances of $(k, t)$-FWL+ which expressive power is closely related to existing expressive GNNs. However, the whole space of $(k, t)$-FWL+ is less explored. Especially, how many different $ES(\mathbf{v})$ exists? Moreover, how do theoretically and quantitatively analyze the expressive power of $(k, t)$-FWL+? Finally, although $(k, t)$-FWL+ can be arbitrarily powerful even within $O(n^2)$ space from the theoretical side, we observe it is sometimes hard to optimize the model to achieve its theoretical power by using fewer embedding. Specifically, how to ensure the injectiveness of the multiset function? For example, if we use large $t$ and small $k$ for $(k, t)$-FWL+ to conduct a graph isomorphism test, for each $k$-tuple, the hierarchical multiset $\{\!\{\}\!\}_t$ contains exponentially increased elements. So far, we only use the summation as the function to encode this hierarchical multiset. Although it is an injective function from a theoretical view, we find it works badly with information loss when the number of elements is too large. It is worth investigating how to design model architecture that can unleash the full power of $(k, t)$-FWL+. We leave all these to our future work.

# D Additional discussion on N²-FWL and N²-GNN

## D.1 Detailed proofs for N²-FWL

In this section, we provide all detailed proofs for N²-FWL. To make it clear, we denote $\hat{\mathcal{C}}^l(v_1, v_2)$ as the color of tuple $(v_1, v_2)$ at iteration $l$ for N²-FWL. We rewrite Equation 9 as:

$$\hat{\mathcal{C}}^l(v_1, v_2) = \text{HASH}(\hat{\mathcal{C}}^{l-1}(v_1, v_2), \{\!\!\{ \left( \hat{\mathcal{C}}^{l-1}(u_1, u_2) | (u_1, u_2) \in Q^F_{(w_1, w_2)}(v_1, v_2) \right)$$
$$(w_1, w_2) \in (\mathcal{N}_1(v_2) \times \mathcal{N}_1(v_1)) \cap (\mathcal{N}_h(v_1) \cap \mathcal{N}_h(v_2))^2 \}\!\!\}). \tag{46}$$

We start with the first lemma:

**Lemma D.1.** *Given $h$ large enough, N²-FWL is more powerful than SLFWL(2) [12].*

*Proof.* Given $h$ large enough, it is clearly that the neighbor of N²-FWL becomes $\mathcal{N}_1(v_2) \times \mathcal{N}_1(v_1)$. For any $(w_1, w_2) \in \mathcal{N}_1(v_2) \times \mathcal{N}_1(v_1)$, it is easy to verify that $Q^F_{(w_1, w_2)}(v_1, v_2)$ contains all tuples in the aggregation of SLFWL(2). Further, the elements in $\mathcal{N}_1(v_2) \times \mathcal{N}_1(v_1)$ includes all elements in $Q_1(v_2) \cup Q_1(v_1)$. Therefore, it is easy to prove that N²-FWL is more powerful than SLFWL(2) by doing induction on $l$. Here we omit the detailed proof. □

**Lemma D.2.** *Given $h$ large enough, N²-FWL is more powerful than edge-based subgraph GNNs.*

*Proof.* Given $h$ large enough, it is clear that the neighbor of N²-FWL becomes $\mathcal{N}_1(v_2) \times \mathcal{N}_1(v_1)$. The N²-FWL exactly matches the instance in Proposition 3.8 with additional root nodes. As the initial feature of root tuples($(v_1, v_1), (v_2, v_2)$) are different from other tuples, its information will not affect other neighbors. This concludes the proof. □

It is intuitive to conjecture that N²-FWL with the number of $h$ is more powerful than edge-based subgraph GNNs with $h$-hop subgraph and we leave the detailed proof in the future. Now, we can prove Theorem 4.1 in the main paper. We restate it here:

**Theorem D.3.** *Given $h$ is large enough, N²-FWL is more powerful than SLFWL(2) [12] and edge-based subgraph GNNs.*

*Proof.* Lemma D.1 and Lemma D.2 directly implies the theorem. This concludes the proof. □

Next, we prove Theorem 4.3 in the main paper. We restate it here:

**Theorem D.4.** *N²-FWL can count up to (1) 6-cycles; (2) all connected graphlets with size 4; (3) 4-paths at node level.*

*Proof.* Given Theorem 4.1, N²-FWL is more powerful than I²-GNN [19]. Given Theorem 4.1, 4.2, 4.3 in [19], I²-GNN [19] is able to count up to (1) 6-cycles; (2) all connected graphlets with size 4; (3) 4-paths at node level. This concludes the proof. □

Then, we prove Corollary 4.2 in the main paper. We restate it here:

**Corollary D.5.** *N²-FWL is strictly more powerful than all existing node-based subgraph GNNs and no less powerful than 3-WL.*

*Proof.* Given Theorem 4.1, this corollary can be directly concluded given Theorem 7.1 in [12] and Proposition 4.1 in [19]. But we further give an example to show that even with $h = 1$, N²-FWL is already powerful enough to distinguish some non-isomorphic strongly regular graph pairs that 3-WL cannot distinguish. When $h = 1$, the neighbors of N²-FWL is $(\mathcal{N}_1(v_2) \times \mathcal{N}_1(v_1)) \cap (\mathcal{N}_1(v_1) \cap \mathcal{N}_1(v_2))^2$. Consider the Shrikhande graph and 4×4 Rook's graph in Figure 2. It is well known this pair of graphs cannot be distinguished by 3-WL. First, look at node $v_1$ and $v_2$ in Shrikhande graph, $\mathcal{N}_1(v_1) \cap \mathcal{N}_1(v_2)$ includes two green nodes, $v_1$, and $v_2$. This means that $(w_1, w_2) \in (\mathcal{N}_1(v_2) \times \mathcal{N}_1(v_1)) \cap (\mathcal{N}_1(v_1) \cap \mathcal{N}_1(v_2))^2$ contains a pair of $(w_1, w_2)$ such that $w_1$ is one green node and $w_2$ is the other. Given Definition A.2, $(w_1, w_2) \in Q^F_{(w_1, w_2)}(v_1, v_2)$. This means that the information of $\hat{\mathcal{C}}^0(w_1, w_2)$ will be encoded into $\hat{\mathcal{C}}^1(v_1, v_2)$ and we have $(w_1, w_2)$ is not an

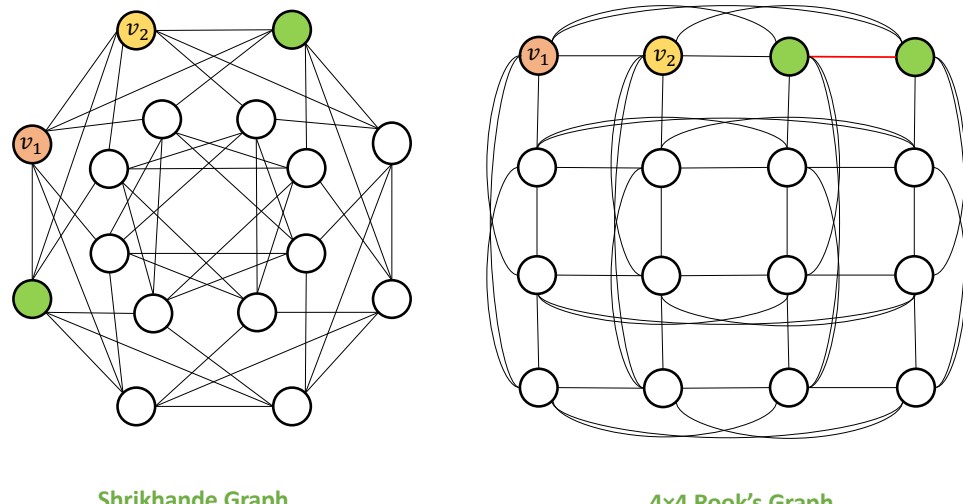

<p style="text-align:center">**Shrikhande Graph**    **4×4 Rook's Graph**</p>

Figure 2: The Shrikhande graph and $4 \times 4$ Rook's graph.

edge. Similarly, in $4 \times 4$ Rook's graph, $\mathcal{N}_1(v_1) \cap \mathcal{N}_1(v_2)$ includes two green nodes, $v_1$, and $v_2$. Thus, $\hat{\mathcal{C}}^0(w_1, w_2)$ will be encoded into $\hat{\mathcal{C}}^1(v_1, v_2)$. However, $(w_1, w_2)$ here is an edge, which will have a different isomorphism type than $(w_1, w_2)$ in the Shrikhande graph, thus resulting in different $\hat{\mathcal{C}}^1(v_1, v_2)$ and further graph color histogram. This concludes the proof. $\qquad\square$

Finally, we prove Proposition 4.4 in the main paper. We restate it here:

**Proposition D.6.** *If $\mathbf{U}^l$, $\mathbf{M}^l$, and $\mathbf{R}$ are injective functions, there exist a $N^2$-GNN instance that can be as powerful as $N^2$-FWL.*

*Proof.* If $\mathbf{U}^l$, $\mathbf{M}^l$, and $\mathbf{R}$ are injective, they are exactly the same as HASH used in $N^2$-FWL. Therefore, it is easy to prove it by doing induction on $l$ to show that if $N^2$-GNN outputs the same color for two tuples, $N^2$-FWL also does for any $l \geq 0$ and vice versa. We omit the detailed proof here. $\qquad\square$

### D.2   Complexity analysis

In this section, we provide a complexity analysis of $N^2$-GNN. Let $m$ denote the number of edges in the graph and $d$ denote the average degree of the graph. First, we consider $h$ is large enough, where the neighbor of $N^2$-GNN becomes $\mathcal{N}_1(v_2) \times \mathcal{N}_1(v_1)$. As $N^2$-GNN is an instance of $(2, 2)$-FWL, it is straightforward that the space complexity of $N^2$-GNN is upper-bounded by $O(n^2)$. Further, each tuple needs to aggregate $(d + 1)^2$ different neighbors, which results in a total time complexity of $O(n^2 d^2) = O(m^2)$. Notes that the time complexity of $N^2$-GNN is the same as edge-based subgraph GNNs. However, the space complexity of $N^2$-GNN is $O(n^2)$, which is far less than edge-based subgraph GNNs with a $O(nm)$ space complexity.

If we consider a pratical $h$, the neighbors of $N^2$-GNN is $(\mathcal{N}_1(v_2) \times \mathcal{N}_1(v_1)) \cap (\mathcal{N}_h(v_1) \cap \mathcal{N}_h(v_2))^2$. This means, only $(w_1, w_2)$ that within the overlapping $h$-hop subgraph of $v_1$ and $v_2$ can send message to $v_1, v_2$. Therefore, the time complexity is bounded by $O(nd^{h+2})$, which is still the same as edge-based subgraph GNNs with $h$-hop subgraph selection policy. Further, it is easy to verify that only tuple $(v_1, v_2)$ with $\mathrm{SPD}(v_1, v_2) \leq h$ can update its representation. Thus, in memory-sensitive tasks, we can only keep representation for tuples that can be updated, resulting in a space complexity of $O(nd^h)$.

### D.3   Model implementation details

Here we provide the details about model implementation. As $N^2$-GNN is an instance of $(2, 2)$-FWL+, we are dealing with a tuple of size 6 in the aggregation process. Note that for any tuple $(v_1, v_2)$, $Q^F_{(w_1, w_2)}(v_1, v_2)$ will also include itself. However, it does not increase the expressive power as

we already encode its information. Therefore, in the implementation, we remove it from the tuple. Thus, we implement the neighbor tuple as follows: for any 2-tuple $(v_1, v_2)$, the message from each neighbor $(w_1, w_2) \in N^2(v_1, v_2)$ is defined as $((v_1, w_1), (v_1, w_2), (w_1, v_2), (w_2, v_2), (w_1, w_2))$. Note the order of tuples is flexible as long as it is uniform across different neighbors, according to Definition A.2. We implement $N^2$-GNN in a memory-saving way. To make it more clear, we slightly redefine some notations. Let $h^l(v_1, v_2)$ be the representation of tuple $(v_1, v_2)$, $m^l_{(v_1,v_2)}(w_1, w_2)$ be the message from the neighbor $(w_1, w_2)$, $N^2(v_1, v_2)$ be the set of all neighborhoods of $(v_1, v_2)$ in $N^2$-GNN with the output embedding hidden size of $hs$. We let the initial feature of tuple $h^0(v_1, v_2) = \mathrm{MLP}(l_G(v_2) + \mathrm{SPD}(v_1, v_2))$. The message function is implemented as:

$$u^l_i(v_1, v_2) = \mathrm{LIN}_1(h^{l-1}(v_1, v_2)) * \mathrm{Tanh}(\mathbf{W}_i), \tag{47}$$

$$m^l_{(v_1,v_2)}(w_1, w_2) = \mathrm{LIN}_2(\mathrm{CON}(u^l_1(v_1, w_1), u^l_2(v_1, w_2), u^l_3(w_1, v_2), u^l_4(w_2, v_2), u^l_5(w_1, w_2))), \tag{48}$$

where $\mathrm{LIN}_1$ is a linear layer project the embedding from hidden size $h$ to a lower value $d$, named as inner size, $\mathbf{W}_i \in R^d, i \in [5]$ are learnable weight vector with size $d$ and $*$ is element-wise product, CON is the concatenation, and $\mathrm{LIN}_2$ is another linear layer project the concatenated embedding from size $5*d$ back to hidden size $h$. It is easy to validate that this implementation is injective to tuple input. Next, the update function $\mathbf{U}^l$ is implemented as:

$$h^l(v_1, v_2) = \mathrm{MLP}^l\left((1 + \epsilon^l) * h^{l-1}(v_1, v_2) + \sum_{(w_1, w_2) \in N^2(v_1, v_2)} m^l_{(v_1,v_2)}(w_1, w_2)\right), \tag{49}$$

where $\epsilon^l$ is a learnable scalar and $\mathrm{MLP}^l$ is a one-hidden-layer MLP with hidden size and output size equal to $h$ and activation function be ReLu. Note that in the implementation we do not consider the hierarchical multiset but directly pooling all neighbors together. We observe it already achieves superior results in all real-world tasks and is more parameter-saving.

As discussed in [12, 27], for each $h^{l-1}(v_1, v_2)$, add the representation of $h^{l-1}(v_2, v_2)$ can boost the performance in real-world tasks. Therefore, we adopt a similar way for all real-world experiments:

$$a^l(v_1, v_2) = \mathrm{MLP}^l_a\left((1 + \epsilon^l_a) * h^{l-1}(v_1, v_2) + h^{l-1}(v_2, v_2)\right), \tag{50}$$

$$\hat{h}^l(v_1, v_2) = h^l(v_1, v_2) + a^l(v_1, v_2). \tag{51}$$

If add this additional aggregation, we will use the $\hat{h}^l(v_1, v_2)$ instead of $h^l(v_1, v_2)$ as the output representation. We conjecture this works as virtual nodes in real-work tasks, which allows a better mixture of node features and structure information. As indicated by [12], adding this term will not increase the expressive power from the theoretical side. Finally, the readout function is implemented in a hierarchical way:

$$h(v_1) = \mathrm{MLP}\left(\mathrm{POOL}_1(\{\!\{h^L(v_1, w)|w \in V(G)\}\!\})\right), \tag{52}$$

$$h_G = \mathrm{POOL}_2\left(\{\!\{h(w)|w \in V(G)\}\!\}\right), \tag{53}$$

where $\mathrm{POOL}_1$ and $\mathrm{POOL}_2$ are two different pooling function.

In addition, we adopted two implementation strategies for $N^2$-GNN. The first strategy is to keep all $O(n^2)$ tuples even if it has a distance greater than $h$. Notes that it will not aggregate information from other tuples but is only updated in Equation (50) and Equation (51). We observe it has great performance in real-world tasks combined with Equation (52) and Equation (53). The second implementation is used for memory-sensitive tasks. We remove all tuples that have a distance greater than $h$. We refer to the first strategy as the dense version and the second strategy as the sparse version. We will specify which version we used in our experiments later on. The code is implemented using PyG [52]. We provide open-sourced code in https://github.com/JiaruiFeng/N2GNN for reproducing all results reported in the main paper.

## D.4 Limitations

Although $N^2$-GNN has efficient theoretical space complexity, the empirical space complexity can still be high as we need to keep neighbors index for all tuple $(u_1, u_2) \in Q^F_{(w_1, w_2)}(v_1, v_2)$. This introduces

a constant factor (5 in the implementation) on the saving edge index and corresponding gradients during training (see Table 9 and Table 10 for the practical time and memory comparison). It is still worth studying how to improve the empirical space complexity of N$^2$-GNN. Further, in the current implementation, we didn't consider the hierarchical set but directly pooled all elements together to save the parameters and computational cost. Additionally, each tuple needs to aggregate many more neighbors than normal MPNNs, which makes the sum aggregation hard to achieve injectiveness during the aggregation, introducing the optimization issue. These hinder the model from achieving its theoretical power. Finally, although N$^2$-GNN has $O(nd^{h+2})$ time complexity, which is practical for most real-world tasks, the complexity still goes exponentially if the graph is dense (like a strongly regular graph). But as the design space of $(k, t)$-FWL+ is extremely flexible, it is easy to design more practical and expressive instances suitable for dense graphs.

# E Experimental details

This section presents additional information on the experimental settings.

## E.1 Dataset statistics

Table 6: Dataset statistics.

| Dataset | #Graphs | Avg. #Nodes | Avg. #Edges | Task type |
|---------|---------|-------------|-------------|-----------|
| EXP | 1200 | 44.4 | 110.2 | Graph classification |
| CSL | 150 | 41.0 | 164.0 | Graph classification |
| SR25 | 15 | 25.0 | 300.0 | Graph classification |
| BREC | 800 | 34.8 | 143.4 | Pair distinguishment |
| Counting | 5,000 | 18.8 | 31.3 | Node regression |
| QM9 | 130,831 | 18.0 | 18.7 | Graph regression |
| ZINC-12k | 12,000 | 23.2 | 24.9 | Graph regression |
| ZINC-250k | 249,456 | 23.1 | 24.9 | Graph regression |

## E.2 Experimental details

In this section, we provide all the experimental details. First, we discuss some general experimental settings that, if not specified further, are consistent across all experiments.

**Model**. We adopt POOL$_1$ as summation and POOL$_2$ as mean pooling in Equation (52) and Equation (53). We set the $\epsilon^l$ in Equation (49), $\epsilon_a^l$ in Equation (50) to be 0.0 and fix it during training and inference. At each MLP, we add batch normalization.

**Training**. For all experiments, we use Adam as the optimizer and ReduceLRonPlateau as the learning rate scheduler. We set l2 weight decay to 0.0 across all experiments. All experiments are conducted on a single NVIDIA A100 80GB GPU.

### E.2.1 Graph structures counting

**Model**. For the substructure counting dataset, we adopt a similar setting as I$^2$-GNN [19]. Specifically, we set the number of hop $h$ to be 1, 2, 2, 3, 2, 2, 1, 4, 2 for 3-cycles, 4-cycles, 5-cycles, 6-cycles, tailed triangles, chordal cycles, 4-cliques, 4-paths, and triangle-rectangle respectively. For all substructures, we set the number of layers to 5, the hidden size $h$ to 96, and the inner size $d$ to 32. In the substructure count dataset, we remove all normalization layers. Finally, we use the sparse version of data preprocessing.

**Training**. The training/validation/test splitting ratio is 0.3/0.2/0.5. The initial learning rate is 0.001 and the minimum learning rate is 1e-5. The patience and factor of the scheduler are 10 and 0.9 respectively. The batch size is set to 256 and the number of epochs is 2000. For all substructures, we run the experiments 3 times and report the mean results on the test dataset.

### E.2.2 ZINC

**Model**. The setting is the same for both ZINC-Subset and ZINC-Full except for the inner size. We set the number of hop $h$ to be 3. We set the number of layers to 6, and the hidden size $h$ to 96. The inner size $d$ is 20 for the ZINC-subset and 48 for the ZINC-full.

**Training**. The initial learning rate is 0.001 and the minimum learning rate is 1e-6. The patience and factor of the scheduler are 20 and 0.5 respectively. The batch size is set to 128 and the number of epochs is 500. For both ZINC-Subset and ZINC-Full, we run experiments 10 times and report the mean results and standard deviation on the test dataset.

### E.2.3 QM9

**Model**. For all targets, we adopt the same setting. We set the number of hop $h$ to be 3. We set the number of layers to 6, the hidden size $h$ to 120, and the inner size $d$ to 32. We further add the resistance distance feature into the model, similar to the implementation in Feng et al. [22], Huang et al. [19].

**Training**. The initial learning rate is 0.001. The patience and factor of the scheduler are 5 and 0.9 respectively. The batch size is set to 128 and the number of epochs is 350. we run experiments 1 time and report the test result with the model of the best validation result.

### E.2.4 CSL

**Model**: We set the number of hop $h$ to be 4, the number of layers to be 4, the hidden size $h$ to 48, and the inner size $d$ to 16. We use the sparse version of data preprocessing and we do not add root aggregation (Equation (50)-(51)).

**Training**. The initial learning rate is 0.001 and the minimum learning rate is 1e-6. The patience and factor of the scheduler are 20 and 0.5 respectively. The batch size is set to 32 and the number of epochs is 80. We use 10-fold cross-validation and report the average results.

### E.2.5 SR25

**Model**: We set the number of hop $h$ to be 1, the number of layers to be 6, the hidden size $h$ to 64, and the inner size $d$ to 16. We use the sparse version of data preprocessing and we do not add root aggregation (Equation (50)-(51)). Further, the normalization layer in SR25 is set to Layer normalization to avoid test/training mismatch.

**Training**. The initial learning rate is 0.001 and the minimum learning rate is 1e-6. The patience and factor of the scheduler are 200 and 0.5 respectively. The batch size is set to 15 and the number of epochs is 800. We report the single-time results.

### E.2.6 EXP

**Model**: We set the number of hop $h$ to be 3, the number of layers to be 4, the hidden size to 48, and the inner size $d$ to 24. We use the sparse version of data preprocessing and we do not add root aggregation (Equation (50)-(51)).

**Training**. The initial learning rate is 0.001 and the minimum learning rate is 1e-6. The patience and factor of the scheduler are 20 and 0.5 respectively. The batch size is set to 32 and the number of epochs is 200. We use 10-fold cross-validation and report the average results.

### E.2.7 BREC

**Model**: We set the number of hop $h$ to be 8, the number of layers to be 4, the hidden size to 64, and the inner size $d$ to 32. Meanwhile, the output channel is set to 16 for computing the similarity. We use the sparse version of data preprocessing and we do not add root aggregation (Equation (50)-(51)).

**Training**. We follow the same training and evaluation procedure as described by BREC [45]. The initial learning rate is 0.001 and the weight decay is set to 0.0001. The batch size is set to 4 and the number of epochs is 20. Due to the resource limitation, we only ran the experiment on 340 graph pairs (removed 20 4-vertex-condition graphs and 40 CFI graphs), which follow the same protocol as the reported result for $I^2$-GNN.

# F Additional experimental results

## F.1 Ablation studies for $(k, t)$-FWL+

In this section, we provide an additional ablation study on $(2, t)$-FWL+ by varying different $t$ and $ES$. We select two important $ES$ mentioned in our paper—$Q_1(v)$ and $\mathcal{SP}(v_1, v_2)$. We conduct experiments on both Expressiveness verification datasets and counting datasets. The experiment setup is the same as what is discussed in the above section. The results can be seen in Table 7 and Table 8. Notes that we find the performance of $\mathcal{SP}$ or $\mathcal{SP} \times \mathcal{SP}$ on counting dataset is similar to MPNN and thus omit it in Table 8.

We can see that as long as $t = 2$, we can have a perfect performance on SR25 no matter if we use $\mathcal{SP}$ or $Q_1(v_1)$, this aligns with our theoretical results. Further, $\mathcal{SP}(v_1, v_2) \times Q_1(v_1)$ achieve great results on all counting tasks and expressiveness verification. This indicates the $\mathcal{SP}$ can also be a great choice in the practical scenario.

Table 7: Expressive power verification (Accuracy) for different $(2, t)$-FWL+.

| Datasets | $t=1, Q_1(v_1)$ | $t=1, Q_1(v_1) \cup Q_1(v_2)$ | $t=1, \mathcal{SP}(v_1, v_2)$ | $t=2, \mathcal{SP}(v_1, v_2) \times Q_1(v_1)$ | $t=2, \mathcal{SP}(v_1, v_2) \times \mathcal{SP}(v_1, v_2)$ |
|---|---|---|---|---|---|
| EXP | 100 | 100 | 100 | 100 | 100 |
| CSL | 100 | 100 | 100 | 100 | 100 |
| SR25 | 6.67 | 6.67 | 6.67 | 100 | 100 |

Table 8: Evaluation of different $(2, t)$-FWL+ variants on Counting Substructures (norm MAE), cells with MAE **greater** than 0.01 are colored.

| Target | $t=1, Q_1(v_1)$ | $t=1, Q_1(v_1) \cup Q_1(v_2)$ | $t=2, \mathcal{SP}(v_1, v_2) \times Q_1(v_1)$ | $N^2$-GNN |
|---|---|---|---|---|
| Tailed Triangle | 0.0033 | 0.0031 | 0.0022 | 0.0025 |
| Chordal Cycle | 0.0019 | 0.0017 | 0.0021 | 0.0019 |
| 4-Clique | 0.0013 | 0.0014 | 0.0016 | 0.0005 |
| 4-Path | 0.0046 | 0.0043 | 0.0076 | 0.0042 |
| Tri.-Rec. | 0.0043 | 0.0053 | 0.0081 | 0.0055 |
| 3-Cycles | 0.0005 | 0.0006 | 0.0004 | 0.0002 |
| 4-Cycles | 0.0027 | 0.0022 | 0.0037 | 0.0024 |
| 5-Cycles | 0.0051 | 0.0042 | 0.0037 | 0.0039 |
| 6-Cycles | 0.0113 | 0.0097 | 0.0097 | 0.0075 |

## F.2 Practical complexity of $N^2$-GNN

In this section, we provide practical complexity analysis for $N^2$-GNN. Here we use the cycle counting datasets with a similar parameter size for all models and a batch size of 256. To ensure a fair comparison, in $N^2$-GNN, we use the sparse version for data preprocessing (which is true for all compared models), and set the number of workers as 0 and random seed as 1 for all models. The experiments are run on a single TeslaV100 32GB GPU. We report the training time, maximum training memory, inference time, and maximum inference memory usage during inference for both $h = 1$ and $h = 2$ for an average of 20 epochs. The results is shown in Table 9 and Table 10.

We can see that the empirical memory usage of $N^2$-GNN is much less than $I^2$-GNN when $h = 1$ and comparable when $h = 2$. We conjecture the reason why $N^2$-GNN require more training memory than $I^2$-GNN when $h = 2$ is that when $h$ increase, each tuple will need to aggregate much more neighbors than each node in $I^2$-GNN and each aggregation require a constant size (5 in $N^2$-GNN) of more memory for saving the gradients. Meanwhile, As we mentioned in limitations (Appendix D), to enjoy some extent of parallelism, the current implementation of $N^2$-GNN needs to save all neighbor indices, which introduces a constant factor of more memory on saving. But the inference memory of $N^2$-GNN is less than $I^2$-GNN in both $h = 1$ and $h = 2$. The training time and inference time of $N^2$-GNN are also more efficient than $I^2$-GNN. However, currently, the time and memory cost by $N^2$-GNN are higher than node-based subgraph GNNs. We do want to highlight several points: (1) By changing the way of implementation, $N^2$-GNN can be implemented strictly within $O(n^2)$ space. Therefore, the merit of $N^2$-GNN still holds. (2) The experiment is conducted on the same parameter budget level. However, in real-world tasks, $N^2$-GNN needs much fewer parameters to achieve SOTA results (Table 5). (3) In this work, we not only aim to propose a new model but also mean to introduce a new flexible framework, $(k, t)$-FWL+. The empirical success of $N^2$-GNN demonstrates the great potential of this framework for designing new expressive GNN variants.

Table 9: Practical usage comparison for $h = 1$.

| $h = 1$ | # Params | Training time (ms/epoch) | Training memory (GB) | Inference time (ms/epoch) | Inference Memory (GB) |
|---|---|---|---|---|---|
| NGNN | 186k | 478.1 | 0.398 | 463.4 | 0.089 |
| ID-GNN | 135k | 455.5 | 0.375 | 432.8 | 0.094 |
| GNN-AK+ | 251k | 546.5 | 0.479 | 490.1 | 0.090 |
| $I^2$-GNN | 194k | 666.9 | 1.326 | 613.5 | 0.292 |
| $N^2$-GNN | 202k | 368.3 | 0.744 | 311.8 | 0.140 |

Table 10: Practical usage comparison for $h = 2$.

| $h = 2$ | # Params | Training time (ms/epoch) | Training memory (GB) | Inference time (ms/epoch) | Inference Memory (GB) |
|---|---|---|---|---|---|
| NGNN | 186k | 516.5 | 0.916 | 470.7 | 0.198 |
| ID-GNN | 135k | 521.7 | 0.858 | 471.5 | 0.211 |
| GNN-AK+ | 251k | 617.4 | 1.082 | 556.1 | 0.200 |
| $I^2$-GNN | 194k | 926.8 | 3.012 | 770.1 | 0.675 |
| $N^2$-GNN | 202k | 892.9 | 4.026 | 709.8 | 0.660 |

## F.3 Additional discussion on BREC experiment

In Table 11, we provide a complete comparison of all baseline methods on the BREC dataset. Please see the detailed description of all baseline methods in BREC [45]. We can see that $N^2$-GNN achieve the best result among all GNN baselines and the second best result among all baselines, even if we only test on 340 pair of graphs. Particularly, we surpass $I^2$-GNN on all parts. This result empirically verified Theorem 4.1 that the theoretical power of $N^2$-GNN is more powerful than $I^2$-GNN. Meanwhile, we notice that SSWL_P can distinguish more CFI graph pairs than ours, which may contradict Theorem 4.1. This may be due to the fact that we remove 40 CFI graph pairs in the experiments given the resource limitation and SSWL_P uses 8 layers in the experiment and we only use 4 layers. Meanwhile, we don't implement the hierarchical multiset in the current version of $N^2$-GNN, which slightly compromises the expressive power.

Table 11: Complete comparison on BREC dataset

| Model | Basic Graphs (60) | | Regular Graphs (140) | | Extension Graphs (100) | | CFI Graphs (100) | | Total (400) | |
|---|---|---|---|---|---|---|---|---|---|---|
| | Number | Accuracy | Number | Accuracy | Number | Accuracy | Number | Accuracy | Number | Accuracy |
| 3-WL | 60 | 100% | 50 | 35.7% | 100 | 100% | 60 | 60.0% | 270 | 67.5% |
| SPD-WL | 16 | 26.7% | 14 | 11.7% | 41 | 41% | 12 | 12% | 83 | 20.8% |
| $S_3$ | 52 | 86.7% | 48 | 34.3% | 5 | 5% | 0 | 0% | 105 | 26.2% |
| $S_4$ | 60 | 100% | 99 | 70.7% | 84 | 84% | 0 | 0% | 243 | 60.8% |
| $N_1$ | 60 | 100% | 99 | 85% | 93 | 93% | 0 | 0% | 252 | 63% |
| $N_2$ | 60 | 100% | 138 | 98.6% | 100 | 100% | 0 | 0% | 298 | 74.5% |
| $M_1$ | 60 | 100% | 50 | 35.7% | 100 | 100% | 41 | 41% | 251 | 62.8% |
| NGNN | 59 | 98.3% | 48 | 34.3% | 59 | 59% | 0 | 0% | 166 | 41.5% |
| DE+NGNN | 60 | 100% | 50 | 35.7% | 100 | 100% | 21 | 21% | 231 | 57.8% |
| DS-GNN | 58 | 96.7% | 48 | 34.3% | 100 | 100% | 16 | 16% | 222 | 55.5% |
| DSS-GNN | 58 | 96.7% | 48 | 34.3% | 100 | 100% | 15 | 15% | 221 | 55.2% |
| SUN | 60 | 100% | 50 | 35.7% | 100 | 100% | 13 | 13% | 223 | 55.8% |
| SSWL_P | 60 | 100% | 50 | 35.7% | 100 | 100% | 38 | 38% | 248 | 62% |
| GNN-AK | 60 | 100% | 50 | 35.7% | 97 | 97% | 15 | 15% | 222 | 55.5% |
| KP-GNN | 60 | 100% | 106 | 75.7% | 98 | 98% | 11 | 11% | 275 | 68.8% |
| $I^2$-GNN | 60 | 100% | 100 | 71.4% | 100 | 100% | 21 | 21% | 281 | 70.2% |
| PPGN | 60 | 100% | 50 | 35.7% | 100 | 100% | 23 | 23% | 233 | 58.2% |
| $\delta$-k-LGNN | 60 | 100% | 50 | 35.7% | 100 | 100% | 6 | 6% | 216 | 54% |
| KC-SetGNN | 60 | 100% | 50 | 35.7% | 100 | 100% | 1 | 1% | 211 | 52.8% |
| GSN | 60 | 100% | 99 | 70.7% | 95 | 95% | 0 | 0% | 254 | 63.5% |
| DropGNN | 52 | 86.7% | 41 | 29.3% | 82 | 82% | 2 | 2% | 177 | 44.2% |
| OSAN | 56 | 93.3% | 8 | 5.7% | 79 | 79% | 5 | 5% | 148 | 37% |
| Graphormer | 16 | 26.7% | 12 | 10% | 41 | 41% | 10 | 10% | 79 | 19.8% |
| $N^2$-GNN | 60 | 100% | 100 | 71.4% | 100 | 100% | 27 | 27% | 287 | 71.8% |

## F.4 Ablation studies for $N^2$-GNN

In this subsection, we perform ablation studies to investigate each component in $N^2$-GNN. All ablation studies are performed on ZINC-Subset and target $C_v$ in QM9. Except hyper-parameter mentioned later, all other setting is exactly the same as what is described in Section E.2.

In the first ablation study, we investigate different components of the model. Specifically, we analyze two components. The first component is aggregation $h^l(v_1, v_2)$ described in Equation (50)-(51). The second component is the dense/sparse embedding described in Section D.3. In the ablation study, we verify all different combinations of two components. For the model without Equation (50)-(51), to ensure that the size of the model is comparable, we increase the model hidden size to 120 and 136 for

ZINC-Subset and QM9, respectively. The results are shown in Table 12. First, we can see adding Equation (50)-(51) indeed improve the performance across different tasks. Second, by explicitly keeping $O(n^2)$ embedding the performance, of N$^2$-GNN on ZINC-Subset improved from 0.079 to 0.059. This may indicate that even though the two models have the same expressive power from the theoretical side, adding additional feature embedding can still boost the performance significantly in real-world tasks as it allows a better mixture of both feature and structure information. Note that these two components were used in many previous models like ESAN [43], SUN [27], and SSWL+ [12], etc. However, our model still outperforms them by a large margin. This indicates that the expressive power of the model is still the most critical ingredient, adding these additional components is just a way to unleash all the potential of the model for real-world tasks.

Table 12: Ablation study of model components.

| Model design | ZINC-Subset $\downarrow$ | QM9 ($C_v$) $\downarrow$ |
|---|---|---|
| Full | **0.059 $\pm$ 0.002** | **0.0760** |
| w/o $h^l(v_2, v_2)$ | 0.064 $\pm$ 0.003 | 0.0811 |
| sparse | 0.079 $\pm$ 0.003 | 0.0876 |
| w/o $h^l(v_2, v_2)$ + sparse | 0.084 $\pm$ 0.003 | 0.0847 |

Table 13: Ablation study of $h$.

| $h$ in N$^2$-GNN | ZINC-Subset $\downarrow$ | QM9 ($C_v$) $\downarrow$ |
|---|---|---|
| $h$=1 | 0.150 $\pm$ 0.009 | 0.0823 |
| $h$=2 | 0.127 $\pm$ 0.005 | 0.0808 |
| $h$=3 | **0.059 $\pm$ 0.002** | **0.0760** |
| $h$=4 | 0.063 $\pm$ 0.004 | 0.0826 |

In the second ablation study, we evaluate the effect of the number of hop $h$. We vary $h$ from 1 to 4 and results are shown in Table 13. We can see an increase of $h$ can boost the performance of the model, which has already been verified in previous subgraph-based methods. Particularly, for ZINC-Subset, we find that the third hop is critical to the performance. However, if we continue to increase the number of hops, the performance drops, which is a sign of overfitting.

