# OpenReview forum: "Extending the Design Space of Graph Neural Networks by Rethinking Folklore Weisfeiler-Lehman"
_NeurIPS.cc/2023/Conference — NeurIPS 2023 poster_

### Official Review · Reviewer_vTyW · 2023-07-01

**Soundness:** 3 good
**Presentation:** 3 good
**Contribution:** 3 good
**Rating:** 7
**Confidence:** 3

**Summary:**

The paper builds an extension to the $k$-FWL isomorphism test algorithms called $(k,t)$-FWL+. The proposed extended family subsumes the existing $k$-FWL and adds finer variants to the expressivity landscape of isomorphism testing algorithms. The $(k,t)$-FWL+ algorithm is a combination of two modified schemes to the original $k$-FWL algorithm.

The first one, called $(k,t)$-FWL, which also colors $k$ tuples, and draws insparation from the tuple based aggregation of the $k$-FWL algorithm vs. $k$-WL. The proposed algorithm changes the construction of neighborhood multi-stets used in the encoding step of the $k$-FWL algorithm. Briefly explained, instead of consisting of $n$ tuples of size $k$ , $(k,t)$-FWL aggregates a multi-sets consisting of $n^t$ tuples of size $\sum_{i=0}^{\min (k,t)}{ k \choose i} {t \choose i}$. Where each tuple in the new proposed algorithm holds more information than in the original $k$-FWL algorithm. The authors prove that for a fixed $k$, increasing $t$ induces a strict hierarchy where there exist a large enough $t$ for which the algorithm solves the isomorphism problem.

The second modification, called $k$-FWL+, unifies many previously introduced approaches, and suggests modifying the neighborhoods of tuples to be equivariant sets of nodes instead of the entire node set of the graph. Which can reduce time complexity of the algorithm.

The main contributions of the paper are (i) introducing a novel finer hierarchy of isomorphism testing algorithms; (ii) showing that it is possible to construct a strict hierarchy at fixed memory complexity that can achieve solving the isomorphism problem; (iii) coming up with a practical neural instantiation; and (iv) achieving state of the art results on common benchmarks.





**Strengths:**

- The paper is well written and successfully introduces the limitations of the $k$-WL hierarchy for the evaluation and design of more expressive GNNs. It is part of a body of works that refine the WL hierarchy aiming for more efficient algorithms with higher expressive power.

- The paper introduces a general framework that is also shown to unify several recent approaches and can provide a canonical design space for previous and future GNNs.
- The construction of the $(k,t)$-FWL hierarchy naturally arises from the original $k$-FWL algorithms and elegantly shifts memory complexity to runtime complexity in the fixed $k$ scenario.
- The experimental evaluation is extensive and the proposed method seems to achieve SOTA performance.





**Weaknesses:**

- The paper does not properly discuss the "no free lunch" that hides in the proposed method. Indeed, the authors came up with a fixed space complexity hierarchy which, for large enough parameter $t$, can solve the isomorphism problem. But, this comes at the cost of time complexity - having to aggregate an exponentially growing number of tuples ($O(n^t)$). The authors deferred most of this discussion to the appendix, while I believe it should be placed in the main body of the paper.
- A valuable ablation study lacking in the paper is an experiment showing the performance of several variants of the new proposed hierarchy on the substructure counting benchmark. Since it is unclear what each variant gains in expressiveness, I believe such an evaluation will contribute to the paper's intuition and wholeness.
- Questionable results in the QM9 dataset. See the questions section.



**Questions:**

- A neural counterpart is suggested only for a single instance in the hierarchy. An interesting realm to explore would be to find a neural analog to the proposed hierarchy (similar to [1])
- There seems to be a mismatch in the units for the QM9 experiment between your code and the code in the $I^2$-GNN repository [2]. They seem to convert back to previously used units to be consistent with previous works while your code uses the new conversion. Can the authors please clarify this issue? To my understanding, for example, the value that should be reported for $\tilde{U}_0=U_0/0.04336414=0.5744$, where $U_0$ is the value reported in this paper.
- Can the authors report runtimes of their GNN training and how it compares to other approaches? Since their approach trades space complexity for runtime complexity, I think it is necessary to show that the method remains scalable where the authors hinted that runtimes might increase when graphs are dense.

- In the appendix, the authors say (l1094): "...we observe it some time hard to optimize
1095 the model to achieve its theoretical power by using fewer embedding". Can the authors clarify this statement?


[1] Provably Powerful Graph Neural Networks. Maron et. al. (NuerIPS 2019)

[2] [$I^2$-GNN GitHub](https://github.com/GraphPKU/I2GNN/blob/master/run_qm9.py#L37)




**Limitations:**

- The authors describe the limitations of the method mainly in the appendix, I think some of the discussion should be moved to the main body (see questions section)

**General comment:** I lean towards raising my score upon clarification of the experimental results. I think the proposed framework is a natural extension of the $k$-FWL family that elegantly formulates the trade-off between time and space complexity.

---

> ### Author Rebuttal · Authors · 2023-08-07
>
> Thank you for your acknowledgment of our contribution and constructive comments! We reply to all your concerns below.
>
> ### Weaknesses:
>
> >1. Further discussion on $(k, t)$-FWL in the main paper.
>
>
> **Answer:**
> Thanks for the helpful comment! We mentioned the exponentially growing on the time complexity when increase the $t$ in $(k, t)$-FWL in line 159-160. However, due to the space limitation, we can only give more discussion in Appendix. We will provide in-depth discussion on the "no-free lunch" in the main paper in the revision.
>
> >2. Experiment on different variant of $(k, t)$-FWL+ hierarchy.
>
> **Answer:**
> Thanks for the insightful comment! In general response point 3, we provide further ablation study on testing different $ES$ and $t$ on counting substructure dataset. In the experiment, we can observe clear expressive power difference on different variants.
>
> ### Questions:
> >1. Neural analog to the proposed hierarchy.
>
> **Answer:**
> Although we didn't mention the neural analog of $(k, t)$-FWL+ but only $N^2$-FWL. It is straightforward to obtain neural analog for any variant of $(k, t)$-FWL+ by replacing $HASH$ with MLP, multiset pooling with injective multiset function like DeepSet [1], or summation. We will add detailed discussion in the revision.
>
> >2. Issue on the QM9 experiment.
>
>  **Answer:**
> We check our code and the code provided in $I^2$-GNN Github again. We believe we are following the same evaluation standard as previous works including $I^2$-GNN [2], KP-GNN [3], NGNN [4]. For all methods including ours, we train the model using the unconverted but normalized target value. After training, we first un-normalize it by multiplying the standard deviation and then use the conversion to convert the value and report the converted one. The only difference between ours and $I^2$-GNN is that they use a slightly different constant for **HAR2EV** in the conversion, where we use 27.211386246 and $I^2$-GNN use 27.2113825435. However, the value we used is consistent with the original code in PyG [5], KP-GNN [3], and NGNN [4]. Additionally, the constant factor for $U_0$ is HAR2EV=27.211386246 but not KCALMOL2EV=0.04336414. Therefore, we believe the comparison of the QM9 dataset is fair.
>
> >3. Practical usage of $N^2$-GNN.
>
> **Answer:**
> In general response point 2, we provide further analysis and discussion on both time and memory complexity of $(k, t)$-FWL and $N^2$-GNN.
>
> >4. Optimization issue on the $(k, t)$-FWL+.
>
> **Answer:**
> Thanks for mentioning that! The key point here is the injectiveness of the multiset function. For example, if we use large $t$ and small $k$ for $(k, t)$-FWL+ to conduct graph isomorphism test, for each $k$-tuple, the hierarchical multiset $\{\{\}\}_t$ will contains exponentially increased elements. So far, we only use the summation as the function to encode this hierarchical multiset. Although it is an injective function from a theoretical view, we find it works badly with information loss when the number of elements is too large. A similar issue can be founded in over-squashing [6]. However, we believe this issue can be alleviated by more powerful injective multiset functions like DeepSet [1] or increase the width of the embedding.
>
>
> **References**
>
> [1] Zaheer et al., Deep Sets, NeurIPS17.
>
> [2] Huang et al., Boosting the cycle counting power of graph neural networks with i$^2$-GNNs, ICLR23.
>
> [3] Feng et al., How powerful are k-hop message passing graph neural network, NeurIPS22.
>
> [4] Zhang and Li, Nested graph neural networks, NeurIPS21.
>
> [5] Fey and Lenssen, Fast Graph Representation Learning with PyTorch Geometric, ICLR19 workshops.
>
> [6] Topping et al., Understanding over-squashing and bottlenecks on graphs via curvature, ICLR22.

---

> > ### Comment · Reviewer_vTyW · 2023-08-14
> >
> > I thank the authors for their answers and clarification of the experimental results.
> >
> > I agree with the other reviewers on the overselling (or the 'no free lunch') of the memory complexity O(n^2).
> > However, I think this paper presents a general framework, whether one decides to frame it as a WL-hierarchy of a subgraph enhancement approach, that introduces flexibility in the design of algorithms and unifies existing ones.

---

> > > ### Author Response · Authors · 2023-08-15
> > >
> > > Thanks again for your efforts in reviewing our work and providing insightful suggestions to improve our work. We will try our best to incorporate all of your comments in the revision.

---

### Official Review · Reviewer_eNLB · 2023-07-04

**Soundness:** 3 good
**Presentation:** 3 good
**Contribution:** 2 fair
**Rating:** 4
**Confidence:** 5

**Summary:**

The Weisfeiler-Leman algorithm has been established as the formal tool of choice for measuring the expressivity of GNNs. The paper tackles the problem of designing "higher-order" GNNs, which has been previously considered by several papers.

The authors propose and combine two new ideas to supplement k-WL: (1) add in further subgraph information in a space-efficient manner via a hyperparameter $t$, and (2) allow neighbor selection in folklore-k-WL from any equivariant fashion, instead of the standard sets such as the set of all vertices or the set of all neighbors.

Combining these two ideas yields the (k,t)-FWL+ algorithm. The version for k=2 and t=2, i.e. (2,2)-FWL+, is thoroughly evaluated on several datasets, and it yields very good experimental results, esp. on the ZINC dataset.


**Strengths:**

1) The (2,2)-FWL+ performs very well on several datasets. On the ZINC datasets, they report 10.6% and 40.9% improvement over the second-best methods.

2) The authors identify the problem of space limitations for higher-order GNNs and they seek to address them in their model.


**Weaknesses:**

1) The proposed algorithm lacks novelty: the claimed theoretical contributions are superficial. Graph Isomorphism can be obviously solved in quadratic space, if we were to allow iteration over all possible mappings between the two vertex sets. Such an algorithm might be space-efficient on paper but does not indicate any tractability since it brute-force tries all possible mappings. The proofs outlining the power and limitations of (k,t)-FWL are mild generalizations of known results, and lack any novel insights.

2) Allowing equivariant sets for vertex choice in k-FWL is a reasonable idea: however, I would expect the authors to compare several "equivariant set strategies" and demonstrate the relative gains over non-classic equivariant sets such as V(G) or N(v,G), keeping other things equal (an ablation study).

Overall, I feel that the paper is not much different from a multitude of papers on designing higher order GNNs using subgraph enhancement. Given the work that had already been done on this problem, the proposed model is incremental.

**Questions:**

-

**Limitations:**

Yes.

---

> ### Author Rebuttal · Authors · 2023-08-07
>
> Thank you for your helpful comments! We reply all concerns below.
> ### Weaknesses:
> >1. Further clarification on the novelties.
>
> **Answer:**
> We agree that the Graph Isomorphism problem can be solved by brute-force enumerating all possible mappings in quadratic space. However, our $(k, t)$-FWL does not aim to do this but aims to provide a flexible and theoretically-guaranteed way to balance the expressive power and space complexity. $(k, t)$-FWL also brings a new view on how to trade-off the memory and time complexity of WL-based algorithms, which was not mentioned in any previous literature (previous $k$-FWL-based algorithms' time and space complexity **need to increase simultaneously**). In our general response point 1, contribution (1), we further clarified the contribution of $(k, t)$-FWL.
>
> >2. Further study on the $ES$.
>
> **Answer:**
> In our general response point 3, we provide further ablation study on testing different $ES$ and $t$ using the expressiveness and counting datasets.
>
> We kindly refer the reviewer to take the time to go through the general response and let us know if there are any further concern. We are more than glad to discuss them all!

---

> > ### Comment · Reviewer_eNLB · 2023-08-14
> >
> > I thank the authors for addressing my questions.
> >
> > My main problem with the paper is: I am not convinced that the "quadratic space" perspective is a significant contribution which deserves to be the centerpiece of this paper. My reasons stand as follows.
> >
> > The presented work (the “t”-extenstion and the “+”-extension) is more in the nature of feature extraction, rather than something which yields actual insights into how and why WL learns on graphs. The claimed premise that “quadratic-space WL is the right inductive bias that unifies several existing architectures” is too strong a statement which requires much better evidence that what is presented.
> >
> > Theoretically, the main results (Proposition 3.1, Theorem 3.2 and Proposition 3.3) and their proofs are either trivial or comprise trivial modifications to existing literature. Yes, indeed, the paper points out one could re-parameterize Weisfeiler-Leman from $k$ to $(k,t)$ to obtain a finer hierarchy. Yet, in my opinion, the main contribution of this re-parameterization is the following: the (k,t)-FWL algorithm performs "subgraph enhancement using subgraphs of size $t$" in "quadratic space" instead of "potentially $n^t$ space". Hence, the proposed re-parameterization is more of a \emph{subgraph enhancement} GNN which works in quadratic space. It is in this context that I find the novelty lacking, given the plethora of existing "subgraph enhancement" GNNs already.
> >
> > Experimentally, the most promising results are for the case k=2 and t=2. The "2-tuple" regime is also heavily researched with a fairly big crowd of similar architectures for aggregating information for 2-tuples: It is not clear at all if the empirical success of (2,2)-FWL+ can be attributed solely to the "quadratic space" inductive bias, or something else. It seems to me that the empirical success can be attributed mainly to the "+"-extension (i.e. choosing equivariant sets) rather than the "t"-extension (i.e. using small space).
> >
> > It would be much better if the authors had built this paper around their solid contribution, namely the (2,2)-FWL+ algorithm, which speaks for itself with its strong performance on the ZINC dataset. Instead the authors have chosen to focus on the quadratic space perspective, which is not very insightful, at least to me, and forms an artificial framework that I find hard to justify. In its current form, the paper does not meet the NeurIPS bar in my opinion.

---

> > > ### Author Response · Authors · 2023-08-15
> > > **Follow-up discussion part 1**
> > >
> > > We thank the reviewer for the follow-up discussion. We are happy to further clarify them all.
> > >
> > > > 1. The presented work (the “t”-extenstion and the “+”-extension) is more in the nature of feature extraction, rather than something which yields actual insights into how and why WL learns on graphs.
> > >
> > > **Answer:**
> > > Why WL can learn on graphs is a complex question. In $k$-WL/FWL, the isomorphism type of each $k$-tuple ($k$-subgraph) is encoded as the initial color. As long as the $k$ increases, $k$-WL/FWL can naturally encode higher-order subgraph information about the graph and learn more fine-grained information through the message passing. However, in order to keep permutation invariant, $k$-WL/FWL **needs to encode all possible $k$-tuples**, which is why $k$-WL/FWL has the **concurrent growth of both the time and space complexity**. Our $(k, t)$-FWL is motivated exactly by this limitation as well as the difference between $k$-WL and $k$-FWL: $k$-FWL could infer higher-order structure information through the **tuple-style** aggregation using the same space complexity as $k$-WL. Then, the $(k, t)$-FWL is designed to extend this **insight** to encode arbitrary higher-order structures (controlled by $t$) with fixed-length tuples (controlled by $k$). Through theoretical results (Proposition 3.1, 3.3, Theorem 3.2), we show that $(k, t)$-FWL constructs **a full expressiveness hierarchy with any fixed-length tuples**. The $(k, t)$-FWL+ further extends the design space by allowing arbitrary equivariant sets to define neighbor tuples. Therefore, our work **at least provides the following insights** into WL: 1) The space complexity need not grow with the time complexity to define a full hierarchy, and 2) $k$-WL/FWL has a much broader design space than originally thought by extending it to $(k, t)$-FWL+. Our work is not in the nature of feature extraction, but rather a general new framework for designing arbitrarily expressive GNNs.
> > >
> > > > 2. The claimed premise that “quadratic-space WL is the right inductive bias that unifies several existing architectures” is too strong a statement which requires much better evidence that what is presented.
> > >
> > > **Answer:**
> > > we never stated that “quadratic-space WL is the right inductive bias that unifies several existing architectures”. Instead, $(k, t)$-FWL is a framework that not only works on quadratic space, but can construct an expressiveness hierarchy given any fixed space complexity ($O(n^k), k \geq 2$). That is also one of the key reasons why $(k, t)$-FWL+ is very flexible and can unify several existing architectures. So far, we support our claim with a series of theoretical results (**Proposition 3.4-3.9**), which already cover most of the existing works discussed in our related works (Section 5). In fact, there might be a misunderstanding on the "$O(n^2)$ Space" in our title. We emphasize the "$O(n^2)$ Space" mainly to deliver a message that the space complexity can be fixed instead of growing with time for defining a WL hierarchy. We did not mean that the $O(n^2)$ space should be the right or only inductive bias. Instead, $O(n^3), O(n^4)$ may work even better for different problems, as long as we can afford the memory cost.
> > >
> > > > 3. Theoretically, the main results (Proposition 3.1, Theorem 3.2 and Proposition 3.3) and their proofs are either trivial or comprise trivial modifications to existing literature.
> > >
> > > **Answer:**
> > > Although our proof leverages some existing techniques used in the previous works [1-2], we do want to highlight that the core part of our proof is how to represent high-dimensional tuples using low-dimensional tuples, which, to our best knowledge, does not exist in any previous work and is nontrivial to do. Furthermore, our main contributions stand on the design of the $(k, t)$-FWL+ framework with a theoretical guarantee instead of proposing novel proof strategies.
> > >
> > > > 4. the main contribution of this re-parameterization is the following: the (k,t)-FWL algorithm performs "subgraph enhancement using subgraphs of size
> > > $t$ in "quadratic space" instead of "potentially $n^t$ space".
> > >
> > > **Answer:**
> > > To clarify, $(k, t)$-FWL can encode subgraphs of size $k+t$ with $O(n^k)$ space instead of $O(n^{k+t})$ space for $(k+t)$-WL. Only when $k=2$, we can have a quadratic space complexity. The idea of encoding subgraphs of size $k+t$ with $O(n^k)$ space never shows in any previous works and is a novel contribution of our work.
> > >
> > > We kindly refer reviewer to discussion part 2 for further clarification.

---

> > > > ### Author Response · Authors · 2023-08-15
> > > > **Follow-up discussion part 2**
> > > >
> > > >
> > > > > 5. Hence, the proposed re-parameterization is more of a **subgraph enhancement** GNN which works in quadratic space. It is in this context that I find the novelty lacking, given the plethora of existing "subgraph enhancement" GNNs already.
> > > >
> > > > **Answer:**
> > > > In our Related works (Section 5), we provide a comprehensive discussion on existing subgraph enhancement GNNs. Here we give a more detailed discussion of the difference between theirs and ours.
> > > >
> > > > **Subgraph + MPNN[2-13]:** This type of method designs different subgraph extraction strategies and then runs MPNN on these subgraphs and aggregates them together to get final results. The difference between them and ours is that: (1) All of them require copying and separately running the model on all subgraphs. Suppose there are $s$ subgraphs, the space complexity is $O(sn)$, which results in $O(n^2)$ for node-based strategies; $O(nm)$ for edge-based strategies; $O(n^{k+1})$ for $k$-tuple based strategies. However, $(k, t)$-FWL can be implemented in any fixed space complexity. (2) All of them use MPNN as the base encoder. In [2], authors show that the advantage of the tuple-based aggregation is theoretically more expressive than local node aggregation used in MPNN. In the follow-up discussion to point 6, we will further support it from the empirical side.
> > > >
> > > > **Subgraph + $k$-WL[14]:** [14] propose $k, l$-WL, which run $k$-WL on all $l$-tuples' subgraphs. As we discussed in lines 164-165, they also require $O(n^{k+l})$ space complexity to achieve a comparable expressive power to $(k +l)$-WL.
> > > >
> > > > **Injecting subgraph features [15-20]:** This type of method tries to inject high-order subgraph features that cannot be learned by MPNN into the model to improve the expressive power of the MPNN. The difference between them and ours is that: (1) They need to explicitly design what type of subgraph features to encode into the model, which requires additional domain knowledge and is hard to generalize. Instead, $(k, t)$-FWL relies on the aggregation to automatically learn such patterns. (2) Further, injecting features into the model mostly works on the synthetic datasets but becomes easy to overfit on real-world datasets and the results are sub-optimal (e.g. performance on ZINC-subset/ZINC-full). Instead, our method has a better ability to integrate structure information and feature information to achieve SOTA results on real-world tasks.
> > > >
> > > > **High-order message-passing scheme [21-23]:** This type of method designs different message-passing schemes on high-order substructures like cycles or paths. The main differences between them and ours are: (1) the substructure and the message passing scheme need to be pre-designed, which requires additional domain knowledge and is hard to extend. (2) The space complexity depends on the message passing scheme and could still be high given the different choices of substructure/message-passing scheme. Instead, $(k, t)$-FWL can construct an expressiveness hierarchy within a fixed space complexity.
> > > >
> > > > We kindly refer the reviewer to discussion part 3 for the further clarification.

---

> > > > > ### Author Response · Authors · 2023-08-15
> > > > > **Follow-up discussion part 3**
> > > > >
> > > > > > 6. Experimentally, the most promising results are for the case k=2 and t=2. The "2-tuple" regime is also heavily researched with a fairly big crowd of similar architectures for aggregating information for 2-tuples: It is not clear at all if the empirical success of (2,2)-FWL+ can be attributed solely to the "quadratic space" inductive bias, or something else. It seems to me that the empirical success can be attributed mainly to the "+"-extension (i.e. choosing equivariant sets) rather than the "t"-extension (i.e. using small space).
> > > > >
> > > > > **Answer:** Thanks for your comments. Theoretically, we already proved that the $t$ extension helps to increase the expressive power of the $N^2$-FWL (surpass both node-based subgraph GNNs and edge-based subgraph GNN) within $O(n^2)$ space. To further verify the effectiveness of each component in our framework, we provide an additional ablation study on ZINC-subset.
> > > > >
> > > > > | Model  | Space complexity | ZINC-subset. |
> > > > > |:---:|:---:|:---:|
> > > > > |SSWL+ [2]|$O(n^2)$|0.070 $\pm$ 0.005|
> > > > > |I$^2$-GNN [13]| $O(nm)$ |$0.083 \pm 0.001$|
> > > > > |3-IDMPNN [14]| $O(n^4)$ | $0.085 \pm 0.003$|
> > > > > |$t=1, k=2$, $ES$ is the one-hop neighbors of $v_1$| $O(n^2)$ |0.065 $\pm$ 0.003|
> > > > > |$t=1, k=2$, $ES$ is the one-hop neighbors of either $v_1$ or $v_2$| $O(n^2)$ |0.065 $\pm$ 0.004|
> > > > > |t=2, k=2, **N$^2$-GNN**| $O(n^2)$ |**0.059 $\pm$ 0.002**|
> > > > >
> > > > > The results of SSWL+, I$^2$-GNN, and 3-IDMPNN are reported from their papers and the results for last three rows are reported on an average of 10 runs with a similar model architecture and parameter size. The $ES$ in the N$^2$-GNN is the product of the one-hop neighbors of $v_1$ and the one-hop neighbors of $v_2$. From the results, we have the following conclusions:
> > > > >
> > > > > (1) I$^2$-GNN and 3-IDMPNN require higher space complexity than $O(n^2)$ theoretically but achieve sub-optimal results compared to methods in $O(n^2)$. This shows that a small space might indeed bring benefits to real-world tasks. Nevertheless, 3-IDMPNN subsamples the subgraphs, so a fair comprison to full $O(n^4)$ models is lacking.
> > > > >
> > > > > (2) Increasing $t$ from 1 to 2 also contributes hugely to the final result (by comparing the second/third to last row and the last row). This further verifies our design principle---increasing the expressive power by increasing $t$ successfully improves the results on real-world tasks.
> > > > >
> > > > > (3) The second to the last row can be viewed as a tuple-style aggregation version of SSWL+ (using MPNN as the base encoder). The performance gain empirically verified the superiority of tuple-style aggregation used in $(k, t)$-FWL+ compared to MPNN used in most of the existing subgraph-enhancement GNN.
> > > > >
> > > > > To conclude, the ablation study supports all of our claims and further verifies the effectiveness of the quadratic space and each component in the $(k, t)$-FWL+.
> > > > >
> > > > >
> > > > > > 7. It would be much better if the authors had built this paper around their solid contribution, namely the (2,2)-FWL+ algorithm, which speaks for itself with its strong performance on the ZINC dataset. Instead the authors have chosen to focus on the quadratic space perspective, which is not very insightful, at least to me, and forms an artificial framework that I find hard to justify. In its current form, the paper does not meet the NeurIPS bar in my opinion.
> > > > >
> > > > > **Answer:**
> > > > > Thanks for the suggestion. We do admit that our title "Towards Arbitrarily Expressive GNNs in $O(n^2)$ Space by Rethinking Folklore Weisfeiler-Lehman" may make the reviewers confused about our contributions. Here we would like to emphasize:
> > > > >
> > > > > (1) Indeed, solving graph isomorphism problem in $O(n^2)$ space is not new and has already been explored in previous literature like relational pooling [24]. However, the main contribution of $(k, t)$-FWL is not to solve graph isomorphism problem in quadratic space but to construct **an expressiveness hierarchy** like $k$-FWL/WL but **only within a fixed space complexity $O(n^k)$** for any $k \geq 2$. This idea never appears in previous works.
> > > > >
> > > > > (2) In this work, we not only aim to propose a new model N$^2$-GNN but also mean to introduce a new flexible framework, $(k, t)$-FWL+. Therefore, we believe limiting the discussion to $(2, 2)$-FWL+ is not appropriate and might compromise our overall contributions. Nevertheless, we appreciate the reviewer's acknowledgement on N$^2$-GNN's strong empirical performance.
> > > > >
> > > > > (3) In the revision, we will further improve our work by 1) making a better clarification on the main contributions of our proposed frameworks and 2) giving a detailed discussion on the limitations of all the proposed methods mentioned by all the reviewers.
> > > > >
> > > > > We kindly refer the reviewer to discussion part 4 for all references mentions in the follow-up discussion.

---

> > > > > > ### Author Response · Authors · 2023-08-15
> > > > > > **Follow-up discussion part 4**
> > > > > >
> > > > > > **References:**
> > > > > >
> > > > > > [1] Morris et al., Weisfeiler and leman go sparse: Towards scalable higher-order graph embeddings, NeurIPS20.
> > > > > >
> > > > > > [2] Zhang et al., A complete expressiveness hierarchy for subgraph gnns via subgraph weisfeiler-lehman tests, ICML23.
> > > > > >
> > > > > > [3] Zhang and Li. Nested graph neural networks, Neurips21.
> > > > > >
> > > > > > [4] Zhao et al., From stars to subgraphs: Uplifting any GNN with local structure awareness, ICLR22.
> > > > > >
> > > > > > [5] Sandfelder et al., Ego-gnns: Exploiting ego structures in graph neural network, ICASSP21.
> > > > > >
> > > > > > [6] Frasca et al., Understanding and extending subgraph GNNs by rethinking their symmetries, Neurips22.
> > > > > >
> > > > > > [7] You et al., Identity-aware graph neural networks, AAAI21.
> > > > > >
> > > > > > [8] Li et al., Distance encoding: Design provably more powerful neural networks for graph representation learning, Neurips20.
> > > > > >
> > > > > > [9] Papp et al., DropGNN: Random dropouts increase the expressiveness of graph neural networks, Neurips21.
> > > > > >
> > > > > > [10] Cotta et al., Reconstruction for powerful graph representations Neurips21.
> > > > > >
> > > > > > [11] Bevilacqua et al., Equivariant subgraph aggregation networks, ICLR22.
> > > > > >
> > > > > > [12] Qian et al., Ordered subgraph aggregation networks, Neurips22.
> > > > > >
> > > > > > [13] Huang et al., Boosting the cycle counting power of graph neural networks with i$^2$-GNNs. ICLR22.
> > > > > >
> > > > > > [14] Zhou et al., From relational pooling to subgraph gnns: A universal framework for more expressive graph neural networks, ICML23.
> > > > > >
> > > > > > [15] Bouritsas, et al., Improving graph neural network expressivity via subgraph isomorphism counting, Arxiv, 2020.
> > > > > >
> > > > > > [16] Wijesinghe and Wang, A new perspective on ”how graph neural networks go beyond weisfeiler-lehman?" ICLR22.
> > > > > >
> > > > > > [17] Feng, et al., How powerful are k-hop message passing graph nerual networks, Neurips22.
> > > > > >
> > > > > > [18] Kong et al., Geodesic graph neural network for efficient graph representation learning, Neurips22.
> > > > > >
> > > > > > [19] Yan et al., Efficiently counting substructures by subgraph gnns without running gnn on subgraphs, Arxiv, 2023.
> > > > > >
> > > > > > [20] Puny et al., Equivariant polynomials for graph neural networks, ICML23.
> > > > > >
> > > > > > [21] Bodnar et al., Weisfeiler and Lehman Go Topological: Message Passing Simplicial Networks, LCML21.
> > > > > >
> > > > > > [22] Bodnar et al., Weisfeiler and lehman go cellular: CW networks. Neurips21.
> > > > > >
> > > > > > [23] Thiede et al., Autobahn: Automorphism-based Graph Neural Nets, Neurips21.
> > > > > >
> > > > > > [24] Murphy et al., Relational pooling for graph representations, ICML19.

---

> > > ### Author Response · Authors · 2023-08-19
> > >
> > > Dear reviewer eNLB:
> > >
> > > We would like to sincerely thank you again for your efforts and constructive comments on our work. As the discussion period ends soon, could we kindly know if the responses have addressed your concerns and if further explanations or clarifications are needed? We provide a detailed follow-up discussion on (1) comparing different subgraph enhancement methods (point 5), (2) additional ablation study on methods with different space complexity (point 6), and (3) further clarification of our work (points 1-4, 7). If there is no further concern, we kindly hope that the reviewer could consider re-evaluating our work. Thank you again.
> > >
> > > Authors.

---

> > > > ### Comment · Reviewer_eNLB · 2023-08-19
> > > >
> > > > Thanks for your detailed comments. I will respond to only some of these comments, which I believe are the most relevant to the discussion.
> > > >
> > > > Point 1:
> > > > "Therefore, our work at least provides the following insights into WL: 1) The space complexity need not grow with the time complexity to define a full hierarchy, and 2) k-WL/FWL has a much broader design space than originally thought by extending it to k-FWL+. Our work is not in the nature of feature extraction, but rather a general new framework for designing arbitrarily expressive GNNs."
> > > >
> > > > As I mentioned before, Graph Isomorphism can be trivially solved in quadratic space if we are ready to brute-force try all permutations between two graphs. To be more precise, the (k,t)-FWL algorithm yields a full hierarchy because for a graph on $n$ vertices, one can always make $t$ large enough to satisfy $k+t > n$, at which point the algorithm is essentially no better than brute-force trying all $O(t!)$ mappings between graphs.
> > > >
> > > > That "one can save space by resorting to brute-force enumeration" is hardly any valuable insight into WL-type hierarchies. Just the fact that "there exists a WL-hierarchy solving Graph Isomorphism which uses bounded space" is not a solid enough contribution in my opinion. Of course, it does not help that the accompanying proofs have no solid theoretical insights/contributions.
> > > >
> > > > Point 6: “Increasing $t$ from 1 to 2 also contributes hugely to the final result (by comparing the second/third to last row and the last row). This further verifies our design principle---increasing the expressive power by increasing $t$  successfully improves the results on real-world tasks.”
> > > >
> > > > I would argue that among the three experimental observations (1), (2) and (3) in Point (6), the  above-stated observation (2) needs to be strongest, since this is supposed to substantiate the main hypothesis of the paper, namely using “small-space WL" as inductive bias.
> > > >
> > > > Let us examine the (k,t)-FWL design for $k=2, t=2$, without the plus extension. In this case, the (2,2)-FWL aggregates information for a tuple $(x,y) \in V(G)^2$ using each 2-tuple $(u,v) \in V(G)^2$, thereby utilizing the current states for the 2-tuples $(u,y)$, $(x,v)$ and $(u,v)$. The plus extension essentially ensures that the chosen tuples $(u,v)$ have something to do with the tuple $(x,y)$. This is in a similar spirit to the myriad of subgraph enhancement algorithms, existing in literature. Of course, it is not exactly same as any of them, but it is not radically different in design either. This is merely due to the fact that the computationally efficient regime for WL style architectures is fairly small, and has already been heavily explored.
> > > >
> > > > On the other hand, the (2,1)-FWL aggregates information for a tuple $(x,y) \in V(G)^2$ using each vertex $u \in V(G)$, thus utilizing the current states of the 2-tuples $(x,u)$ and $(u,y)$: this is simply the 2-WL algorithm. The plus extensions of (2,1)-FWL have been already studied in literature, as local 2-WL algorithms or shortest-path-based neural architectures.
> > > >
> > > > Does the standalone fact that “(2,2)-FWL+ performs much better than 2-WL and its standard variants” adequately demonstrate the design principle of “increasing $t$” and justify the entire "small-space WL hierarchy"? I do not think so. The computationally feasible regime for WL and subgraph enhancement is fairly limited and highly special: The provided experiments are not sufficient to adequately demonstrate the main hypothesis of the paper (i.e. “bounded-space WL hierarchy”), in my opinion. At the most, the paper can claim the contribution for space-efficiency improvements.
> > > >
> > > > My final reviewer assessment is as follows: The strong performance on standard datasets such as ZINC **does not adequately justify** the "small-space WL hierarchy" hypothesis. Conversely, the "small-space WL hierarchy" **does not adequately explain** this performance on standard datasets.
> > > >
> > > > My final decision remains the same: The paper needs substantial revision. Ideally, (2,2)-FWL+ should be the center of the paper, the theoretical contribution should be the space-efficiency improvements, and the experimental contribution should be the solid results on datasets such as ZINC. In its current form, the paper does not meet the NeurIPS threshold, in my humble opinion.

---

> > > > > ### Author Response · Authors · 2023-08-19
> > > > >
> > > > > >  Just the fact that "there exists a WL-hierarchy solving Graph Isomorphism which uses bounded space" is not a solid enough contribution in my opinion.
> > > > >
> > > > > We would like to clarify again that we are not trying to solve the graph isomorphism problem. The contribution of $(k, t)$-FWL is: $(k, t)$-FWL construct **an expressiveness hierarchy** like $k$-FWL/WL but **only within a fixed space complexity $O(n^k)$** for any $k \geq 2$, different from concurrently growth of both space and time complexity in $k$-FWL/WL. To our best knowledge, there is no similar proof or even discussion in previous work. We admit that the criteria for novelty can be subjective. But we would kindly ask the reviewer to provide related works you think are relevant so that we can further clarify it for you.
> > > > >
> > > > > > The plus extension essentially ensures that the chosen tuples $(u, v)$ have something to do with the tuple $(x, y)$. This is in a similar spirit to the myriad of subgraph enhancement algorithms, existing in literature. Of course, it is not exactly same as any of them, but it is not radically different in design either..
> > > > >
> > > > > In your previous discussion, you stated that "Hence, the proposed re-parameterization in $(k, t)$-FWL is more of a subgraph enhancement GNN" (**follow-up discussion point (4)**). So we believe your main concern is in the **t**-extension instead of **+** extension. In the follow-up discussion point (5), we give a comprehensive discussion on almost all existing subgraph-based methods. None of them possess the similar idea of us.
> > > > >
> > > > > Here, your concern suddently changes to the **+** extension. Although we feel confused, we are glad to further clarify it. First, the **+**-extension states that we can use **any equivariant set** as the neighbor set, instead of only the subgraph-based aggregation. We choose a subgraph-based equivariant set in N$^2$-GNN mainly because of its provably strong substructure counting power (Theorem 4.3).  However, N$^2$-GNN is only one instance designed based on our $(k, t)$-FWL+ framework. In the general response point 3, we also evaluate **different equivariant sets like $\mathcal{SP}(v_1, v_2)$, which is totally different from subgraph based methods** and it also works well in tasks like distinguishing strong-regular graphs. This further demonstrates that the **+**-extension and $(k, t)$-FWL+ is not a subgraph enhancement algorithm but a general framework for designing expressive GNN.
> > > > >
> > > > > We admit that the criteria for novelty can be subjective. But we would kindly ask the reviewer to provide related works you think are relevant so that we can further clarify it for you.
> > > > >
> > > > > > The "small-space WL hierarchy" hypothesis
> > > > >
> > > > > We think most of your concerns may be derived from the so-called "small-space" WL. Although we already clarify it several times, we would like to underscore again that our contribution is never only the "small-space" WL. Our main contributions are (1) $(k, t)$-FWL: a theoretical-guaranteed expressive hierarchy with a fixed space complexity $O(n^k)$ for any $k\geq2$; (2) $k$-FWL+: a generalized aggregation scheme to extend WL based algorithm; (3) $(k, t)$-FWL+: a flexible and power framework to design different expressive GNN with different space/time budget; (4) N$^2$-GNN: a practical design of $(k, t)$-FWL+ that achieve SOTA results on most of benchmark datasets. We believe they are all valuable insights to the community and it is not appropriate to limit our discussion to only a subset of all points.

---

### Official Review · Reviewer_E38T · 2023-07-06

**Soundness:** 4 excellent
**Presentation:** 3 good
**Contribution:** 3 good
**Rating:** 6
**Confidence:** 2

**Summary:**

The authors propose generalizations of k-WL and k-FWL to (k, t)-FWL, and k-FWL+, which when combined, is notated as (k, t)-FWL+. They argue that (k, t)-FWL+ allow a more "flexible and fine-grained" space for exploring the graph expressiveness hierarchy, which is helpful in designing new GNN architectures. As a practical application, the authors propose Neighborhood^2-GNN which is based off a version of (2, 2)-FWL+, and outperforms state-of-the-art models on standard and synthetic tasks alike.

**Strengths:**

The paper's strengths lie in the authors' extensions of FWL, Neighborhood^2-FWL (N^2-FWL), and its strong empirical performance with regards to baselines.

The new formulation of (k, t)-FWL+ allows researchers to easily derive new extensions of FWL algorithms (and lead to new GNN models as a result). N^2-FWL is an empirical demonstration of the power of (k, t)-FWL+, showing that even with O(n^2) space complexity, these algorithms can be highly expressive, which is supported with multiple experiments.

**Weaknesses:**

The paper's weaknesses lie in its lack of clarity. Especially as a rather technical paper, there seems to be a lack of intuition-based explanations and motivations for a large portion of the paper, which makes it hard to follow at points. More figures may help as well.

In addition, the authors' do not mention any details about time in the main paper (only in the Appendix), with a time complexity of $O(nd^{h+2})$, which they claim to be practical for real-world tasks. It would be nice to see empirical run-time, as well as memory consumption charts/tables which would help solidify the empirical contribution.

Typo on line 181.

**Questions:**

1. How was N^2-FWL motivated? Why was it chosen over other possible (k, t)-WFL+ configurations? It seems as if it is not the only configuration that gives such properties.
2. How does the runtime and memory consumption compare to node-based GNN models? And to traditional edge-based subgraph models?

**Limitations:**

Some limitations are addressed, but only in the Appendix. Most points are addressed in the section in the appendix, but would prefer if it were in the main paper.

No potential negative societal impact.

---

> ### Author Rebuttal · Authors · 2023-08-07
>
> Thank you for your positive and constructive comments! We fix typo in revision and reply to all other concerns below.
>
> ### Weaknesses:
> >1. Lack of clarity.
>
> **Answer:**
> We apologize for the confusion regarding the intuitions and motivations as we want to ensure the correctness and completeness of all theoretical results in the paper. To make it easy to follow, we also provide further explanations (e.g. discussion at the beginning of Section 3.1, 3.2), examples (e.g. many definitions), and figures (Figure 1). In the general response point 1, we further clarified the main motivations and contributions of our paper. We will try our best to further improve the readability of our paper in the revision.
>
>
> >2. Further analysis on the complexity and practical usage.
>
> **Answer:**
> In general response point 2, we provide further analysis and discussion on both time and memory complexity of $(k, t)$-FWL and $N^2$-GNN.
>
>
> ### Questions:
>
> >1. The motivation  behind the $N^2$-FWL.
>
> **Answer:**
> Thanks for your great comment! The motivation behind the $N^2$-FWL is that we want to demonstrate $(k, t)$-FWL+ can be used to design a model that can fit the complexity of real-world tasks. Since most current works evaluate the model on molecular tasks, we also target on it. The success of molecular tasks highly depends on the substructure counting ability [1] and we choose $N^2$-FWL because of its provable power on substructure counting (Theorem 4.3).
>
> >2. Practical usage of $N^2$-GNN.
>
> **Answer:**
> In general response point 2, we provide further analysis and discussion on both time and memory complexity of $(k, t)$-FWL and $N^2$-GNN.
>
> **References**
>
> [1] Huang et al., Boosting the cycle counting power of graph neural networks with i$^2$-GNNs, ICLR23.

---

> > ### Comment · Reviewer_E38T · 2023-08-14
> >
> > Thank you for your answers. At the moment, there does not seem to be sufficient new information to increase or decrease my score, so I am inclined to keep it.

---

> > > ### Author Response · Authors · 2023-08-15
> > >
> > > We thank the reviewer for diligently reading our responses and giving valuable suggestions. We will try our best to further improve the clarity of the paper in the future version by providing additional examples/figures.

---

### Official Review · Reviewer_3qR2 · 2023-07-07

**Soundness:** 2 fair
**Presentation:** 2 fair
**Contribution:** 2 fair
**Rating:** 3
**Confidence:** 4

**Summary:**

The paper deals with supervised learning on graphs. Specifically, it proposes a more fine-grained version of the (folklore) k-WL (k-FWL) hierarchy. In turn, these variants are neuralized via standard techniques.

Specifically, the authors propose the $(k,t)$-FWL, which extends the $k$-FWL by the parameter $t$, which controls which tuples are used for refining the color of a tuple. That is, following Eq. (6), the authors consider $t$-tuples to control the cardinality of the neighborhood of a given $k$-tuple. Subsequently, building on **standard constructions** from the $k$-WL literature, they investigate how, for a fixed $k$, varying $t$ controls the expressivity in terms of distinguishing non-isomorphic graphs (Prop. 3.1 to 3.3).

They further propose a simple extension of the above, the $k$-FWL, which uses a relaxed version of the neighborhood between a given $k$-tuple and a vertex in the underlying graph. Furthermore, they show that some recent GNN enhancements are subsumed by their framework.

The theory is complemented by an experimental study showing that the neuralized variants exhibit SOTA predictive performance on two common benchmark datasets.




**Strengths:**

- The authors report good experimental results on established benchmark datasets. The experimental protocol looks meaningful.

**Weaknesses:**

- The paper lacks clarity. For example, the exact definition of $Q^F_w(v)$ (line 136ff) is unclear.  It is a central concept introduced in the paper, and it should be properly formalized. Moreover, the definitions in Section 3.3 are not entirely clear and make it hard to verify the formal proofs.

- The authors seem to **oversell their contribution**, e.g.,

  - The discussion on the benefits of the k-FWL over the k-WL (lines 119 to 135) is folklore. There is no new insight. This also **holds for the
  quadratic complexity of their algorithm** (e.g., lines 60 to 67). Note that big O notation is wrongly used here. Big Omega should be appropriate here.
  - Subsection 3.1 just contains a simple extension of established ideas, e.g., from [8,12].
  - Results in Section 4.2 are obvious by standard results from the GNN literature
  - ...

- The running time and memory complexity of the algorithms are **not formally proved**. This seems crucial for the present work. Also the authors seem to confuse big O with big Omega notation.
- Relevant related work such as https://arxiv.org/abs/2203.13913 seems missing.

Comments:
- It is not correct saying that the $k$-WL requires $\mathcal{O}(n^k)$ space, it is a **lower** bound not an upper bound.



**Questions:**

- What is the definition of "hierarchical multiset" (line 145)?
- Can you quantify the h in Theorem 4.1?
- What is required h for the statement of Theorem 4.3


**Limitations:**

See weaknesses.

---

> ### Author Rebuttal · Authors · 2023-08-07
>
> We thank the reviewer for the constructive comments! We will add corresponding references in the revision and reply to all other concerns below.
> ### Weaknesses:
> >1. Further clarification of the definitions.
>
> **Answer:**
> We are sorry for the confusion. Briefly speaking, $Q^F_{\mathbf{w}}(\mathbf{v})$ contains all possible tuples that pick up $m=0, \ldots, min(k, t)$ elements from a $t$-tuple $\mathbf{w}$ to replace elements in $k$-tuple $\mathbf{v}$. Due to the page limitation, we provide the formal definition of neighborhood tuple $Q^F_{\mathbf{w}}(\mathbf{v})$ in Appendix A (as indicated in line 141). To further clarify it, we also provide an example in lines 140-141 and its construction procedure in Figure 1. For definitions in Section 3.3, we assume you are confused about the definition of equivariant set $ES$ as all other notations are standard notations. We provide the formal definition of $ES$ in lines 179-181. Please let us know if there are still confusions and we are more than glad to explain them all.
>
> >2. Further clarification on the novelties and big O notation.
>
> **Answer:**
> In section 3.1, the discussion on folklore is mainly for showing the advantage of $k$-FWL over $k$-WL, i.e., we can achieve higher expressiveness with the same space complexity by redesigning the neighborhood aggregation, which is the insight and motivation behind our proposed $(k, t)$-FWL. There is no previous work, to our best knowledge, that leverages this insight further to propose similar methods as ours. In our general response point 1, contribution (1), we further clarify the contribution of the proposed $(k, t)$-FWL.
>
> The big Omega notation $\Omega(n^k)$ means the function is no less complex than $n^k$. It is usually used when we cannot prove the upper bound of a function is $n^k$. However, it is well-known that for $k$-WL, we have implementations whose complexity is exactly $n^k$, which obviously belongs to $O(n^k)$. If you are right, we believe there is no algoritm that can use the big O notation, since there definitely exists some inefficient implementation that requires larger complexity than its known lower bound.
>
>
> >3. Subsection 3.1 just contains a simple extension of established ideas, e.g., from [1,2].
>
> **Answer:**
> In subsection 3.1, we introduce the $(k, t)$-FWL, which is totally different from the idea in [1, 2]. In subsection 3.2, we introduce equivariant set $ES$ to replace the $V(G)$ used in the original $k$-WL/FWL. Methods in [1, 2] are only special cases of our proposed method. Further, as discussed in the general response point 1, contribution (2-3), this extension allows us to design high-expressiveness GNNs in a fine-grained design space. It also allows the proposed framework to unify the majority of existing high-expressiveness GNNs (**Proposition 3.4-3.9**), which is hard to achieve and new to the community. These obviously cannot be done by the methods proposed in [1, 2].
>
>
> >4. Discussion on results in Section 4.2.
>
> **Answer:**
> The reuslt in Section 4.2 (**Proposition 4.4**) is necessary for the completeness of our work. It shows that $N^2$-GNN can be as powerful as $N^2$-FWL.
>
>
> >5. further discussion on the complexity.
>
>
> **Answer:**
> In the general response point 2, we provide further analysis and discussion on both the time and memory complexity of $(k, t)$-FWL and $N^2$-GNN.
>
>
> ### Questions:
>
> >1. What is the definition of "hierarchical multiset" (line 145)?
>
> **Answer:**
> We provide the definition and an example of hierarchical multiset in line 145-147. In a hierarchical multiset $\{\{\mathbf{v} | \mathbf{v}\in V^t(G)\}\}_t$ over a $t$-tuple, elements are grouped hierarchically according to the node order of the tuple. For example, to construct $\{\{(v_1, v_2, v_3)|(v_1, v_2, v_3)\in V^3(G) \}\}_3$ from $\{\{(v_1, v_2, v_3)|(v_1, v_2, v_3)\in V^3(G) \}\}$, we first group together all elements with the same $v_2$ and $v_3$. That is,  $\forall v_2, v_3 \in V(G)$, we denote $t(v_2, v_3)=\{\{(v_1, v_2, v_3)|v_1 \in V(G)\}\}$ as the grouped result. Next, use the similar procedure, we have $\forall v_3 \in V(G)$, $t(v_3) = \{\{t(v_2, v_3)|v_2 \in V(G)\}\}$. Finally, we group all possible $v_3 \in V(G)$ to get $\{\{(v_1, v_2, v_3)|(v_1, v_2, v_3)\in V^3(G) \}\}_3 = \{\{t(v_3)|v_3 \in V(G)\}\}$.
>
>
> >2. Can you quantify the h in Theorem 4.1?
>
> **Answer:**
> $h$ is a subgraph hop hyperparameter to restrict the subgraph scope to find neighbor tuples (thus the name subgraph GNN), as also used in previous work such as $I^2$-GNN [3]. From an implementation point, it can be understood as the radius of the subgraph to extract for each tuple. It does not affect the expressive power analysis, since we can simply set $h$ to be large enough to cover the whole graph so as to match the theoretical full-graph-view setting. For SLFWL(2) [2], as they don't have the $h$ hyper-parameter and operate on the whole graph (the full-graph-view setting), Theorem 4.1 need $h$ to be large enough to cover the whole graph.
>
> >3. What is required h for the statement of Theorem 4.3.
>
> **Answer:**
> The required $h$ is the same as stated in [3] as the model need to see upon such hop to have the full picture of a particular substructure. That is, 1, 2, 2, 3, 2, 2, 1, 4, 2 for 3-cycles, 4-cycles, 5-cycles, 6-cycles, tailed triangles, chordal cycles, 4-cliques, 4-paths, and triangle-rectangles, respectively. We also use this setting in our experiments. We will include all the discussion in our revision.
>
> **References:**
>
> [1] Morris et al., Weisfeiler and leman go sparse: Towards scalable higher-order graph embeddings, NeurIPS20.
>
> [2] Zhang et al., A complete expressiveness hierarchy for subgraph gnns via subgraph weisfeiler-lehman tests, ICML23.
>
> [3] Huang et al., Boosting the cycle counting power of graph neural networks with i$^2$-GNNs, ICLR23.

---

> > ### Comment · Reviewer_3qR2 · 2023-08-11
> > **Answer to rebutall.**
> >
> > I am very well aware of the definition of the big Omega. There does not exist proof that the $k$-WL runs in $\mathcal{O}(n^k)$. Simply iterating over all possible $k$-tuples already takes $\Omega(n^k)$ time; note that does not include the aggregation over neighboring $k$-tuples.
> >
> > I do not believe that the work in its current form is ready for publication. I also strongly agree with the points mentioned by Reviewer eNLB.
> >
> > The only merit of this work is its strong empirical performance.

---

> > > ### Author Response · Authors · 2023-08-12
> > > **Follow-up discussion**
> > >
> > > Thanks for your follow-up comments. We further clarify them as follows.
> > > > Big O vs Big Omega
> > >
> > > **Answer:** First, we rewrite the update function of $k$-WL here:
> > >
> > > $ \mathcal{C}^{l-1}(\mathbf{v}) = \text{HASH}( \mathcal{C}^{l-1}(\mathbf{v}), (  \\{\\!\\!\\{ \mathcal{C}^{l-1}( \mathbf{u}) | \mathbf{u}  \in Q_j(\mathbf{v})\\}\\!\\!\\} |\ j \in [k] )) $,
> > >
> > > where $Q_j(\mathbf{v})= \\{\mathbf{v}_{w/j}|w \in V(G)\\}$
> > >
> > > and $ \mathbf{v}_{w/j} $=
> > >
> > > $\left(v_{1}, \ldots, v_{j-1}, w, v_{j+1}, \ldots, v_{k} \right)$.
> > >
> > >
> > > In our paper, we state that the $k$-WL has a space complexity of $O(n^k)$ and time complexity of $O(n^{k+1})$ (line 33-34). Now, we show there exists an implementation of $k$-WL that satisifies the above complexity. First, since the number of $k$-tuples in an $n$-node graph is $n^k$, to save the initial color of each $k$-tuple, we need $n^k$ space.
> > >
> > > At each iteration, to update the color of each $k$-tuple $\mathbf{v}$:
> > >
> > > (1) We first create a new space to save all the updated colors. This results in another $n^k$ space.
> > >
> > > (2) Then, $\forall i \in [k]$, $(v_{1}, \ldots, v_{i-1}, v_{i+1}, \ldots, v_{k}) \in V^{k-1}(G)$, we compute the result color of the aggregated multiset $\\{\\!\\!\\{\mathcal{C}^{l-1}(v_{1}, \ldots, v_{i-1}, u, v_{i+1}, \ldots, v_{k})| u \in V(G) \\}\\!\\!\\}$ and save it to a new space as an intermediate result. Since there are in total $kn^{k-1}$ different multisets and each multiset requires $n$ time to compute result, this step requires another $kn^{k-1}$ space and $kn^k$ time complexity.
> > >
> > > (3) Next, we update the color of each tuple $\mathbf{v}$. Since the color of each multiset is already computed and saved in step (2), we could use the saved result to directly update the color of each tuple $\mathbf{v}$, and save it to the space we created in step (1). This results in another $k+1$ time complexity (as there are $k$ multisets per tuple plus the tuple itself). Thus, the total time complexity in this step is $(k+1)n^k \leq (n+1) n^{k} = O(n^{k+1})$.
> > >
> > > (4) Finally, we delete all the saved colors except for the updated color for each tuple.
> > >
> > > In summary, for space complexity, we have $n^k$ (saving old color) + $n^k$ (saving new color) + $kn^{k-1}$ (intermidiate results) = $O(n^k)$. For time complexity, we have $kn^k$ (compute all possible multisets) + $O(n^{k+1})$ (update color for each $k$-tuple) = $O(n^{k+1})$. We are aware that the running time of $k$-WL can be more tightly bounded by $O(kn^k)$. However, the $O(n^{k+1})$ stated in the paper still holds.
> > >
> > > In the "Comments" of your review, you stated "It is not correct saying that the $k$-WL requires $\mathcal{O}(n^k)$ space, it is a lower bound not an upper bound." So we assume your main concern is on space complexity. Now we have proved that the $\mathcal{O}(n^k)$ space complexity is indeed feasible. Therefore, the big Omega notation need not be used.
> > >
> > > >  Strongly agree with the points mentioned by Reviewer eNLB.
> > >
> > > **Answer:** We admit that the criteria for novelty can be subjective. In the general response point 1, we further clarified the main contributions of our paper. We kindly ask the reviewer to go through the response and inform us which points you are concerned about. We are more than glad to explain them further.
> > >
> > > > The only merit of this work is its strong empirical performance.
> > >
> > > **Answer**: We believe the strong empirical performance of $N^2$-GNN further proves that our proposed $(k, t)$-FWL+ framework has great potential for designing new expressive GNN variants, as evidenced by the two new SOTA results on ZINC and ZINC-full (general response point 1, contribution (4)). We strongly believe $(k, t)$-FWL+ can inspire more novel and expressive GNNs and broaden the design space of current high-expressiveness GNNs.

---

> > > > ### Comment · Reviewer_3qR2 · 2023-08-13
> > > > **Answer**
> > > >
> > > > Thank you for the clarification, I will have another detailed look at the paper.

---

> > > > > ### Comment · Reviewer_3qR2 · 2023-08-14
> > > > > **Answer**
> > > > >
> > > > > I did reread the paper, and I do believe that the work is interesting and the empirical results are promising. However, I agree with Reviewer eNLB that the quadratic complexity result is oversold and that the main contribution lies in the subgraph enhancement, which makes the novel contribution of the work less clear.  Moreover, I still believe that the paper lacks clarity.
> > > > >
> > > > > I would encourage the authors to restructure their work and clarify where the architecture is novel over the large literature on subgraph-enhanced GNNs.

---

> > > > > > ### Author Response · Authors · 2023-08-15
> > > > > > **Follow-up discussion 2**
> > > > > >
> > > > > >
> > > > > > Thank you for your follow-up discussion. We are happy to further clarify your concern.
> > > > > >
> > > > > > >  I do believe that the work is interesting and the empirical results are promising.
> > > > > >
> > > > > > **Answer:** Thanks for the positive feedback!
> > > > > >
> > > > > > > However, I agree with Reviewer eNLB that the quadratic complexity result is oversold and that the main contribution lies in the subgraph enhancement, which makes the novel contribution of the work less clear.
> > > > > >
> > > > > > **Answer:** We thanks reviewer for the comment. We kindly refer the reviewer to our reply to the reviewer eNLB where we further clarfied the "oversold contribution" and "subgraph enhancement". Please let us know if you have any further concerns.
> > > > > >
> > > > > > > Moreover, I still believe that the paper lacks clarity.
> > > > > >
> > > > > > **Answer:** We do apology for any lack of clarity. We will try our best to make the paper more readable in the future version by providing additional figures/examples.

---

> > > > > > ### Author Response · Authors · 2023-08-19
> > > > > >
> > > > > > Dear reviewer 3qR2:
> > > > > >
> > > > > > We would like to sincerely thank you again for the time and effort spent reviewing our work and providing valuable comments. As the discussion period ends soon, could we kindly know if the responses have addressed your concerns and if further explanations or clarifications are needed? To make it easy for you, we kindly refer the reviewer to our response to reviewer eNLB for the detailed discussion on subgraph enhancement (point 5), additional ablation study on algorithms with different space complexity (point 6), and oversold contribution (point 7). If there is no further concern, we kindly hope that the reviewer could consider re-evaluating our work. Thank you again.
> > > > > >
> > > > > > Authors.

---

> > > > > > > ### Comment · Reviewer_3qR2 · 2023-08-19
> > > > > > > **Answer**
> > > > > > >
> > > > > > > Thank you, I will keep my score. I strongly believe that the work needs a major revision in terms of clarity and in terms of clarifying its actual main contribution. I also strongly support the additional detailed answers of Reviewer eNLB.

---

### Official Review · Reviewer_X7wK · 2023-07-29

**Soundness:** 3 good
**Presentation:** 3 good
**Contribution:** 4 excellent
**Rating:** 7
**Confidence:** 3

**Summary:**

The authors start by considering the k-dim Weisfeiler-Lehman test (WL). To solve the well-known problems of k-WL/FWL (high space complexity $O(n^k)$ and rigid design space), the authors propose two extensions of the k-FWL, named (k, t)-FWL and k-FWL+.
The first extension is based on the simple observation that k-WL aggregates each color separately while k-FWL aggregates the tuple of colors. Expressive results of (k,t)-FWL are provided (3.1 - 3.3). The second extension is based on the simple modification of k-FWL; instead of considering the whole vertex set, consider only a particular neighbor. These two extensions are then combined to (k, t)-FWL+, of which expressiveness results are provided compared to the previous gallery of GNN/WL models (3.5 - 3.9). Lastly, the authors then propose N2-GNN, an implementation of (2, 2)-FWL+ as well as empirical results showing the efficacy of N2-GNN.

**Strengths:**

- Despite the complex nature of the concepts, the exposition of their main algorithm and the conceptual design was very well done (especially Figure 1)

- I very much appreciate the simplicity of the idea that leads to space-efficient variants of (F)WL tests which are still quite powerful.

- Although I did not check the proofs in detail, the authors provide an impressive list of theoretical results regarding their proposed (k, t)-FWL+, all of which provide solid theoretical superiority over previous works.

- The experimental results seem promising.

**Weaknesses:**

- Although the concepts are well delivered, all technical details, especially the proofs, are entirely relegated to the Appendix. Are there any notable technical novelties in the proofs of the myriad of theoretical results, or are they somewhat standard?

- It is expected that the expressive power of (k, t)-FWL+ depends heavily on the definition of ES, which the authors addressed via a myriad of propositions (3.4 - 3.9) for specific instances. However, if possible, it would be great to have some intuitions on the interplay of the choices of (k, t) and ES. Maybe the authors could choose a single proposition and try to explain the intuition behind it further (e.g., Proposition 3.4) via additional figures or discussions.

- Some of the references are in poor condition (e.g., [11], [12], [16]).

**Questions:**

1. It states on pg. 4 lines 159-160, that "size of Q_wF(w) will grow exponentially with an increase in the size of t". Then does that mean that requiring O(n^2) is equivalent to confining t=2 (or at least small t)? Also, throughout the paper, the only time t is not 2 is in the theories when the authors prove a hierarchy w.r.t. t. Are there situations in which t should be set to 3 or higher?

2. Is there any way of comparing the expressiveness of (k, t)-FWL and (k', t')-FWL with k+t = k'+t'? For instance, can one compare the expressiveness of (2, 3)-FWL and (3, 2)-FWL?

3. Before combining (k,t)-FWL and FWL+, is it possible to also obtain expressiveness hierarchy results on FLW+ depending on the choice of ES? For instance, with fixed t=1 and k=2, can you say anything about the expressiveness across the choices of ES(v) as considered in Proposition 3.4 - 3.9?

4. It is mentioned in the conclusions that the practical complexity of N2-GNN can still be unbearable, especially for dense graphs. Thus I'm curious about the running times of all the algorithms in comparison. Is there a trade-off if one sets the training budget approximately the same for all the algorithms?

5. There was a mention of Puny et al. [43] that performs O(n^2) space 3-WL via graph polynomial features. Although the authors mentioned that they suffer from overfitting, I'm still curious. Are there any theoretical results (e.g., expressiveness hierarchy) or empirical results that compare [43] to (k, t)-FWL+?

(minor suggestion: it would be nice to have a table of contents, maybe at the start of the appendix, to make it easier to navigate)

**Limitations:**

Although this work needs not address any societal impact, the work has no mention of its limitations and possible future works. I would like to ask the authors to include some of the possible limitations of their work (possibly with possible solutions, but I won't be negative if there are none for now, which by the way is also quite natural in my eyes.)

---

> ### Author Rebuttal · Authors · 2023-08-07
>
> Thank you for your positive feedback and insightful comments! We revise all references and add contents for Appendix in our revision and reply to all other concerns below.
>
> ### Weaknesses:
> >1. The novelties in the proofs.
>
> **Answer:**
> Thanks for your mention. For all proofs, we mainly follow the previous strategies used in [1, 2]. However, we do want to highlight that most of our proof involving represent high-dimensional tuples using low-dimensional tuples, which is, to our best knowledge, not exists in any previous works and not trivial to prove. For example, in Proposition 3.7, we match the expressiveness of edge-based subgraph GNNs using $(k, t)$-FWL+ with $k=2$ (2-tuple based). However, it is well-known that edge-based subgraph GNNs are intrinsically 3-tuple-based. How to align each element in a 3-tuple to a 2-tuple-based method is non-trivial.
>
> >2. The interplay between $k$, $t$, and $ES$ and design principles.
>
> **Answer:**
> This is a great question and we would like to answer it from two parts.
>
> **Interplay between $(k, t)$ and $ES$:** In general, this is a hard question to tackle. The introduction of $(k, t)$-FWL allows us to fine-tune between the time and space complexity to achieve the desire expressive power. Increasing either $k$ or $t$ can increase the expressive power (As indicated in Proposition 3.3). The introduction of $ES$ makes the expressive power not bounded by either $k$ or $t$. For example, we can let $k=2$, $t=1$, and $ES(\mathbf{v})$ be all nodes in $4$-clique pass nodes $v_1$. This will give the model a better ability on counting this particular substructure. However, it is well-known this cannot be counted by 2-FWL or 3-WL [3].
>
> **Intuitions behind Proposition 3.4-3.9:** We think the most important principle in our mind when proving Proposition 3.4-3.9 is to find a variant that can best match the expressiveness of the existing one. Therefore, the choice of $k$, $t$, and $ES$ all serve this purpose. We do believe there exist multiple solutions for some methods and leave it to our future works.
>
>
>
>
>
> ### Questions:
> >1. Further clarification on the space complexity.
>
> **Answer:**
> The space complexity of the $(k, t)$-FWL only depends on $k$ (the size of the tuple and thus the number of tuple $|V^k(G)|=O(n^k)$). If we fix $k=2$, the space complexity should be $O(n^2)$ no matter which $t$ we use. In the situation where we only have a limited memory budget and we still want a high-expressive GNN, a large $t$ is a great solution here.
>
> >2. The comparison on the expressiveness of $(k, t)$-FWL and $(k', t')$-FWL with $k+t = k'+t'$.
> >
> **Answer:**
> Thanks for your insightful comment! We do believe there exists a further relationship between $(k, t)$-FWL and $(k', t')$-FWL with $k+t = k'+t'$ and had some attempts to prove that. However, we find it non-trivial to give formal theoretical proof and leave it to our future works.
>
> >3. The expressiveness of $ES$.
>
> **Answer:**
> Thanks for your great question! As we mentioned in the answer to weakness 2, the choice of $ES$ is the key to the final expressive power of the model and its power will not be limited by either $k$ or $t$. In general response point 3, we further provide additional ablation study and show the practical power of different choices of $ES$. However, currently, it is hard to provide a canonical characterization of the expressive power of different $ES$.
>
> >4. Practical usage of $N^2$-GNN.
>
> **Answer:**
> In general response point 2, we provide further analysis and discussion on both time and memory complexity of $(k, t)$-FWL and $N^2$-GNN.
>
>
> >5. Discussion on Puny et al [4].
>
> **Answer:**
> Thanks for your insightful comment, [4] Introduce a new way to evaluate the expressive power of the graph model---the ability of the model to compute the equivariant polynomials up to a certain degree. However, as mentioned in [4], the equivariant polynomial is highly related to subgraph counting. Theoretically, we think equivariant polynomial is more suitable to compare with models like CIN [5]. It is hard to directly connect $(k, t)$-FWL+ with equivariant polynomials. Empirically, The comparison depends on the implementation of $(k, t)$-FWL+ and the choice of equivariant polynomial. Puny et al [4] do introduce PPGN++, which adds additional polynomial features up to 6 degrees that cannot be computed by PPGN and test its performance on ZINC-subset (0.071) and ZINC-full (0.020), which is lower than $N^2$-GNN.
>
> ### Limitations:
> Thanks for your great question! Due to the page limitation, we mainly discuss the limitation of $N^2$-GNN in Appendix D.3. Specifically, (1) although $N^2$-GNN holds its theoretical space complexity, it could still be memory-costly in dense graph due to the way of implementation. This can be solved by changing the way of implementation. (2) The model become hard to optimize with large $t$ and dense graph. The key reason is how to realize the injectiveness of multiset pooling function in neural version. So far, we only test the summation. We will investiage more powerful multiset pooling function like DeepSet [6] in the future. These are all future topic in this work. Meanwhile, as you mentioned, the in-depth theoretical analysis on different $ES$ is also very interesting and worth further investigation! We will provide complete discussion on the limitation and future work in the revision.
>
> **References:**
>
> [1] Morris et al., Weisfeiler and leman go sparse: Towards scalable higher-order graph embeddings, NeurIPS20.
>
> [2] Zhang et al., A complete expressiveness hierarchy for subgraph gnns via subgraph weisfeiler-lehman tests, ICML23.
>
> [3] Arvind et al., On Weisfeiler-Leman invariance: Subgraph counts and related graph properties. Journal of Computer and System Sciences, 2020.
>
> [4] Puny et al., Equivariant polynomials for graph neural networks, ICML23.
>
> [5] Bodnar et al., Weisfeiler and Lehman Go Cellular: CW Networks, Neurips21.
>
> [6] Zaheer et al., Deep Sets, NeurIPS17.

---

> > ### Comment · Reviewer_X7wK · 2023-08-11
> >
> > I thank the authors for providing detailed responses to my questions and comments. I'm satisfied with the responses and intend to keep my score.

---

> > > ### Author Response · Authors · 2023-08-11
> > >
> > > We thank you again for your constructive feedbacks and suggestions!

---

### Author Rebuttal · Authors · 2023-08-07

We thank all the reviewers for their constructive and insightful comments. Here we respond to some general concerns mentioned by the reviewers. **All tables mentioned in the response are provided in the PDF**.

### 1. Further clarify the motivations and contributions.

**Motivations:** We try to tackle two intrinsic limitations of $k$-WL/FWL. Namely, (1) the concurrent growth of both the time and space complexity when increasing $k$ and (2) the huge expressiveness gap between two consecutive values of $k$.

**Contributions:**
(1) To tackle the first limitation, we propose $(k, t)$-FWL, which replaces the nodes to traverse (used for defining a $k$-tuple's neighbors) with $t$-tuples. We theoretically show that by varying $t$, we can construct an expressiveness hierarchy with a fixed $k$ (fixed space complexity) (**Proposition 3.1; 3.3, Theorem 3.2**). Although it is true we can solve the graph isomorphism problem by permuting all possible mappings between two graphs ($O(n^2)$ space complexity), there is no previous work theoretically establish an expressiveness hierarchy on a fixed space complexity---this hierarchy enables **flexibly tuning the expressive power** (with theoretical guarantee) in a fixed space budget, which is **not feasible** by brute-force methods.


(2) To tackle the second limitation, we propose $k$-FWL+. In $k$-FWL+, we introduce the equivariant set $ES$ to replace the $V(G)$ used in the original $k$-WL/FWL. Although previous works [1, 2] also modified the $V(G)$, they only introduce a particular case, like 1-hop neighboring nodes of the $k$-tuple. Instead, we **generalize it to a much broader space**. This brings two advantages, (1) a new insight into how to design GNNs based on $k$-WL/FWL using fine-grained ES definitions and (2) unification of a wide range of previous GNNs. These cannot be achieved by any previous work like [1, 2].

(3) Combining (1-2) results in a new framework named $(k, t)$-FWL+. $(k, t)$-FWL+ is a very flexible framework that can unify the majority of previously proposed high-expressive GNNs (**Propositions 3.4-3.9**). Based on the $(k, t)$-FWL+ framework, we propose $N^2$-FWL and its neural version $N^2$-GNN, which is provably more powerful than all node-based subgraph GNN and SLFWL(2)[2], as well as existing edge-based subgraph GNNs [3] (**Theorems 4.1-4.2**). We also prove its strong substructure counting power in **Theorem 4.3**.

(4) We evaluate $N^2$-GNN on many benchmark datasets. $N^2$-GNN achieves **new SOTA results** on ZINC-subset/full datasets with the **smallest parameter size** (306k in ZINC-subset/ZINC-full compared to ~500k of other models). Specifically, on ZINC-subset, we improve the previous SOTA MAE **from 0.066 (Specformer) to 0.059**, which is the first time a model can reach below 0.06 MAE. On ZINC-full, the advantage is even huger, **with 0.013 new SOTA MAE** compared to the previous 0.022. These results are **significant enough** for the community, and further demonstrate the power and flexibility of $(k, t)$-FWL+ on designing GNNs fitting the complexity of real-world tasks.


### 2. Analysis of time and space complexity of the proposed methods.

**Theoretical analysis**: For $N^2$-GNN, we provide complexity analysis in Appendix D.2. For $(k, t)$-FWL, the space complexity is $O(n^k)$ as we only need to store representations of all $k$-tuples. For time complexity, if $k$ and $t$ are relatively small, to update the color of each $k$-tuple, $(k, t)$-FWL needs to aggregate all $V^t(G)$ possible neighborhood tuples, resulting in $O(n^{k+t})$ time complexity. If both $k$ and $t$ are large enough (related to $n$), the time complexity further increases to $O(n^{k+t}\cdot m\cdot(\frac{q!}{m!})^2)$, where $m=min(k, t)$ and $q=max(k, t)$.  For $k$-FWL+ and $(k, t)$-FWL+, the space complexity is $O(n^k)$. However, since the time complexity highly depends on the choice of $ES$, it is hard to provide a formal analysis for it.

**Empirical study**: Here we show the practical usage of $N^2$-GNN (Table 1, 2) using the cycle counting datasets with a similar parameter size for all models and a batch size of 256. We report the training time, inference time, and maximum memory usage during inference for both $h=1$ and $h=2$ for an average of 20 epochs.

We can see that the empirical memory usage of $N^2$-GNN is almost the same as $I^2$-GNN and the time is worse. As we mentioned in limitations (Appendix D.3), to enjoy some extent of parallelism, the current implementation of $N^2$-GNN needs to save all neighbor indices, which introduces a constant factor of more memory. However, we do want to highlight several points: (1) By changing the way of implementation, $N^2$-GNN can be implemented strictly within $O(n^2)$ space. Therefore, the merit of $N^2$-GNN still holds. (2) The experiment is conducted on the same parameter budget level. However, in real-world tasks, $N^2$-GNN needs much fewer parameters to achieve SOTA results. (3) In this work, we not only aim to propose a new model but also mean to introduce a new flexible framework, $(k, t)$-FWL+. The empirical success of $N^2$-GNN demonstrates the great potential of this framework for designing new expressive GNN variants.


### 3. Additional ablation study.

We provide additional ablation study on $(2, t)$-FWL+ by varying different $t$ and $ES$ (Table 3 and 4). We select two important $ES$ mentioned in our paper---$Q_1(v)$ and $\mathcal{SP}(v_1, v_2)$. Notes that we find the performance of $\mathcal{SP}$ or $\mathcal{SP} \times \mathcal{SP}$ on counting dataset is similar to MPNN and thus omit it in Table 4.

**References:**

[1] Morris et al., Weisfeiler and leman go sparse: Towards scalable higher-order graph embeddings, NeurIPS20.

[2] Zhang et al., A complete expressiveness hierarchy for subgraph gnns via subgraph weisfeiler-lehman tests, ICML23.

[3] Huang et al., Boosting the cycle counting power of graph neural networks with i$^2$-GNNs, ICLR23.

---

### Decision · Program_Chairs · 2023-09-21

**Decision:**

Accept (poster)

**Comment:**

This paper is an attempt to address two limitations in the existing expressiveness hierarchy $k$-WL/FWL: 1) they require both the time and space complexity to grow simultaneously, 2) the only adjustable parameter is $k$ and there is no other fine-grained knob. To address the first limitation, the authors propose $(k,t)$-FWL, a novel hierarchy that allows one to increase expressive power while fixing space complexity (by fixing $k$ and increasing $t$). The second limitation is addressed by $k$-FWL+, where the authors replace the vertex set with "equivariant set" to allow for more flexibility. The paper then proceeds to combine the two variations to yield $(k,t)$-FWL+, and show that many existing works can be subsumed by the new framework. The authors then proceed to propose and examine a specific example of $(2,2)$-FWL+ named $N^2$-FWL and its GNN version. The proposed architecture shows promising performance, achieving the SOTA for the ZINC dataset.

There were active and extensive discussions between the authors and reviewers, and unfortunately, even after the discussion phase, the evaluations stayed divergent. After carefully reading the reviews, rebuttals, and the follow-up discussions, my conclusion is that the merits of this paper outweigh the shortcomings.

As pointed out by the reviewers, the paper has some downsides, such as overselling the space complexity and limited novelty. For example, it was pointed out that solving the graph isomorphism problem in $O(n^2)$ space is possible in a brute-force way (if time is not a problem). However, I am convinced by the authors' response that the paper's key contribution is in the novel hierarchy with fixed space complexity (which could be any $n^k$, not just $n^2$) rather than achieving the narrow goal of solving the graph isomorphism problem in $n^2$ space. As for proof novelty, too, I believe that the novelty lies in the new framework, so the fact that the proofs follow from standard known results does not look like a major problem to me. Reviewer eNLB questioned the paper's novelty due to connection to subgraph enhancement GNNs, but I am more or less convinced by the authors' detailed responses that the proposed framework contains novel elements.

That said, there is certainly room for improvement that the authors should address in the revision. As pointed out by multiple reviewers, the main body does not discuss the time complexity in detail and defers discussions of limitations to the appendix; such limitations ("no free lunch") should be better highlighted/discussed in the main text. Two reviewers commented that clarity needs to be improved, and I also found it quite hard to read the paper; for example, I would advise against deferring key mathematical definitions such as $Q^F_w$ to appendices. Please also make sure to include ablation studies and runtime analysis in the revision, as they were also brought up in the reviews.